ARTICLES

# OPEN
# The microbiome of a bacterivorous marine choanoflagellate contains a resource-demanding obligate bacterial associate

David M. Needham [1,2] ✉, Camille Poirier[1,2], Charles Bachy [1,2], Emma E. George[3], Susanne Wilken[2,4,5], Charmaine C. M. Yung[1,2,9], Alexander J. Limardo[2,5], Michael Morando [5], Lisa Sudek[2], Rex R. Malmstrom [6], Patrick J. Keeling [3], Alyson E. Santoro [7] and Alexandra Z. Worden [1,2,5,8] ✉

Microbial predators such as choanoflagellates are key players in ocean food webs. Choanoflagellates, which are the closest unicellular relatives of animals, consume bacteria and also exhibit marked biological transitions triggered by bacterial compounds, yet their native microbiomes remain uncharacterized. Here we report the discovery of a ubiquitous, uncultured bacterial lineage we name *Candidatus* Comchoanobacterales ord. nov., related to the human pathogen *Coxiella* and physically associated with the uncultured marine choanoflagellate *Bicosta minor*. We analyse complete 'Comchoano' genomes acquired after sorting single *Bicosta* cells, finding signatures of obligate host-dependence, including reduction of pathways encoding glycolysis, membrane components, amino acids and B-vitamins. Comchoano encode the necessary apparatus to import energy and other compounds from the host, proteins for host-cell associations and a type IV secretion system closest to *Coxiella's* that is expressed in Pacific Ocean metatranscriptomes. Interactions between choanoflagellates and their microbiota could reshape the direction of energy and resource flow attributed to microbial predators, adding complexity and nuance to marine food webs.

Microbe–microbe interactions are important drivers of ecological and evolutionary trajectories in marine environments[1,2]. These interactions underpin trophic transfer of energy and carbon cycling, both being of key concern for understanding potential shifts in community assembly and ecosystem function during changing ocean conditions[1,3]. Protistan bacterivory via phagocytosis has a broadly appreciated role in transfer of energy to higher trophic levels in marine ecosystems[1,4]. Over evolutionary time scales, the simplicity of such predator–prey relationships and resultant energy transfer have been altered in cases of phagocytic engulfment that resulted in either temporary or permanent host–bacterium symbioses[5,6]. However, identification of symbioses, which encapsulate a range of relationships along a continuum from mutualistic to antagonistic[7], has been impeded by challenges in capturing or culturing physically interacting marine microbes, inhibiting the advancement of this critical area of ocean science[1,3].

Among important marine bacterivores[8,9], choanoflagellates have recently been noted for their global distribution[10] and interactions with bacteria. These interactions extend beyond consumption of bacteria as food, to laboratory demonstrations that bacterially-derived compounds initiate both the transition to multicellularity[11,12] and sexual reproduction[13] in choanoflagellates. However, little is known about distributions of specific choanoflagellates or their ecological associations with individual bacterial lineages in the natural environment. Specifically, insights into the existence, nature and consequence of specific cell–cell interactions that extend beyond known predator–prey interactions with bacteria are important for moving past correlative approaches that neither capture the directionality of exchanges in interactions nor hold under non-steady state conditions. Understanding of co-associations and ecological ramifications are essential for characterizing the mechanistic underpinnings of interactions[1,2] that can then be used to improve carbon cycle models and the movement of organic carbon in the ocean.

We hypothesized that the microbiomes of choanoflagellates would reveal a variety of interactions, such as predator–prey or potentially microbial symbioses, with giant viruses of choanoflagellates having been discovered recently[14,15]. To discover such cell–cell interactions without inducing possible selection through culturing, we used Fluorescence Activated Cell Sorting (FACS)[14] to isolate single choanoflagellate cells from Pacific Ocean surface waters.

## Results

**Sorting wild marine choanoflagellates and microbiome analyses.** Scattering properties and labelling of acidic food vacuoles with a fluorescent probe identified a population of active choanoflagellate cells that were then sorted (Fig. 1b). After DNA amplification, 18S rRNA gene V4 amplicon sequencing showed that 90% of 188 sorted choanoflagellate cells had 100% nucleotide identity across the 378 bp of the amplicon sequence to the *Bicosta minor* gene sequence (KU587839[16]). To understand more about the distribution of this uncultivated choanoflagellate, we examined two large global ocean survey datasets, *Tara* Oceans and Malaspina[10,17]. Classification of 18S rRNA amplicons showed *Bicosta* in all *Tara* Ocean surface samples[10], where they made up an average of 12% of the choanoflagellate

[1]Ocean EcoSystems Biology Unit, GEOMAR Helmholtz Centre for Ocean Research, Kiel, Germany. [2]Monterey Bay Aquarium Research Institute, Moss Landing, CA, USA. [3]Department of Botany, University of British Columbia, Vancouver, British Columbia, Canada. [4]Institute for Biodiversity and Ecosystem Dynamics, University of Amsterdam, Amsterdam, the Netherlands. [5]Ocean Sciences Department, University of California, Santa Cruz, CA, USA. [6]DOE Joint Genome Institute, Berkeley, CA, USA. [7]Department of Ecology, Evolution and Marine Biology, University of California, Santa Barbara, CA, USA. [8]Max Planck Institute for Evolutionary Biology, Plön, Germany. [9]Present address: Department of Ocean Science, The Hong Kong University of Science and Technology, Clear Water Bay, Kowloon, Hong Kong. ✉e-mail: dneedham@geomar.de; azworden@geomar.de

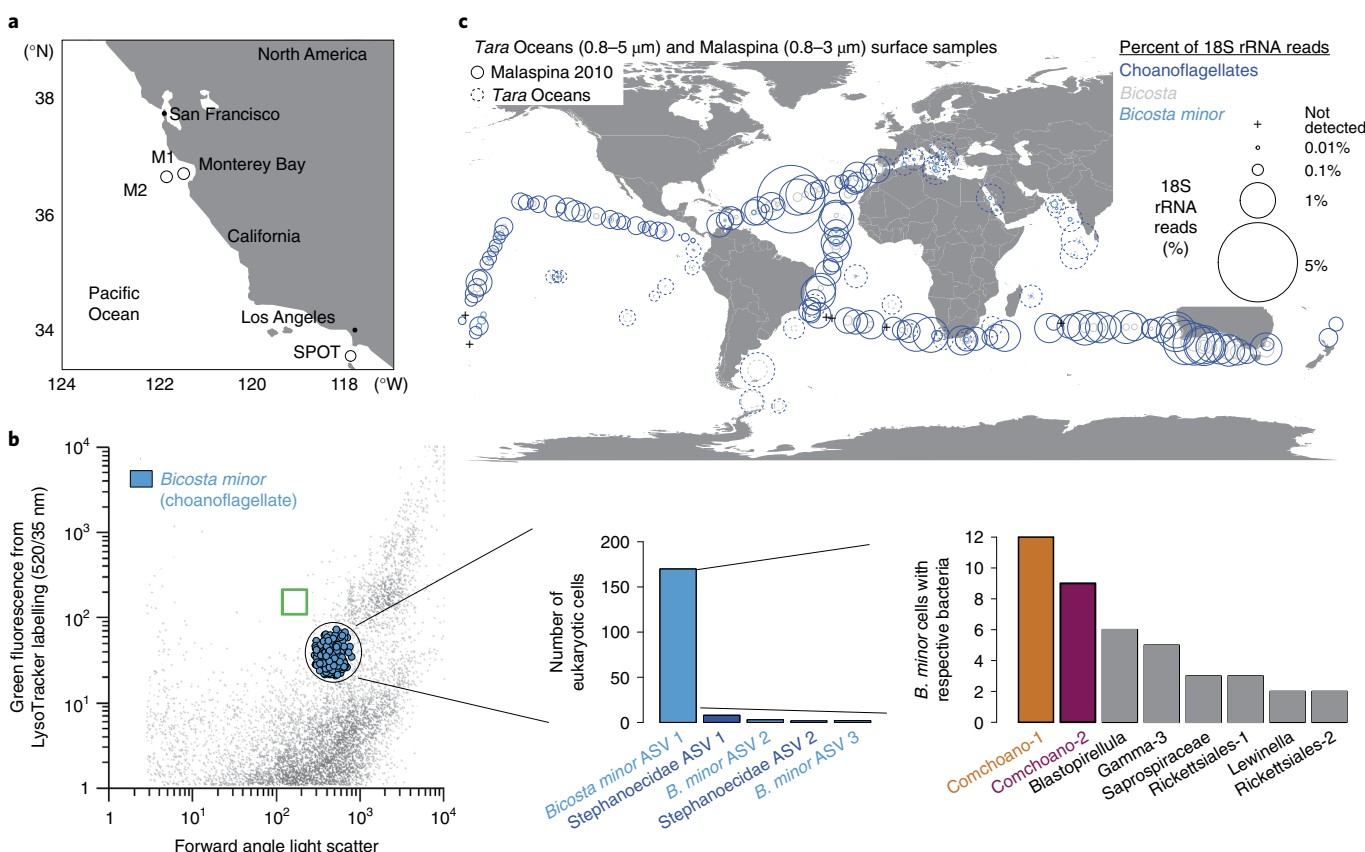

**Fig. 1 | Identification of a physical association between the uncultivated choanoflagellate *B. minor* and a divergent Gammaproteobacterial lineage.**
**a**, Locations of the single-cell sorting experiment (Station M2), metatranscriptome sequencing (Stations M1 and M2) and amplicon-based rRNA gene sequencing surveys performed herein at the Monterey Bay Time Series (MBTS; M1, M2) and amplicon data analysed from the San Pedro Ocean Time-series (SPOT). **b**, Left: population of choanoflagellates, which belong to the Opisthokont supergroup that includes metazoans[37], sorted by FACS on the basis of forward angle light scatter (proxy for cell size), fluorescence labelling of food vacuoles and absence of fluorescence from photosynthetic pigments. The green box indicates the position of bead standards run before and after sorting at the same settings. Middle: choanoflagellates sorted were almost exclusively *B. minor* (family Salpingoecidae). Here, '*B. minor* ASV 1' (100% nucleotide identity to uncultivated *B. minor*) refers to the dominant *B. minor* ASV (90% of all choanoflagellate cells), while the others are less frequently observed choanoflagellate ASVs. Right: the most common bacteria detected with the dominant *B. minor* ASV (only bacterial ASVs present with more than two *B. minor* cells are shown) comprised two Gammaproteobacteria (Comchoano-1 and Comchoano-2) and a third, less common Gammaproteobacterium (86% 16S rRNA amplicon identity to Comchoano with 2% of cells), alongside a Planctomycete (*Blastopirellula*, 93% identity to *Mariniblastus fucicola* with 3% of cells), two Flavobacteria (Saprospiraceae and *Lewinella* 93% and 91% identity to closest relatives *Membranicola marinus* and *Portibacter lacus*, respectively, with 2% and 1% of cells) and two Rickettsiales (87% identity to an endosymbiont of *Oligobrachia haakonmosbiensis* with 2% and 1% of cells, respectively). **c**, Oceanic distributions of choanoflagellates, *Bicosta* and *B. minor* as classified by QIIME 2[71]. Relative abundances in surface samples are represented as percentage of 18S rRNA gene V4 amplicons from Malaspina 2010 (3 m depth, solid circles) and V9 amplicons from *Tara* Oceans (5 m depth, dashed circles) circumnavigations, with metazoan sequences (about 0.001% on average) excluded. Data were summed across ASVs (Malaspina) or OTUs (operational taxonomic unit; *Tara* Oceans) for the taxonomic level assigned by QIIME 2. Choanoflagellates were not detected in six samples (all from Malaspina).

community, and in 54% of Malaspina samples, where they made up 7% of the choanoflagellate community (Fig. 1c). Thus, *B. minor* appears to be of relevance not only to coastal and fjord environments where it was previously reported[16], but also in the open ocean.

To determine whether a microbiome exists in individual choanoflagellate cells, we sequenced 16S rRNA gene V4 amplicons from the sorted cells using primers that exclude the most probable bacterial prey, that is, the abundant surface ocean bacteria, SAR11. Low bacterial diversity was observed from the *B. minor* cells, with 28% exhibiting bacterial amplicon sequence variants (ASVs) at an average of $1.6 \pm 1.3$ ASVs per *B. minor* cell. In total, 22 ASVs were detected, eight of which were associated with *B. minor* in more than one instance (Fig. 1b). The most common microbiome members were represented by two Gammaproteobacterial ASVs that had low nucleotide identity (89–90%) to the closest cultured bacteria, members of the Coxiellales order, among which is the animal pathogen

*Coxiella burnetii* responsible for Q fever in humans[18]. These two divergent Gammaproteobacteria were present in 12% of sorted *B. minor* cells and were detected only once in the same *B. minor* cell. The low average number of bacterial ASVs ($1.6 \pm 1.0$) in *B. minor* cells harbouring one of the two Gammaproteobacteria indicated extremely low community diversity within these *B. minor* cells. Inspired by the association between the microbiome dominants and the choanoflagellate, we tentatively named these two bacteria 'Comchoano', meaning 'with choano(flagellate)'.

We next sequenced genomic DNA from the sorts exhibiting Comchoano co-associated with *B. minor* cells. The low diversity within the single *B. minor* cells facilitated assembly of the full-length *B. minor* 18S rRNA gene (1,568 out of 1,569 bp identity to KU587839[16]) as well as full-length Comchoano 16S rRNA gene sequences. In each case, the respective full-length rRNA genes were always identical to the respective ASVs. Phylogenetic analyses of the

full-length Comchoano 16S rRNA gene sequences showed that they belonged to a divergent, diverse and entirely uncultured marine lineage and confirmed that their closest cultured relatives belong to the Coxiellales order. The overall clade to which Comchoano belonged was statistically supported and otherwise comprised members of the enigmatic UBA7916 order (Fig. 2a) recognized in 2018[19].

**Geographic distribution and interaction ecology.** Surprised by the co-association between this uncultivated marine bacterial lineage and wild choanoflagellate cells, we next mapped Comchoano distributions. Phylogenetic placement of amplicons from a variety of marine environments[17] demonstrated that, like *B. minor* (Fig. 1c), they are globally distributed (Extended Data Fig. 1a) albeit at low relative amplicon abundances. We then turned to time-resolved data to gain insight into ecological relationships, specifically to data collected daily-to-weekly at SPOT, a time-series study in Pacific waters of the Southern California Bight (Fig. 1a). Across the daily portion of the time-series, Comchoano and *B. minor* relative amplicon abundances increased contemporaneously in the 'protistan size fraction' (1–80 μm) ($r = 0.73$, 0.80; $P = 0.001$, 0.0002 for Comchoano-1 and 2, respectively; Fig. 2b and Supplementary Data 1). Moreover, across the full daily-to-monthly time-series at SPOT (53 samples over 6 months), pairwise Spearman correlation analyses showed that in the protistan size fraction, Comchoano relative abundance correlates with choanoflagellates ($P$ and $q$ <0.05), with higher percentage of correlations to choanoflagellates than to other eukaryotes apart from telonemids (Extended Data Fig. 1b,c). However, in the 'free-living prokaryote fraction' (0.2–1 μm) of the daily time-resolved part of the time-series, Comchoano 16S rRNA amplicon relative abundances increased as *B. minor* 18S rRNA amplicon relative abundances decreased. Thus, a significant time-lagged positive correlation was observed between *B. minor* in the protistan size fraction and Comchoano in the free-living prokaryote fraction (2 d time-shifted $r = 0.62$, 0.78; $P = 0.019$, 0.0011 for Comchoano-1 and 2, respectively; Supplementary Data 1). These results suggest that Comchoano were potentially released from lysed or dying *B. minor* cells (depending on the limitations of relative abundance-based analyses), and the time-delayed increase in Comchoano is consistent with a lytic parasitic or pathogenic role, akin to extensions of Lotka-Volterra equations for host and parasite[20], or could arise by other phenomena whereby the choanoflagellate decreases in abundance while Comchoano are released into the environment. The low relative abundances of Comchoano in global ocean data at SPOT and at Stations M1 and M2 (Fig. 1) indicate that they are not probable prey items. Indeed, at the more northerly Pacific stations where sorting was performed, monthly data showed that relative abundances of Comchoano-1 and 2 were approximately 500 and 150 times lower than that of the most abundant bacterial taxon, respectively (Extended Data Fig. 2a,b). Despite their rarity among other bacteria, the frequency of Comchoano co-associations with *B. minor* (12% of *B. minor* cells) is comparable to other co-associations known to have major biogeochemical impacts[21]. For example, recent estimates suggest that about 1% of unicellular plankton in the ocean are typically infected by viruses[22,23] and about 25% are infected during blooms of specific taxa when virus–host encounter rates are higher[22]. The observed environmental distributions, the fact that Comchoano are physically co-associated with choanoflagellates, the relatively long-branch relationships[24] with other bacteria and their phylogenetic proximity to bacterial pathogens of animals[18] all point to Comchoano being obligately host associated, in this case with *B. minor*.

**Complete genomes of Comchoano-1 and Comchoano-2.** We found that the genomes of Comchoano were starkly different from those of common free-living marine bacteria. Assembly efforts rendered a single consensus circular genome of 1.01 Mb for

Comchoano-1 (Fig. 3a) and two consensus contigs of 1.07 Mb for Comchoano-2 (Extended Data Fig. 3 and Supplementary Data 2). Comchoano genomes are small relative to most marine prokaryotes, smaller than all but one (archaeal) metagenome-assembled genome (MAG) among available marine MAGs, single amplified genomes (SAGs) and cultured prokaryotic genomes (see Methods for dataset description) considered to be as complete as Comchoano (Fig. 3b and Extended Data Fig. 4a). With respect to predicted protein numbers, Comchoano-1 encodes 951 and Comchoano-2 encodes 1,004 proteins, fewer than the genome-streamlined free-living bacterial lineage SAR11[25], which has ~40% more predicted proteins. Estimates suggest that the Comchoano genomes are 96.6% complete based on predictions from single-copy gene (SCG) counts[26]. However, multiple lines of evidence contrast with a prediction of 'incompleteness': (1) the circularized Comchoano-1 genome sequence, (2) clear origin and terminus of replication in both Comchoano genomes, (3) nearly identical genome assemblies from across multiple single (*B. minor*) cells, including identical 16S rRNA gene sequences (Supplementary Data 2) and (4) the observation that Comchoano genomes always lack the same two SCGs, which are also commonly missing from other UBA7916 lineage members (Supplementary Data 3). By these measures, we concluded that the Comchoano consensus genome sequences are effectively complete, providing a springboard for investigation of gain and loss of functions in this putatively obligately associated lineage.

The level of Comchoano-1 and 2 genome completeness is high compared with most marine SAG assemblies so far (Extended Data Fig. 4b,c)[27], providing potential insights into how the interaction between Comchoano and *B. minor* manifests. The results point to the presence of multiple Comchoano cells in each sorted *B. minor*, which would minimize the impacts of multiple displacement amplification (MDA) bias. The assemblies clearly benefited from low sequence variation in Comchoano genomes. Specifically, we observed $0.49 \pm 0.34$ and $27 \pm 38$ single nucleotide polymorphisms (SNPs) per 100 Kb based on read mapping to the Comchoano-1 consensus genome assembled from different *B. minor* cells and the Comchoano-2 consensus genome, respectively (Fig. 3c and Supplementary Data 2). Only one SNP occurred more than once across eight Comchoano-1 genomes (each from a different *B. minor* cell), indicating that they are essentially clonal. The observed population structure for these dominant *Bicosta*-microbiome members contrasts with the extensive microdiversity in abundant free-living marine bacteria—a factor considered important to niche breadth, population stability and frequency-dependent interactions, such as host–virus dynamics[27–30]. Our results demonstrate that the evolutionary pressures acting on Comchoano differ from those acting on most known free-living pelagic bacteria[31].

**Evolutionary relationships within Gammaproteobacteria.** With the Comchoano genomes in hand, we resolved evolutionary relationships between this unique lineage, UBA7916 and other Gammaproteobacteria using phylogenomic approaches. The phylogenomic reconstruction provided robust statistical support for Comchoano placement within the UBA7916 order (Extended Data Fig. 5a,b), as first indicated by full-length 16S rRNA gene analysis (which also included environmental 16S rRNA sequences for which no genomic information is available; Fig. 2a). The relative evolutionary distance (RED) value[19] between Comchoano and existing UBA7916 genomes (0.83) suggests that Comchoano represent a family level of divergence within UBA7916. The genomes do retain some large-scale syntenic patterns (Extended Data Fig. 5c) and the amino acid identity between Comchoano-1 and Comchoano-2 is 49% based on comparison of homologous proteins. Together, these results suggest that despite their close relationship to each other relative to other sequenced bacteria (Fig. 2a and Extended Data Fig. 5a,b), the two Comchoanos are at least separate genera, a conclusion

supported by the 16S rRNA gene nucleotide sequence identity level (95%).

The overall clade representing the UBA7916 order incorporates multiple long-branching, uncultivated Gammaproteobacteria represented only by marine SAGs and MAGs (Extended Data Fig. 5b) which, similar to Comchoano, have smaller than average estimated genome sizes compared with other marine prokaryotes (Fig. 3b and Extended Data Fig. 4a). Small genome sizes are one indicator of obligate endosymbionts and pathogens[6], including some Coxiellales such as the *Coxiella* endosymbiont (0.82 Mb) of *Amblyomma americanum*, a tick species (Supplementary Data 4). Until now, the UBA7916 order has remained ecologically and biologically mysterious. This is largely because no UBA7916 member has been cultured—a situation we now hypothesize persisted due to the disruption of physical associations between these bacteria and possible eukaryotic hosts during most studies or culturing efforts, or selection for other types of bacteria that thrive as free-living cells under culture conditions. The partial MAGs and SAGs available from UBA7916 most closely related to Comchoano (for example, the marine bacterial SAG CACOCF) may therefore also have come from bacteria with a host or even choanoflagellate-associated lifestyle. While lifestyle attributes or associations of UBA7916 members apart from Comchoano remain unknown, the Coxiellales—which again branched as the closest cultivated relatives—and other related orders, including the Berkiellales, Diplorickettsiales and Legionellales (Extended Data Fig. 5a,b), are noted pathogens of insects and terrestrial mammals, as well as amoebozoan and ciliate protists[32–34] unrelated to choanoflagellates. Additionally, a long-branching group (*Candidatus* Azoamicus) in an unsupported position adjacent to Legionellales and Francisellales (with 84% nucleotide identity to Comchoano) is an obligate endosymbiont of an anaerobic ciliate from waste sludge and other freshwater environments where the much-reduced bacterium appears to provide energy to the host, seemingly akin to mitochondria[35]. Alongside small genome sizes, long-branching phylogenetic relationships[24] with other bacteria and low microdiversity are well known features of endosymbionts found in eukaryotes[6].

**Metabolic traits and pathway reductions in Comchoano.** To this point, our findings suggested an obligate association between Comchoano and *B. minor*, leading us to consider the molecular mechanisms needed to support a host-associated lifestyle, the nature of the putative association, and its possible extension to other UBA7916 lineage members. In regard to central cellular biochemistry, Comchoano and UBA7916 have most of the key proteins associated with replication and translation, including a full complement of transfer RNAs and tRNA synthetases, as well as nearly all ribosomal and cell division-related proteins (Fig. 3d, Supplementary Data 5 and 6, and Extended Data Figs. 6 and 7). Additionally, similar to many other marine bacteria, Comchoano and most other UBA7916 encode photoactive microbial rhodopsin proteins (Extended Data Fig. 8). These contain motifs suggestive of a proton-pumping func-

tion with various hypothesized roles, including energy transfer or acidification of host cellular compartments[14,36]. Comchoano do not encode genes for biosynthesis of retinal, the chromophore required for rhodopsin function (Fig. 3d). Thus, Comchoano would need to scavenge the chromophore from prey ingested by the host, akin to chromophore scavenging that has been demonstrated for the cultured choanoflagellate *Choanoeca flexa*[37], or Comchoano could potentially produce retinal in an uncharacterized manner. Strikingly, however, Comchoano central metabolism is disrupted. Multiple glycolysis enzymes are absent, specifically the first two and the last three in the pathway, similar to presence/absence patterns seen in other UBA7916 (Fig. 3d and Extended Data Fig. 9a). Two of the three glycolytic proteins that are retained, 6-phosphofructokinase and fructose-bis phosphate aldolase, are also essential to the pentose phosphate pathway (PPP), which is fully encoded, as are the tricarboxylic acid (TCA) cycle and oxidative phosphorylation (Fig. 3d and Extended Data Fig. 7). Utilizing the pentose phosphate pathway would require fructose-6-phosphate from the choanoflagellate (which can produce it via glycolysis, Supplementary Data 7) and the product, 3-phosphoglycerate, could then be used in processes such as amino acid biosynthesis. In place of losses of function for traditional glycolysis, Comchoano may meet energy demands using an ATP/ADP translocase that they possess, which can directly import ATP from the host environment. This has been reported as the mechanism by which obligate endosymbiotic bacteria[38,39] obtain energy from the host (for example, the obligate intracellular pathogen *Chromulinavorax destructans* found in a freshwater stramenopile protist[40]). The ATP/ADP translocase encoded by Comchoano and most UBA7916 (Fig. 3d and Extended Data Fig. 6) is phylogenetically similar to those reported in other 'energy parasites' (Extended Data Fig. 10a,b)[38]. Some translocases within this broad protein family have been reported to have affinity for nucleotides such as guanosine di/tri-phosphate, calling for further experimental studies of translocase affinities[35,41]. Recently, this translocase was implicated as the mechanism by which *Ca.* Azoamicus endosymbionts provide ATP to their freshwater ciliate hosts. Here we find that the closest affiliation of the Comchoano translocases is with an ATP/ADP translocase in an alphaproteobacterial endosymbiont (Extended Data Fig. 10b), with both maintaining all the motifs for ATP/ADP translocation that have been demonstrated as essential on the basis of site-specific mutations (Supplementary Fig. 1) in versions present in parasitic fungi[42]. Collectively, these results show that Comchoano, and potentially most other UBA7916, have reached a host-dependent state for sugar compounds. With the twist of potential augmentation by microbial rhodopsins, the energetics of these bacteria are akin to those of bacterial parasites of hosts in other environments[40], with the energy requirements of Comchoano being satisfied through exploitation of *B. minor*.

Comchoano have multiple other host-dependencies, with marked losses in biosynthesis of major constituents of bacterial cell walls, mirroring patterns in endosymbiotic and parasitic bacteria, such as *Mycoplasma*[6] and *Chromulinavorax*[40]. In particular,

**Fig. 2 | Phylogenetic relatedness of diverse uncultivated Gammaproteobacterial order and daily time-series dynamic with *B. minor*. a**, ML reconstruction using full-length (including Comchoano) or nearly full-length (minimum 1,200 bp) 16S rRNA gene sequences and 1,593 analysed positions under the GTR+Γ+I model demonstrates that Comchoano-1 (orange star) and Comchoano-2 (purple star) branch within an uncultivated lineage of marine Gammaproteobacteria. All assembled full-length 16S rRNA genes for a respective Comchoano were 100% identical. ML bootstrap support was generated with 1,000 rapid replicates and additional support (posterior probability) was generated via Bayesian inference. Black stars indicate 16S rRNA genes from SAG assemblies. The dashed line orange box indicates sequences that have 95% nt identity to Comchoano. **b**, Daily relative abundances of *B. minor*, Comchoano-1 and Comchoano-2 at SPOT following a phytoplankton bloom (as captured in chlorophyll data; green shading, right axis). *B. minor* and Comchoano values reflect percentages of the total rRNA gene sequence reads of all eukaryotic (18S) and prokaryotic (16S) amplicons, respectively. Lines represent temporal dynamics within the size-fractionated seawater samples, with 'protistan size fraction' (1–80 μm) and 'free-living prokaryote fraction' (0.2–1 μm) being represented by solid and dashed lines, respectively. *B. minor* was correlated with Comchoano-1 and 2 in the protistan size fraction ($r = 0.73, 0.80$; $P = 0.001, 0.0002$, respectively; Supplementary Data 1) and with Comchoano-1 and 2 in the free-living prokaryote fraction (2 d time-shifted $r = 0.62, 0.78$; $P = 0.019, 0.0011$, respectively; Supplementary Data 1).

Comchoano and several UBA7916 lack or have reduced pathways for fatty acid biosynthesis (Fig. 3d, and Extended Data Figs. 6 and 9b). Production of essential phospholipids therefore probably utilizes enzymes present for fatty acid degradation and a partial glycerophospholipid biosynthesis pathway (Extended Data Fig. 7). Biosynthesis of a major structural component of bacterial cell walls, lipopolysaccharide, is also lacking (Fig. 3d). In host-associated model bacteria, including several Gammaproteobacteria, loss of this structural barrier in outer cell membranes increases permeability for hydrophobic molecules[43,44]. Lipopolysaccharides also

stimulate host recognition and immune responses in characterized pathogenic host–microbe interactions[43] and its detection results in phagocytosis of bacteria as they invade multicellular eukaryotes[45]. By functional analogy, lipopolysaccharide loss in Comchoano indicates enhanced capacity for compound acquisition from the host and avoidance of detection and phagocytosis by the host.

Additional Comchoano auxotrophies highlight other facets of a host-dependent lifestyle. Unlike many free-living bacteria, Comchoano encode only a few proteins of B-vitamin and amino acid synthesis pathways and those that they do encode have roles in

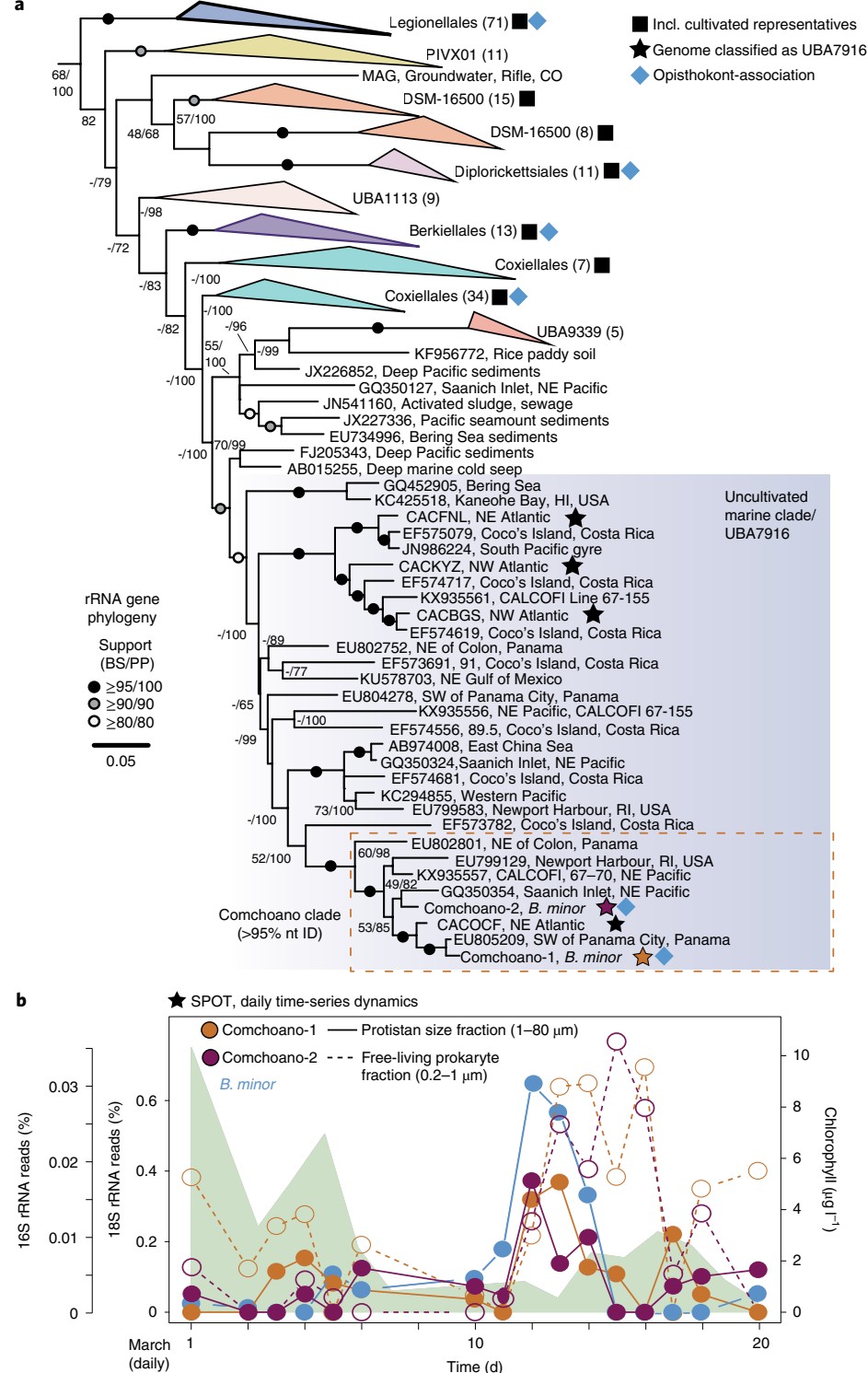

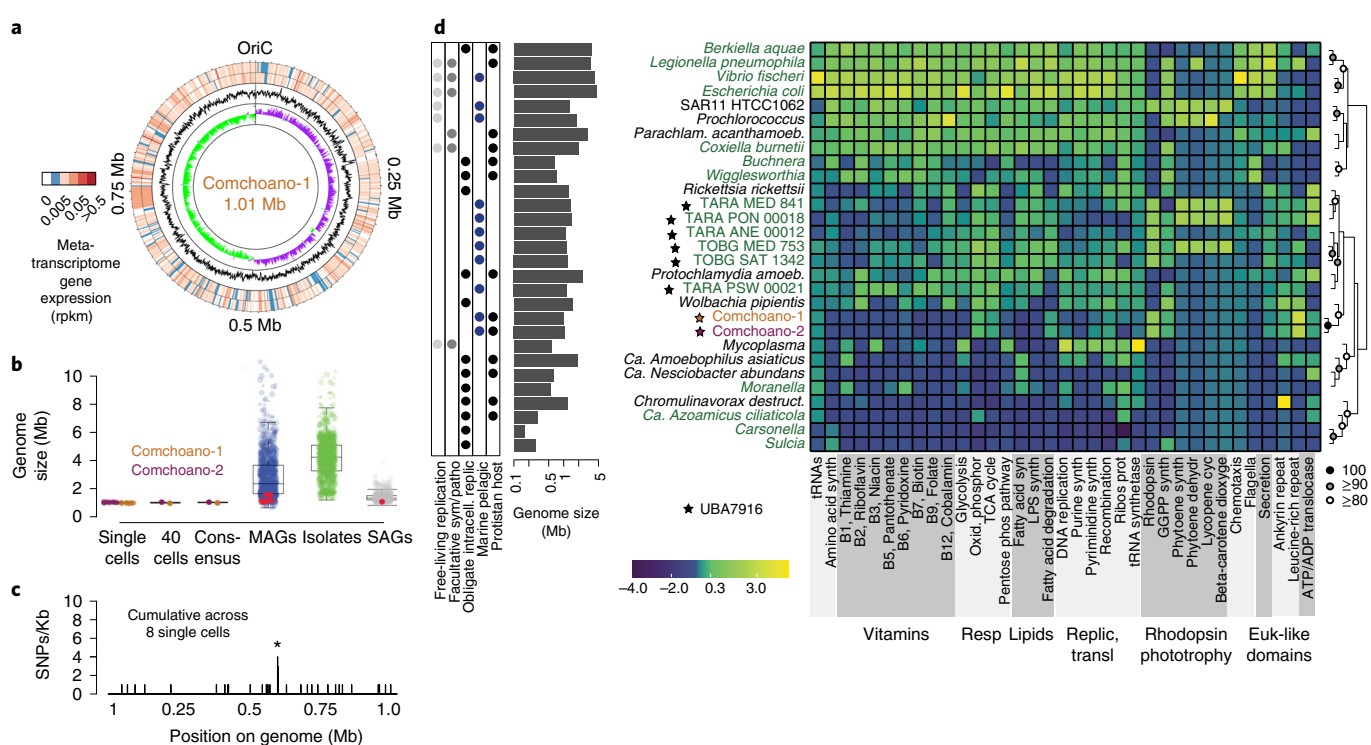

**Fig. 3 | Diminutive genomes and gaps in multiple biosynthesis pathways highlight host dependencies. a**, Genome map of Comchoano-1 genome. The innermost layer shows GC skew with axes of −0.4 to 0.4 (green negative; purple positive), demonstrating the location of the replication origin and terminus. The second layer indicates GC content (with axes of 27–54% GC content; 1,000 bp moving average), showing stability of GC content across the genome at 39% GC content. The two outermost layers indicate metatranscriptome read mapping from Stations M2 and M1 one month after M2 sampling, demonstrating Comchoano gene expression in nature. **b**, Genome size or scaled total size (as opposed to sequence assembly size) for genomes of 'marine' prokaryotes (Methods). Data for Comchoano show consistent estimated size regardless of assembly method and source data. Red circles indicate taxa we identified as belonging to UBA7916 on the basis of GTDB-tk classification (Methods, also see Extended Data Fig. 7). The boxplot represents the lower and upper quartiles, the centre line is the median and whiskers are 1.5× interquartile range. **c**, Binned and summed SNP data across Comchoano-1 from nine *B. minor* single cells, indicating low variability between cells. The total variability considering all SNPs (4.5 ± 3.1 SNPs per cell) showed that just one position was variable across more than one cell (present in a gene lacking functional annotation; indicated by asterisk). **d**, Major pathways present in Comchoano-1 and Comchoano-2, and selected opportunistic pathogens, obligate endosymbionts and free-living bacteria. Genomes indicated by black stars are MAGs (estimated as over 90% complete) belonging to UBA7916. The heat map colour scale corresponds to the number of proteins present in each pathway or function (Methods). The estimated genome size of each bacterium is shown to the left of the taxon names (logarithmic scale). Gammaproteobacteria (apart from the Comchoano) are indicated by green text. Free-living replication refers to bacteria capable of free-living growth as demonstrated by cultivation studies. Genomes are clustered by the scaled values of the metabolic pathways as shown in the heat map (Manhattan distance and complete linkage clustering), with pvclust bootstrap analysis (n = 1,000). Hierarchical clustering is for visualization purposes and may change if additional pathways are considered.

other pathways (Fig. 3d, Extended Data Fig. 7 and Supplementary Data 8). Although they are vitamin[46] and amino acid auxotrophs, there are clear mechanisms for alleviation in their gene repertoires. For example, amino acid auxotrophy is alleviated by 24 and 23 amino acid transporters encoded by Comchoano-1 and Comchoano-2 (Supplementary Data 9), respectively, demonstrating utilization of the host environment for amino acid supplies. The amino acids transported from *B. minor* remain to be established as the specificities of the transporters identified are generally not known even in cultivated bacteria such as *Coxiella*[47]. Comchoano also encode enzymes for amino acid conversions in other pathways, alongside numerous transporters for the uptake of ions (for example, nitrogen, phosphorus and iron), osmolytes and other small organics (Supplementary Data 9). Thus, the hypothesis of an intracellular host-dependent lifestyle was borne out by Comchoano's metabolic capacities and reductions therein.

**Tracing co-association along the symbiosis continuum.** Determination of where microbial associates rank along the continuum from mutualism to pathogenesis is challenging whether in

cultured systems or not, because interactions with hosts are often context dependent and can shift under different environmental scenarios, as well as being labile across evolutionary time[48–50]. Examples of microbial endosymbionts for which genomic information is available abound for heterotrophic protists belonging to other (non-Opisthokonta) eukaryotic supergroups, especially taxa residing in freshwater, soil, sediment and host-associated (for example, protists residing in termite guts) environments[33]. Fewer are known from marine habitats apart from sediments where, for example, symbionts with foraminiferans, excavates and ciliates have been reported[51–53]. Those reported from seawater appear to generally come from ciliates and diplonemids isolated from coastal environments[54] and saltwater aquaria[55]. The best-known examples of microbial symbioses in the pelagic ocean between protists and endosymbiotic bacteria generally involve eukaryotic algal species and nitrogen-fixing cyanobacteria, with the latter providing organic nitrogen to eukaryotic phytoplankton cell in exchange for carbon resources[56].

Visualization of the *Bicosta*-Comchoano relationship has not yet been achieved largely due to their uncultivated status and

ephemeral abundances in their dynamic habitat where the likelihood of re-encountering the targeted interaction on the spatial and temporal scales at which oceanographic research is conducted is low. This presents challenges and statistical limitations for, for example, efforts using hybridization chain reaction fluorescence in situ hybridization (HCR-FISH, as attempted herein; Methods) to capture co-associations that again occur either ephemerally or in relatively low abundances. These efforts can also be hampered by insufficient signal to noise/negative fluorescence ratios in field samples (which also contain naturally autofluorescent particles)[57,58], or by difficulties in interpreting signals from small uncultivated heterotrophic protists such as *B. minor*, which could also contain prey cells. Collectively, these issues render the genomic data collected in the context of unperturbed cell co-associations invaluable. With respect to choanoflagellates, and more resolved transmission electron microscopy (which is possible for abundant cells in culture), one possible endosymbiont has been noted in a cultured Baltic Sea species from brackish waters; however, the putative host–microbe interaction, functional and phylogenetic features of the putative bacterium[59] all remain unknown. Hence, in many regards, the completion and analysis of the Comchoano genomes and technology-enabled methods used herein for establishing their physical association with choanoflagellates provide the most compelling evidence yet possible for their relationship and its ramifications.

Analyses of the Comchoano genomes did not reveal an obvious 'exchange currency' that would benefit the choanoflagellate host. This contrasts with findings from cultured bacterial endosymbionts and endosymbiont consortia of many different animal and protistan hosts[6,60,61]. For example, in carpenter ants, the bacterium *Blochmannia floridanus* produces amino acids that are 'exchanged' for host-supplied carbon and nitrogen substrates. Thus, Comchoano's requirements appear to impose an energy and resource 'tax' on hosts, with no apparent return benefit. The impact of this 'tax' could range from mostly neutral for the host to strict pathogenicity under ocean conditions that are challenging for host growth and energy acquisition.

One feature of confirmed intracellular pathogens like *Legionella*, *Coxiella* and *Chromulinavorax*[32,40] is that they encode a large suite of proteins attributed to pathogen–host interactions. These types of proteins, including effectors used for host manipulation by all of the former, are absent from free-living marine bacteria such as SAR11 but present in both Comchoano (Fig. 3d and Supplementary Data 10). In Comchoano, they also include abundant eukaryote-like domains, especially ankyrin and leucine-rich repeat domains. Additionally, Comchoano-2 encodes three Sel1 repeat-containing proteins, involved in cellular trafficking during *Legionella* infection[62], alongside a transcription activator-like (TAL) effector involved in manipulating host transcription in plant pathogens and fungal endosymbionts[63]. We did not find a recently described polysaccharide lyase (EroS) shown to induce sexual reproduction in the cultivated choanoflagellate *Salpingoeca rosetta*[13]. The identified gene repertoire in Comchoano is generally present in other UBA7916 as well, often in similarly high numbers as in Comchoano (Extended Data Fig. 6 and Supplementary Data 10) and numerous protist-associated endosymbionts (Fig. 3d), as exemplified by the amoeba symbiont *Amoebophilus asiaticus*[64]. These results point to host manipulation being important across most of the UBA7916 lineage and a lifestyle involving multiple levels of host-directed interactions.

**Comchoano has a specialized type IV secretion system.** A unique aspect of Comchoano relative to the vast majority of other marine bacteria is the presence of a complete type IV secretion system (T4SS) of the subtype pT4SSi. All pT4SSi genes were expressed in metatranscriptomes constructed from the station and time point

when co-associated cells were sorted (Fig. 4a). This specific T4SS subtype is homologous to those of the pathogens *Coxiella* and *Legionella*, which secrete a broad array of effector proteins for evading host defenses and manipulating host pathways[32]. Phylogenetic analyses of a protein present across all T4SS (VirB4/IcmB/TraU) placed Comchoano in a statistically supported position within the pT4SSi clade adjacent to those for *Coxiella* (Fig. 4b,c). This placement indicates that pT4SSi is ancestral to UBA7916 and Coxiellales and was present before divergence of marine and non-marine lineages. Moreover, our analyses showed that while other secretion systems are common in marine bacteria, pT4SSi are scarce in marine bacteria (Fig. 4d). It should be noted that pT4SSi lack the relaxase used for conjugation that is typical of other T4SS, and hence cannot function in conjugation between bacterial cells[65]. Apart from Comchoano, pT4SSi were detected in five additional UBA7916 and 25 other marine bacteria (Supplementary Data 11). The majority of the latter are related to described pathogens: seven Micavibrionales (*Bdellovibrio*-like bacteria[66]), four Coxiellales, four Pseudomonadales and two Legionellales (Supplementary Data 11). The presence, synteny and phylogenetic conservation of the pT4SSi of Comchoano and distant relatives such as Coxiellales and Legionellales, imply that host association is probably an ancient trait for these lineages broadly. However, the pT4SSi and other features (for example, eukaryotic-like domains, ATP/ADP translocase) that are conserved among Comchoano and close relatives, paired with variations in the extent of reduction in genome size and metabolic capacities, suggest that the obligate nature of association is a more recent and sporadic trait.

## Discussion

Our findings to this point call for the naming of what has thus far been an enigmatic order of uncultivated bacteria with little known about its ocean roles or lifestyle. We propose the following status: *Candidatus* Comchoanobacterales ord. nov., *Candidatus* Comchoanobacteraceae fam. nov. for what has been known so far as the UBA7916 order and family, respectively, following the protocols of order and family naming after type species (see below and protologue in Supplementary Information). This status is proposed due to Comchoano and related UBA7916 members being phylogenetically distinct from other Gammaproteobacteria orders (Fig. 2a and Extended Data Fig. 5a,b) and their distance based on RED and amino acid identity (AAI) metrics. For the type species, that is, Comchoano-1, we propose the status *Candidatus* Comchoanobacter bicosticola gen. et sp. nov. and for Comchoano-2, *Candidatus* Synchoanobacter obligatus gen. et sp. nov.

Understanding the drivers of microbial dynamics and cell–cell interactions is essential to elucidating marine elemental and energetic cycles[1,21]. Nevertheless, little is known about direct microbe–microbe associations in the oceans and their ecological implications, especially for small heterotrophic protists. Advancements have been hindered by the paucity of methods for preserving intact relationships between cells collected in pelagic marine ecosystems. Here we report the discovery of a microbiome in Pacific Ocean single-celled choanoflagellates enabled by methodologies for circumventing both community changes that occur during cultivation efforts and disruption of associations caused by most field sampling approaches. The implications stemming from the phylogenetic, population and genomic features of the microbiome-dominant Comchoano have still unappreciated ramifications. Recently, obligate bacterial associates of heterotrophic protists have been increasingly noted and their genomes have been sequenced from freshwater and terrestrial habitats[33]. However, the vast majority of bacteria sequenced or cultured so far from the ocean maintain pathways for energy conversions and biosynthesis of essential compounds important to survival as free-living cells in the often resource-depleted marine environment[29,30]. In the case of Comchoano, its membrane modi-

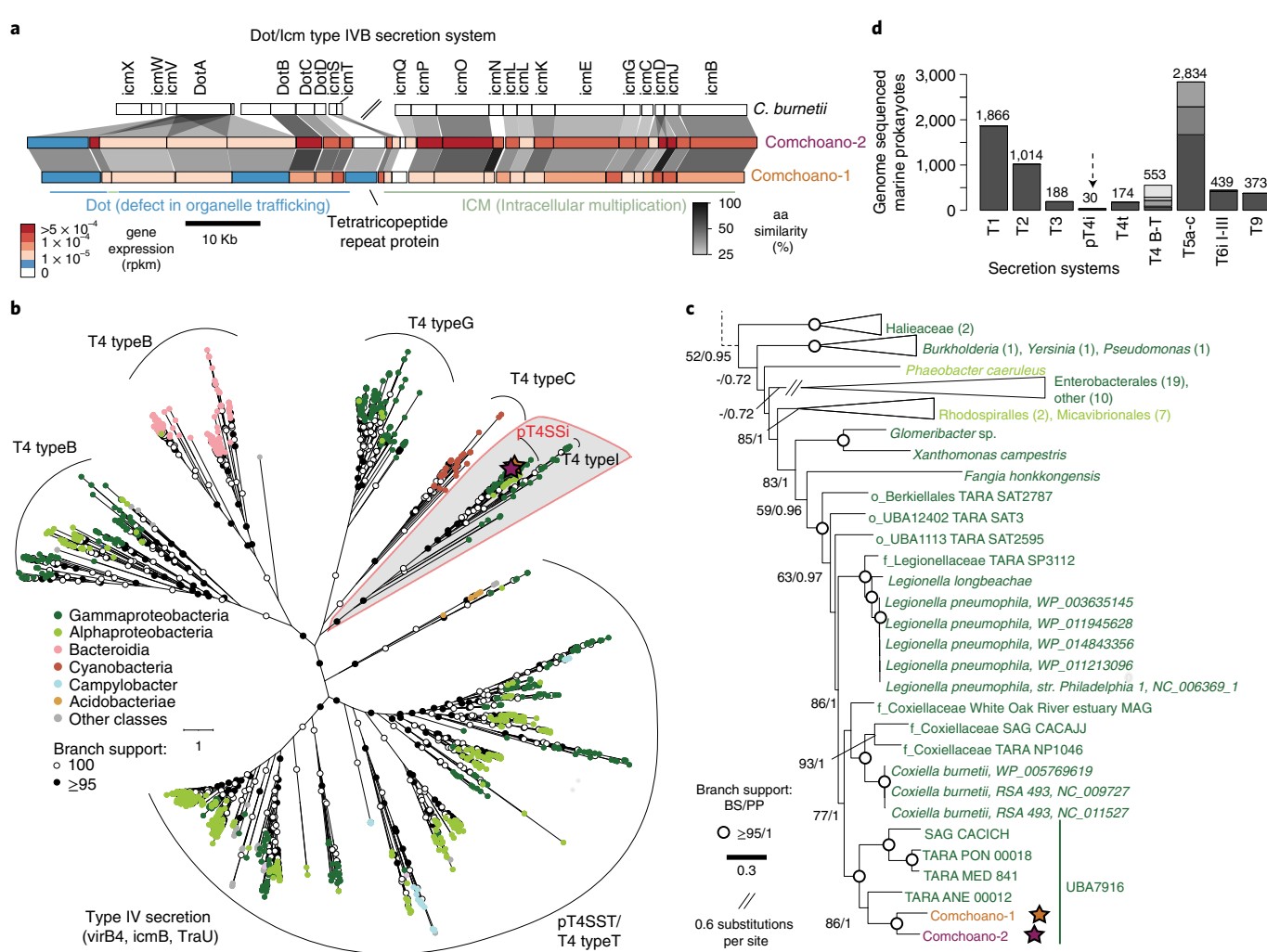

**Fig. 4 | Comchoano's T4SS is rare in free-living marine bacteria and phylogenetically closest to those of confirmed pathogens. a**, T4SS proteins present in *C. burnetii*, Comchoano-1 and Comchoano-2 (represented to scale on the genome scaffold, that is, nucleotides). Synteny is indicated (grey shading), along with percent amino acid similarity between homologous proteins (darkness level of shading). Colour fills indicate metatranscriptome read mapping (reads per kb million, rpkm) from Pacific Ocean station M2 (all pT4SSi genes were expressed). The break mark shown in the *C. burnetii* genomic segment represents ~5,800 bp. **b**, ML phylogenetic reconstruction of 1,629 T4SS VirB/IcmB/TraU proteins with tips coloured by bacterial class. **c**, Rooted subtree ML phylogeny of 72 pT4SSi, T4 type I and related sequences from a supported clade extracted from **b** representing an analysis of homologous positions. For the subtree, some clades were collapsed for display purposes and node support is indicated by open circles (≥95% ML; 1 posterior probability, Bayesian) or numerical percentages (if ≥50% ML or ≥0.9 posterior probability, Bayesian). **d**, Prevalence of secretion systems across genomes of 18,671 marine prokaryotic isolates, SAGs and MAGs, regardless of genome completion level (Methods). Arrow, the pT4SSi possessed by Comchoano and other UBA7916 members, as well as 25 Proteobacteria from other lineages (as defined by the presence of the mandatory VirB/IcmB/TraU and greater than six component genes; Methods), the majority being classified as Coxiellales or Legionellales and relatives of host-associated bacteria in terrestrial ecosystems. Shading within bars reflects distribution of each subtype of the respective secretion system type, where applicable (for example, the T5a–c bar has three shaded segments representing T5a, T5b and T5c).

fications and molecular mechanisms for avoiding host detection and manipulating hosts, alongside limited metabolic capacities, requirement for numerous substrates and direct energy exchange, implicate a 'resource drain' that could have detrimental impacts for the host, depending on environmental conditions. The findings also provide a possible mechanism for concentrated delivery of high levels of bacterial compounds directly to the eukaryotic host, without dilution caused by release in seawater, such as the compound levels required to trigger sex and multicellularity in choanoflagellates[11–13]. The widespread distribution of both the bacterivorous host and bacterial symbiont discovered herein, as well as the diversity of potentially host-associated uncultivated bacteria related to Comchoano, call for intensified efforts to identify cryptic symbioses and deeper

knowledge of the strength and directionality of their influence on resource flow in the ocean.

## Methods

**Sampling and cell sorting.** For cell sorting, seawater was collected on 20 March 2014 at Monterey Bay Time Series station M2 (36.688° N; 122.386° W, 56 km from shore, Fig. 1a) using Niskin bottles mounted on a conductivity, temperature, density instrument (CTD) bearing rosette. As reported in an earlier paper describing the sample preparations for the choanoflagellate sorting experiment[14], water from 20 m depth was pre-filtered with a 30 μm nylon mesh (to remove protists of a size likely to clog the flow cytometer) and concentrated by gravity over a 47 mm diameter, 0.8 μm pore size Supor (Pall) filter to a theoretical concentration of 250×. The latter step concentrates protists while lowering the relative numbers of free-living bacteria in the sample; it is not essential to remove all free-living

bacteria as they will be selectively removed by single-cell sorting of the protists where only co-associated bacteria will be present. The concentrated seawater was labelled with LysoTracker Green DND-26 (Invitrogen), a label that targets acidic food vacuoles of living cells[67], to a final concentration of 25 nM from a working stock of 10 µM diluted in artificial seawater. The labelled sample was analysed on a BD Influx flow cytometer equipped with a 488 nm laser and running on sterile nuclease-free 1x PBS as sheath fluid.

The sorted population was discriminated on the basis of positive LysoTracker signal (that is, fluorescence detected in the 520/35 nm bandpass filter under 488 nm excitation) as compared to an unlabelled sample, as well as absence of chlorophyll-*a* autofluorescence (that is, fluorescence detected in the 692/40 nm bandpass filter) and comparable forward angle light scatter (a proxy for cell size) to select for a coherent population of heterotrophic eukaryotes. Listmodes were analysed using Winlist (version 7.0, Verity Software House). Single cells from the population with these characteristics were sorted in a 384-well plate using the single-cell sorting mode from the BD FACS Sortware (software v1.0.0.650), ensuring that only one cell would be sorted in each well. A subset of wells was left empty or received 20 cells for negative and positive controls, respectively. The plate was illuminated with ultraviolet light for 2 min before performing the sort, and covered with foil and frozen at −80 °C immediately after its completion.

**Sorted-cell sequencing and analysis.** For single-cell sorts, sorted cells were subjected to alkaline lysis at room temperature, followed by whole-genome amplification by MDA with the RepliG single-cell kit (Qiagen) or WGA-X workflow (Supplementary Data 2) in 2 µl reactions set up with an Echo acoustic liquid handler (Labcyte). For single-cell samples, partial 16S rRNA gene sequences were amplified using V4 primers 515F-Y (GTGYCAGCMGCCGCGGTAA) and 806R (GGACTACNVGGGTWTCTAAT)[68] and partial 18S rRNA gene sequences were amplified using V4 primers TAReuk454FWD1 (CCAGCASCYGCGGTAATTCC) and TAReukREV3 (ACTTTCGTTCTTGATYRA)[69]. 18S rRNA gene amplicons were initially processed as reported when the 18S data were originally published[14] using UCLUST[70] to cluster sequences into 99% OTUs (Fig. 1b, left panel). Subsequently, to potentially further resolve OTUs to ASVs using state-of-the-art methods, all amplicon sequences (16S and 18S) were processed within QIIME 2[71] by trimming primers with cutadapt (v1.13)[72], and then quality filtering and denoising with DADA2[73], generating amplicon sequence variants (ASVs). During the denoising step, forward and reverse 16S rRNA gene sequences were trimmed to 210 and 180 bp, respectively; 18S rRNA gene sequences were trimmed to 250 and 200 bp, respectively. Separate MiSeq runs were denoised independently and combined after denoising, as required for generation of accurate error profiles. Both 16S and 18S ASVs were classified in QIIME 2 with classify-consensus-blast with–p-perc-identity 0.85 and–p-maxaccepts 1 using the SILVA 132 99% clustered representative as a reference database[74] and majority_taxonomy_7_levels as the taxonomy file. The dominant 18S rRNA amplicon sequence was affirmed to have 100% similarity across 378 bp to the choanoflagellate *B. minor*, which was hand-picked and sequenced from Danish marine surface waters but remains uncultured[16]. For ASV processing, 16S rRNA ASVs classified as eukaryotic or chloroplasts were removed. The remaining ASVs classified as bacterial and archaeal reads were selected and then those that were classified as mitochondria went through a second curation step. The mitochondrial sequences were searched via blastn[75] against the NCBI nucleotide database, and after excluding the hits to uncultured taxa, those that had best hits to mitochondria were removed from further processing.

After denoising and classification, cells that had fewer than 5,000 reads were removed from further processing. The remaining single-cell samples had an average number (±s.d.) of reads of 710,947 ± 464,742 and 159,492 ± 134,364 for the 16S and 18S rRNA gene sequences, respectively. The environmental samples had 93,352 ± 23,463 and 163,737 ± 16,046 reads for the 16S and 18S rRNA amplicon sequences, respectively. For the single-cell analyses, due to the biases introduced from MDA, ASV relative abundance data were converted into presence and absence data, where read proportions >5% and >25% for the 16S and 18S rRNA gene datasets, respectively, were considered 'present'. These conservative values were chosen to reduce the likelihood of 'cross-talk' (for example, see ref. [76]) between samples influencing co-occurrence frequencies.

**Genome sequencing, assembly, binning, curation and assessment.** Metagenomic sequence libraries were prepared from MDA products as described above with either the Qiagen repliG enzyme or WGA-X (Supplementary Data 2). Metagenomic sequencing was performed with paired-end 2x 150 or 2x 250 Illumina sequencing for population sorts and individual cells (Supplementary Data 2). For each sample, the sequences were assembled with spades (v3.11)[77], with the –sc setting and kmers of 21, 33, 55, 77, 99 and 127. Contigs longer than 1 kb were then binned on the basis of their tetranucleotide coverage and GC (Guanine-Cytosine) content with anvi'o v6.2[78] (Supplementary Fig. 2). Genomes were recovered from 18 single cells and two population sorts (20 cells each) (Supplementary Data 2). From both individual single cells and population sorts, the Comchoano genomes were often highly complete (up to 96.6%, average ± s.d. 92.0 ± 8.0%) (Supplementary Data 2), as estimated with CheckM

taxonomy_wf workflow for domain Bacteria[26]. Assemblies from only one *B. minor* cell with Comchoano rendered a second full-length 16S rRNA gene sequence, a Flavobacterium whose genome was otherwise not well-recovered (estimated <5% complete, Supplementary Data 2).

Preliminary analysis showed the individual genomes to be highly similar within a given type (identical 18S rRNA gene sequences and 99.9 ± 0.08% whole-genome nucleotide identity, FastANI[79]), so we sought to improve the genome assemblies by a combination of automated and manual curation steps. Contigs from each Comchoano individual genome (that is, from individual *B. minor* cells or the combined 20 cell population sorts) were pooled and re-assembled in Geneious Prime (version 2020.2.4), using the Geneious assembler with the following settings: maximum percent gap of 1% per contig, maximum gap size of 50 bp, overlap of 50 bp with a percent identity of 99%, and total mismatch number of 1%. Before this step, three contigs from Comchoano-2 were excluded due to long (>5 kb) repeat sequences not observed in the other contigs. For Comchoano-1, this approach produced a single contig similar in size (1.01 Mb) to the average individual genome size of the individual genome assemblies. Other (smaller) contigs produced by the assembler were observed to be highly similar via progressiveMauve[80] with the large contig with minimal differences probably due to sequencing and assembly artefacts. For these reasons, we proceeded with this single contig for final genome curation and polishing (see below). For Comchoano-2, the Geneious assembler produced multiple contigs of ~696 kb and ~370 kb. Similar to Comchoano-1, these two sets of contigs within these two size ranges were shown to be highly similar via progressiveMauve, so we proceeded with the longest contigs (696,580 and 370,342 bp) for additional genome polishing.

For final genome curation steps, we mapped all the original contigs from single cells and population sorts to their respective Comchoano genome with Minimap2 for highly similar sequences (-k19 -w19 -A1 -B19 -O39,81 -E3,1 -s200 -z200 -N50–min-occ-floor=100)[81]. From these mapped contigs, the consensus was predicted on the basis of nucleotides that were greater than 50%, with coverage greater than two. Subsequent whole-genome alignment demonstrated that the ends of the contigs of each Comchoano tended to be enriched in repeats. In the case of the single Comchoano-1 contig, inspection of the alignment revealed a stretch of nearly identical sequences on the 5' end and near the end of the 3' end (98.6% of nucleotides across 212 bp), which was followed by a highly similar repeat region on the 3' end of the contig. This 3' end was represented by only a single original contig, hence was probably an artefact. Thus, the 3' end of the Comchoano-1 genome was trimmed to remove this redundancy starting at the highly similar overlap. To further curate on the basis of the original paired Illumina reads, all reads were mapped to the single Comchoano-1 contig and two Comchoano-2 contigs with Bowtie 2[82], after which up to 125,000 reads from each sample were examined to confirm consistent coverage and further polish the consensus sequences. From these mapped reads, a new consensus was determined on the basis of 50% majority of mapped reads (the original consensus and mapped-reads curated contigs were greater than 99.99% similar for both Comchoano-1 and Comchoano-2). At this point, in the case of Comchoano-1, reads overhanging each end were identical, enabling circularization (further validated by read mapping and circularization, see below). For each consensus Comchoano genome, the origin of replication was predicted with OriFinder (on the basis of GC skew and DNA replication binding motifs)[83]. Comchoano-1 was subsequently re-oriented, with the first position being the origin of replication and binding the two ends together; because Comchoano-2 was not a single contig, the contigs were re-oriented to be in the proper orientation according to GC skew, but not arranged to start at the origin of replication. For Comchoano-1, the reads from each sample were then re-mapped with Bowtie 2 and up to 200,000 paired reads from each sample were again examined (consensus taken again, resulting in 82 bp changes across the genome) with 100% coverage.

Upon re-mapping of the original contigs from individual single cells with nucmer (default settings, except -b 5000) to the Comchoano-1 working consensus, SNP analysis (predicted with show-snps -ClHTr) showed three instances in which the consensus departed from the majority of single-cell assemblies. Thus, the original contigs were remapped with Minimap2 in Geneious and the majority bases (over 50%) were taken.

After this, SNPs between the consensus Comchoano-1 genome and the assembled Comchoano-1 from single *B. minor* cells were again predicted with nucmer (default settings, except -b 5000)[84] and then SNPs were again predicted with show-snps -ClHTr. Nine positions had more than one cumulative SNP across all cells. All but one of these SNPs occurred across one of two pairs of highly similar paralogous sequences, and as such were probably the result of challenges to assembly of these highly similar sequences. These SNPs were manually inspected via read mapping and visualization in Geneious, which revealed that 1 bp in the consensus was probably incorrect and was thus corrected in the Comchoano-1 consensus genome, resulting in the final consensus Comchoano-1 genome.

To identify single nucleotide polymorphisms between the Comchoano consensus genomes and Comchoano from *B. minor* single cells, reads were mapped with Bowtie 2 default settings from eight *B. minor* cells containing Comchoano sequences. Read pairs that mapped discordantly or more than once were excluded. Coverage was then calculated with samtools mpileup[85] and polymorphic sites predicted with bcftools call–ploidy 1 -P 1.1e-10 -v -m[86]. Resulting SNPs were

further filtered with bcftools filter to exclude SNPs within 3 bases of indels, quality of less than 30 and coverage of 10 or less. Additionally, SNPs were examined between the original contigs from the single cells and population sorts to the consensus genomes with nucmer and show-snps, as described above. The end (2,555 bp) of one 167 kb contig from a single cell was filtered from the final output (Supplementary Data 2) because a region with greater than 10% divergence was observed over this region, which was suspected to be due to assembly error.

**Genome annotation.** Comchoano-1 and Comchoano-2, and other genomes used in comparative analyses were annotated as follows (note, the same methods were used for annotating the genomes compared to Comchoano in Fig. 3d). tRNAs were predicted with the Aragorn pipeline (v1.2.38)[87]. Proteins were predicted by Prodigal (v2.6.3)[88]. Protein annotations were determined using EggNOG emapper.py (version 2.0.1–14)[89,90], with the diamond blastp search option and diamond database downloaded on 19 Mar 2019. Additionally, protein domains were annotated via hmmscan[91] using the pfam database as a reference[92]; ankyrin repeats and leucine-rich repeats identified with hmmscan of pfam were checked using blastp (e-value < 1×10⁻¹⁰). Amino acid auxotrophy and pathway calculations were predicted by annotation on Kbase[93] web server by first predicting proteins via RAST[94] and then applying the fba_tools Predict Genome Auxotrophies tool (v1.7.6)[93]. Metabolic pathway maps (Extended Data Fig. 7) were created using Pathway tools (v22.0)[95]. Putative signal peptides were identified with the Phobius web server (phobius.sbc.su.se)[96]. Rhodopsins and retinal biosynthesis proteins were identified from hmmscan of pfam (e-value < 1×10⁻¹⁰) as follows: rhodopsin (PF01036.19), GGPP synthase (PF00348), phytoene synthase (PF00494.18), lycopene cyclase (PF05834) and beta-carotene 15,15'-dioxygenase (PF15461.5). Phytoene dehydrogenase was identified by blastp (e-value < 1×10⁻¹⁰) search of the conserved protein domain family TIGR02734: crtI_fam. Transport proteins were additionally annotated via web-based transporter annotation tool TransAAP[97] and amino acid transporters were identified by blastp against a dataset of predicted amino acid transporters in *Coxiella burnetii* RSA493[98]. ATP/ADP translocases were identified via hmmscan using the TLC ATP/ADP transporter pfam (PF03219) (e-value < 1×10⁻²⁵). The *B. minor* genome was searched for proteins involved in glycolysis using the EggNOG diamond search blastx search option[89,90].

Initial secretion system analysis prediction (Fig. 3d) was performed via txsscan[65] (galaxy webserver version), excluding type IV pili and flagellar genes. Subsequently, MacSyFinder (version 1.0.5) was used to identify the secretion systems in Comchoano-1 and Comchoano-2, as well as genomes from the same dataset of genomes as used for genome size comparison (n = 18,671, but excluding any genomes <50% complete)[65], with default settings and the ordered replicon (due to the assembled contiguous nature of the genes, even the genomes were in multiple contigs). The '–min-mandatory-genes-required' parameter was set to three for T4SS_typeI, the maximum number for this system (due to some overlap with pT4SSi, in particular for Comchoano-1 and Comchoano-2), but left at default for all other systems. Specifically, Comchoano pT4SSi was identified through the presence of virB4/icmB and ≥6 accessory proteins (each Comchoano has nine), alongside the absence of the conjugation-related relaxase typical of other T4SS subtypes. For pT4SSi comparison with *C. burnetii*, the nomenclature is based on the original publication of the *C. burnetii* genome[99]. T4SS proteins not localized to the genomic regions shown in Fig. 4a are IcmX and IcmW in Comchoano-1, and IcmN and IcmE in Comchoano-2.

Metatranscriptomic reads were mapped to the Comchoano genome assemblies via bbmap.sh[100](v37.17) at a sequence similarity cut-off of 0.99. Mapped reads were parsed via HTSeq-count[101].

**Genome size comparison.** To compare Comchoano-1 and Comchoano-2 genome size and completion to a wide variety of marine bacterial and archaeal genome sizes, 4,931 marine bacterial and archaeal genomes were surveyed from the JGI/IMG database that were identified as ocean, coastal, pelagic or neritic[102], 12,714 marine single-cell genome sequences[27], 894 MAGs from *Tara* Oceans[103] and 4 archaeal MAGs from a recent analysis of the BioGEOTRACES dataset[104]. The genome completion for all genomes was estimated with CheckM taxonomy_wf workflow at the domain level for bacteria and archaea[26]. Genome sizes were then estimated by accounting for the CheckM-derived completion and contamination metrics from each genome. Only those genomes that were estimated to be >80% complete on the basis of SCG estimates and having <5% contamination (3,652 total) were used in the size comparisons.

**Phylogeny and classification.** Initially, to classify bacteria on the basis of their whole genome, Comchoano-1 and Comchoano-2, and the other genomes described above were classified via the GTDB-tk (v1.4.0) classify_wf command[19] which extracts 120 putatively vertically transferred genes, aligns them and then places them on a reference tree with pplacer; this analysis assigned the Comchoano to the order UBA7916, of which the only representatives with genomes are uncultivated oceanic SAGs and MAGs. This command was also used to calculate the RED between Comchoano and sequences already in the GTDB (default settings), as well as to recalculate with newly added sequences (Comchoano plus UBA7916 from other datasets, described below, not found in release 95 of GTDB) with the '–recalculate_red' flag. In each case, the RED values for each

Comchoano were 0.83. To determine the average AAI between Comchoano-1 and Comchoano-2, the aai.rb programme from enveomics[105] was used, which resulted in the two-way AAI value of 49.0 ± 14.8% based on 729 homologous proteins. This AAI similarity is roughly similar to the amount expected from class-level differences[106], yet the differences between the 16S rRNA gene sequences are 95%, which is similar to genus-level differences. With this low similarity, the average nucleotide identity may be unreliable;[107] OrthoANIu[108] suggests an average nucleotide identity of 67.03% on the basis of 159,905 bp on average, which is about 15% of the genome size. To circumvent such a problem of rapid evolutionary divergence in, for example, symbionts, Parks et al.[19] recommend the RED metric that we used, which takes this into account. Subsequently, we also performed phylogenomic reconstruction of the Comchoano and other Gammaproteobacteria, and 38 single-copy ribosomal proteins from Proteobacteria were extracted with GToTree[109] for Comchoano-1 and Comchoano-2, all representative Gammaproteobacterial genomes in the GTDB database (n = 5,784), all identified Gammaproteobacteria marine single-cell genome sequences (n = 1,413)[27], six UBA7916 genomes from a *Tara* Oceans MAG study[103] and one UBA7916 MAG from JGI/IMG not found in the other datasets (IMG Genome ID 2721755926). Additionally, ten Alphaproteobacterial genomes were used as outgroup sequences. Ribosomal proteins were identified via hmmscan with the intrinsic pfam gathering threshold used as cut-offs for each respective protein. Any ribosomal proteins that were detected more than once in a genome or were less than or longer than 30% of the median sequence for a given protein were excluded. Genomes were removed from the analysis if they encoded <30% of the 38 proteins. The ribosomal proteins were aligned with MAFFT[110] and positions with greater than 50% gaps were removed via trimAl[111]. The sequences were then concatenated; the total alignment included 6,744 genomes and 4,502 amino acid positions. Phylogenetic reconstruction was performed with FastTree[112], with the -notop setting selected. Phylogeny was visualized in iTOL[113] with the GTDB order-level taxonomy used as branch colours[19].

To more finely resolve the phylogenomics of Comchoano, on the basis of the FastTree reconstruction produced above, the genomes from the 15 closest Gammaproteobacterial orders to UBA7916 (Berkiellales, Coxiellales, Diplorickettsiales, DSM-16500, Legionellales, Piscirickettsiales, PIVX01, PIWD01, UBA1113, UBA12402, UBA6002, UBA6186, UBA7366, UBA7916 and UBA9339), plus Francisellales as the outgroup, were selected. Notably, these GTDB orders, besides Francisellales, correspond to the Legionellales order of the NCBI taxonomy. GToTree was then re-run for these orders. Genomes with fewer than five of the ribosomal proteins were removed. This selection removed three genomes, one of which (CACFNL) was included in the 16S rRNA tree, as described below. Then, the same alignment and trimming parameters as above were applied, resulting in 187 taxa and 4,573 amino acid positions. Maximum likelihood (ML) phylogenetic reconstruction was then performed using the LG+Γ+I model (the best model for 36 out of the 38 genes using the corrected Akaike information criterion (AICc) scores generated by IQ-TREE[114] for amino acid substitution and 500 rapid bootstraps.

For 16S rRNA gene phylogenetic analysis, we gathered sequences in three ways: (1) rRNA genes were extracted from Comchoano, the genome-sequenced relatives and the same genomes from the 16 orders used in the phylogenomic subtree analysis with barrnap default settings[115]. (2) These sequences were then searched by blastn against a SILVA138 database that had been curated to remove sequences that had less than 1,200 bp and/or any degeneracies. Three sequences with best hits to mitochondria were removed from analysis on the basis of this search. From this search, the five closest hits (based on bit-score) to each remaining genome-derived rRNA gene sequence were selected (excluding the Francisellales which were the outgroup). (3) To more broadly sample the rRNA sequence diversity from the 15 GTDB orders of the SILVA database, five sequences were selected (at random) from the five individual groups that were included among the best scoring hits to any of the genome-derived rRNA sequences (Coxiellales, Diplorickettsiales, EC3, Berkiella and Legionellales). Sequences from genome-sequenced taxa that had degeneracies or were less than 1,200 bp were then removed, and the remaining sequences combined with sequences from steps 1 and 2, and clustered with cd-hit-est[116] at 99% sequence identity, resulting in 240 sequences. The resultant sequences were then aligned with MAFFT and trimmed to remove positions with greater than 95% gaps with trimAl[111]. Phylogenetic reconstruction was performed in RAxML[117] with the GTR+Γ+I model and 1,000 rapid bootstraps. Additionally, a Bayesian inference (BI) phylogenetic reconstruction analysis was performed with MrBayes[118], with two independent runs of 2,500,000 generations with four chains each (that is, one cold and three heated), sampling every 250 generations and printing every 1,000 generations. After a burn-in of the first 25% of trees, posterior probabilities for node supports were calculated on the basis of assessment of convergence among runs using the R package RWTY[119].

For the type IV secretion phylogeny, putative VirB4 ATPases (also known as virB4, icmB or TraU) from the following sources were included: (1) VirB4 from the same dataset of genomes as used for genome size comparison (n = 18,671), (2) VirB4 from the SecReT4 database (n = 570)[120] and (3) VirB4 from an additional study of secretion systems across public databases (n = 562)[65]. Using MacSyfinder as described above, this analysis identified 1,629 unique virB4 of putative homologous proteins from T4SSs. For phylogenetic reconstruction, a tree was

constructed by the approximate ML approach with FastTree[112] using the complete dataset aligned with MAFFT[110] and masking positions having ≥5% gaps. Then, a subset of bacterial sequences that grouped together in a well-supported clade (100%) with sequences identified as pT4SSi or T4SS type I was selected. On this subset, ML and BI phylogenetic analyses were then performed. First, the selected reference sequences were re-aligned with MAFFT using default parameters. The alignment was masked by removing positions having ≥5% gaps. The best-fit model of amino acid evolution was determined on the 956 amino acid positions with ProtTest 3.2[121] as being LG+Γ4+I, using the AICc. In RAxML[117], a tree search was performed with 1,000 nonparametric bootstrap replications using the same evolution model. The BI analyses were conducted in MrBayes 3.2.6[118], with two independent runs of 2,500,000 generations with four chains each (that is, one cold and three heated), sampling and printing every 100 generations. After a burn-in of the first 10% of trees, posterior probabilities for node supports were calculated on the basis of assessment of convergence among runs using RWTY. Taxonomy for genomes in virB4 phylogeny was determined by the GTDB-tk classify_wf command[19] from data release 89.

For rhodopsin phylogeny, representatives of diverse rhodopsins were initially collected from a previous study[14]. Additionally, to broadly survey for rhodopsins similar to those in Comchoano and proteorhodopsins in general, rhodopsin proteins were extracted from global marine metagenome surveys[103,122,123], marine single-cell surveys[27] and predicted proteins from metagenomic assemblies from the North Pacific[124] via hmmscan of the Bac_rhodopsin protein, with a gathering threshold of greater than 26. These sequences plus amplicons from the Red Sea[125] and the MicRhoDE[126] database were then searched by blastp[75] against the Comchoano-1 and Comchoano-2 rhodopsins. The sequences with bit-score greater than 250 were then added to the sequences from the previous study[14], as well as all rhodopsins from MAGs and SAGS classified as UBA7916. This resulted in 480 sequences that were then aligned using MAFFT default settings[110]. The alignment was trimmed to remove positions with greater than 50% gaps via trimAl[111], resulting in an alignment of 250 positions. ML reconstruction was performed in RAxML[117] with 1,000 rapid bootstraps and the LG+Γ+F substitution model as in ref.[14].

For ATP/ADP translocase phylogeny, we leveraged two datasets from recent publications for collecting representative sequences[35,41], in addition to collecting putative ATP/ADP translocases from single-cell prokaryotic genomes from the North Atlantic[27], MAGs from *Tara*[103] and genomes from GTDB[19]. For the latter three datasets, ATP/ADP translocases were predicted by searching against the PFAM[92] database with hmmscan[91] using a gathering cut-off of 20.6. Combining these datasets resulted in 1,695 sequences. These sequences were then clustered at 0.95 amino acid similarity via cd-hit[116] and subsequently filtered to remove sequences shorter than 250 amino acids via seqkit, resulting in 1,379 sequences. The sequences were then aligned with MAFFT and ambiguous positions were trimmed with trimal via the automated heuristic on the basis of similarity statistics ('-automated1'). This resulted in an alignment of positions. Phylogenetic reconstruction was then performed with IQ-TREE[114] with extended model selection (-m MFP) and 1,000 ultrafast bootstraps. Subsequently, the region of the tree containing UBA7916, Comchoano and numerous characterized proteins from parasites and symbionts was extracted (as indicated in Extended Data Fig. 10a), re-aligned and trimmed as performed for the full dataset, resulting in an alignment of 413 sequences and 412 positions. Phylogenetic reconstruction was again performed as above. All trees were visualized in iTOL[113]. To examine important motifs to nucleotide transport, we selected the same subset as in Graf et al. 2020, re-aligned with MAFFT and visualized the alignment in the ESPript web server (https://espript.ibcp.fr)[127].

**Environmental distributions.** For 16S and 18S rRNA gene distributions from Monterey Bay, seawater was collected in 2014 (20 March, 2 April and 5 May) via CTD rosette from the top 1 m at three locations (M2: as above; M1: 36.762° N, 122.038° W; C1: 36.797° N,121.847° W) and 500 ml was filtered through 47 mm diameter, 0.2-μm-pore-size Supor (Pall) filters. Additionally, seawater from the top 5 m was collected via the ship intake at M1 and M2 stations, pre-filtered through a 30 μm nylon mesh (except for the March samples where no pre-filtration occurred) and 20–30 l were sequentially size fractionated through 142 mm diameter, 3 μm Versapor and 0.22 μm Supor filters. One additional sample, from 30 April 2015, was collected from 5 m depth with a CTD rosette, 500 ml were collected on 47 mm diameter, 0.2 μm Supor filters and size fractionated on 2 μm polycarbonate and 0.2 μm Supor (Pall) filters. DNA was extracted with a DNeasy kit, with modifications in an earlier report[128]. In the case of the 142 mm filters from the 2014 size-fractionated samples, only 1/6 of the filter was used for DNA extraction. 16S and 18S rRNA gene V4 amplicons were amplified and processed as described above. Additionally, V4-V5 amplicons were amplified and processed the same way as the V4 amplicons, except for the use of the primers 515F-YA (GTGYCAGCMGCCGCGGTAA) and 926R (CCGYCAATTYMTTTRAGTTT).

For 16S and 18S rRNA gene distributions from Malaspina cruises, data were downloaded from the NCBI via BioProject PRJEB25224 and PRJEB23913. This project analysed prokaryotic (primers, 515F-Y, 926R[129]) and pico-eukaryotic composition (TAReuk454FWD1 (CCAGCASCYGCGGTAATTCC) and TAReukREV3 (ACTTTCGTTCTTGATYRA)[69]) from surface seawater during the

circumnavigating Malaspina 2010 cruise. The data were imported into QIIME 2 where primers were removed from the 18S data (16S primers had already been removed), and 16S and 18S denoised independently via dada2 denoise-paired (16S,–p-trunc-len-f 210–p-trunc-len-r 180; 18S,–p-trunc-len-f 210–p-trunc-len-r 200). Taxonomy was then assigned using QIIME 2 with the same settings as above. To examine possible 16S rRNA gene ASVs with affiliation to UBA7916, ASVs classified as 'Coxiellales' were extracted ($n = 226$; note, in SILVA, 'Coxiellales' is the classification for Comchoano and UBA7916 because UBA7916 is not defined in that rRNA database) for phylogenetic placement with epa-ng[130] with default settings, except for the use of the –no-heur setting, using the alignment and maximum likelihood 16S rRNA gene phylogenetic reconstruction previously described. The best placement was used to determine the affiliation of a given ASV within UBA7916 ($n = 167$).

For 18S rRNA from *Tara* Oceans, data were downloaded as a published OTU dataset amplified with V9 rRNA primers 1389F (TTGTACACACCGCCC) -3' and 1510R (CCTTCYGCAGGTTCACCTAC), universal for eukaryotes[10,131]. To make the taxonomy consistent with the Malaspina 18S data, the *Tara* Oceans V9 OTU representative sequences were reclassified with QIIME 2 as described above.

For the San Pedro Ocean Time-series daily time-series, the published OTU datasets and representative sequence dataset were downloaded from FigShare[132]. As described elsewhere[132], these samples originated from the top 1 m of seawater at 33.55° N, 118.4° W. Seawater was pre-filtered (80 μm), and then cells were sequentially collected on 1 μm AE (Pall) glass filter and 0.2 μm Durapore (Millipore) filter. 16S and 18S rRNA genes (V4-V5 amplicons) were amplified with the 515F (GTGCCAGCMGCCGCGGTAA) and 926R (CCGYCAATTYMTTTRAGTTT) primers, as reported in the paper originally publishing the data[132]. The sequences were clustered at 99% similarity threshold and representative sequences were chosen on the basis of the most abundant sequence within a cluster. Statistical correlations between Comchoano-1, Comchoano-2 and *B. minor* were searched for the initial daily time-series portion of this study (12 March–1 April 2011) via eLSA[133], allowing for a maximum time-delay of 5 d and for the full time-series, allowing three time-point time-delay, but only those with no time delay (Spearman correlation) were used in Extended Data Fig. 1b,c. In both cases, $P$ and $q$-value determination was performed using the 'theoretical' option, not permutation, and the calculations were performed only on taxa that were detected on 33% of sampling points.

**HCR-FISH probes.** To design probes for Comchoano-1, Comchoano-2 and *B. minor* for HCR-FISH[134], 23 probe candidates targeting 16S rRNA for Comchoano-1 and Comchoano-2, and 17 probe candidates targeting 18S rRNA genes for *B. minor* were provided by Molecular Technologies. Subsequently, these sequences were searched against NCBI to be specific only to their target (Comchoano-1, Comchoano-2 or *B. minor*, respectively), as well as across available sequences from closely related organisms (for Comchoano-1 and Comchoano-2 that included searches against each other). Two top candidates were identified for each target on the basis of their sequence specificity. For Comchoano-1: probe '2', tcgggaaaagtgatggcgagtggcggacgggtgagtaatgcgtaggaatcta and probe '5', tgccgatgaaggctttcgggtcgtaaagcactttcagttgggaagatggctta; for Comchoano-2: probe '2', tcggaagaaatgatggcgagtggcgaacgggtgagtaatgcgtaggaatcta and probe '14', ctttagtaataaagggtgccttcgggaaccgagatacaggtgttgcatggc; for *B. minor*: probe '5', tgattcttcgagtcttcctctcgtagttgtttggcgcacttgattgggtgcc and probe 'v4-1', tctgattcgaaagatcggtccgccgcaaggcgagcactgattcttcgagtct. For each probe set, these sequences were converted into 'even' and 'odd' split-initiator probes in accordance with the HCR v3.0 protocol[134]. Because the three cell types are currently not in culture, we used a Clone-FISH approach[135] in *Escherichia coli* for positive and negative control testing of probes. Ultimately, both the evaluated *B. minor* and Comchoano-1 probes were deemed to be specific in a limited set of cross-reactivity tests, meaning fluorescent signal was not appreciable when the *B. minor* probe was paired with another choanoflagellate sequence and fluorescence was also not appreciable when Comchoano-1 probes were paired with Comchoano-2 sequences (clones). However, only one (probe '14') of the Comchoano-2 probes was specific, as Comchoano-2 probe '2' also amplified in Comchoano-1 clones. Comchoano-2 probe '14' can be used individually to target only Comchoano-2.

**Reporting summary.** Further information on research design is available in the Nature Research Reporting Summary linked to this article.

## Data availability

Single and multi-cell sort raw data (short-read archives, SRA) are available via NCBI Project Number PRJNA640955, which includes V4 16S rRNA gene amplicon sequences from single cells, whole-genome shotgun sequences from single cells, MBTS 16S and 18S V4 rRNA gene amplicons, and MBTS V4-V5 rRNA gene amplicons (see Supplementary Data 12 for individual list of SRA accessions). 18S V4 rRNA gene amplicons are available as part of Needham et al.[14]. Comchoano-1 and Comchoano-2 whole-genome sequences are available via accessions CP092900 and JAKUDN000000000 and their full-length 16S rRNA gene sequences are deposited as OM801198 and OM801197. Alignments, tree files and processed amplicon data are available via FigShare (https://doi.org/10.6084/m9.figshare.c.5850662). As cited, the publicly available databases used in the

manuscript include PFAM (http://ftp.ebi.ac.uk/pub/databases/Pfam/), SILVA 132 (https://www.arb-silva.de/documentation/release-132/) and 138 (https://www.arb-silva.de/documentation/release-138/), eggnog (http://eggnog5.embl.de/download/eggnog_5.0/), diamond (http://github.com/bbuchfink/diamond/releases/download/v2.0.14/diamond-linux64.tar.gz), MicRhode (http://application.sb-roscoff.fr/micrhode/download), GTDB (https://data.gtdb.ecogenomic.org/releases/release95/), Malaspina (PRJEB23913 and PRJEB25224), *Tara* Oceans (PRJEB7988), SecReT4 (https://bioinfo-mml.sjtu.edu.cn/SecReT4/download.html), JGI/IMG (https://img.jgi.doe.gov/) and daily time-series of SPOT data (https://doi.org/10.6084/m9.figshare.2069403.v2) and are publicly available.

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

## Acknowledgements

We thank captains and crews of the RV *Western Flyer* and RV *Rachel Carson*, as well as Worden Lab members who assisted on these expeditions. This work was supported by the Gordon and Betty Moore Foundation (GBMF 3788 to A.Z.W.; GBMF 3307 to P.J.K., A.E.S and A.Z.W. as well as T.A. Richards), MBARI and GEOMAR. Work conducted by the US D.O.E. J.G.I. was supported under Contract No. DE-AC02-05CH1123. We thank K. Turk and J. Zehr for microscope access, and T. A. Richards for constructive criticism on the initial manuscript and A.Z.W. thanks the Harvard University Radcliffe Institute for Advanced Study.

## Author contributions

C.P., S.W., P.J.K., A.E.S. and A.Z.W. designed the research; D.M.N., C.P., S.W., C.C.M.Y., A.J.L., M.M., L.S., R.R.M and A.Z.W. performed the research; D.M.N., C.P., E.E.G., C.B. and C.C.M.Y. analysed the data; D.M.N. and A.Z.W. wrote the paper, with input from P.J.K. and A.E.S., and edits from all authors.

## Funding

## Competing interests

The authors declare no competing interests.

## Additional information

**Extended data** is available for this paper at https://doi.org/10.1038/s41564-022-01174-0.

**Correspondence and requests for materials** should be addressed to David M. Needham or Alexandra Z. Worden.

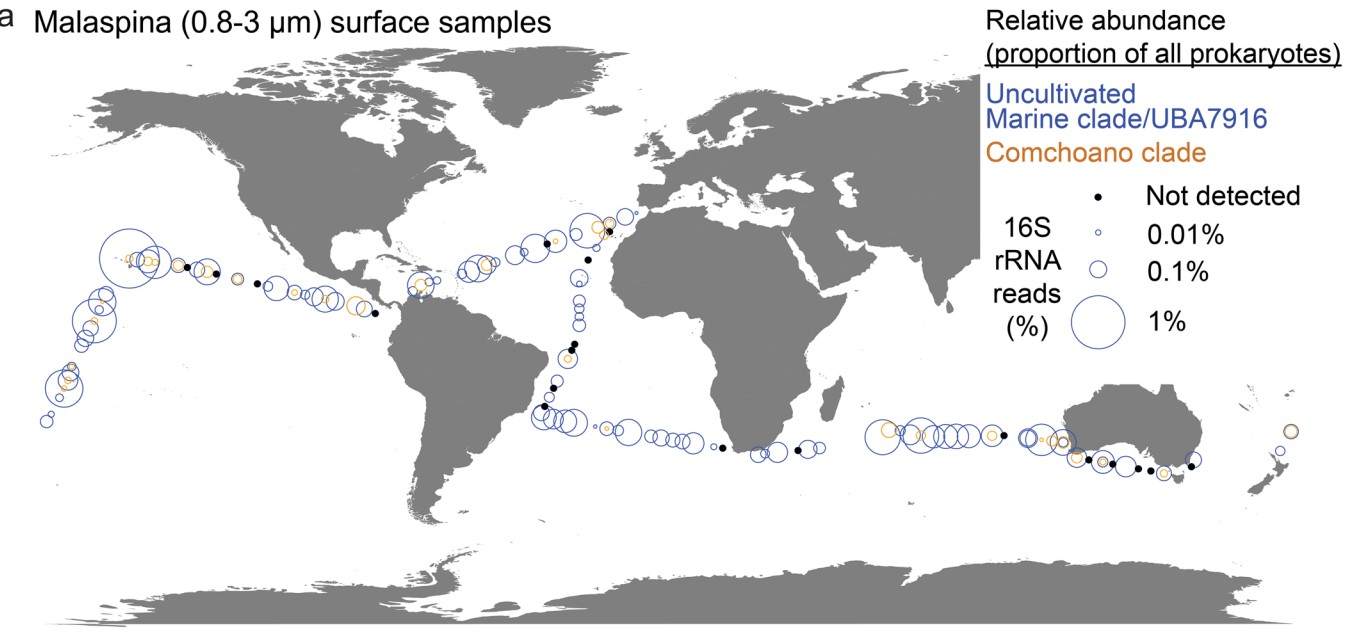

**a** Malaspina (0.8-3 μm) surface samples

Relative abundance
(proportion of all prokaryotes)

Uncultivated
Marine clade/UBA7916
Comchoano clade

16S
rRNA
reads
(%)

- Not detected
- 0.01%
- 0.1%
- 1%

**b** San Pedro Ocean Time-series,
Protistan size fraction (1-80 μm)

Spearman Correlation > 0.35
p < 0.05; q < 0.05

27087 Chrysochromulina
5355 Dino-Group-III 1
39972 Stephanoecidae H
25614 Stephanoecidae D
1188 Stephanoecidae E
19407 Haptolina brevifila
11505 MAST-1B
14092 Stephanoecidae D
17338 Novel-clade-2
29824 Telonemia Group-2
8378 Chrysochromulina
17654 Telonemia Group-2
23141 Micromonas Clade-A

0.37
0.40
0.40
0.45
0.46
0.39
0.41
0.46
0.59
0.42
0.48
0.39
0.37

Comchoano-1

43861 Phaeocystales
4266 Prymnesiophyceae
31812 MAST-3E
1875 Dinophyta
35930 Haptophyta
9023 Dinophyceae 61
4381 Pelagophyceae
19285 Dino-Group-II-Clade-10-and-11
21292 Dino-Group-III
24161 Bicosta minor

0.39
0.51
0.41
0.52
0.43
0.52
0.46
0.50
0.38
0.43
0.50
0.38
0.37
0.46
0.41
0.43

Comchoano-2

**c**

Choanoflagellida, Telonemia, Cercozoa, Haptophyta, Chlorophyta, Stramenopiles, Dinophyta, Ochrophyta

OTUs from lineage correlated to Comchoano-1 (%)

OTUs from lineage correlated to Comchoano-2 (%)

**Extended Data Fig. 1 | See next page for caption.**

**Extended Data Fig. 1 | Oceanic distributions of UBA7916 and Comchoano shows wide distribution of Comchoano and co-occurrence of Comchoano with eukaryotes in the SPOT daily-to-monthly time-series showing enrichment for choanoflagellates.** Relative abundance of phylogenetically placed 16S rRNA gene ASVs of the broader Uncultivated Marine Clade/UBA7916 (blue shading in Fig. 2a) and the Comchoano clade (dashed orange box in Fig. 2a) showed frequent presence and low relative frequencies in Malaspina 2010 surface data (3 m, 0.2–3.0 µm size fraction). 'Not detected' is only explicitly indicated for UBA7916, since its absence necessarily implies no detection of the Comchoano clade. **b**, Correlation network of Spearman Correlation values (p < 0.05; q < 0.05, Spearman rho > 0.35; computed via eLSA[133]) for Comchoano (from the 1–80 µm, Protistan size fraction) and eukaryotes (also from the Protistan size fraction) from the SPOT time-series where each node represents the eukaryotic OTUs or Comchoano. Colours correspond to the Class level classification via PR2 database as described in Needham et al., 2016. Unique numbers for each node correspond to the denovo OTU identification number (based on 99% sequence clusters). Lines connecting nodes represent positive correlations between the respective nodes, with only correlations between Comchoano and eukaryotes shown. The thickness of the line indicates the strength of the correlation, also indicated by the values printed on each edge. P-values were calculated based on a theoretical approximation (not permutations) to better scale with the large amount of input data and as explicitly developed and described for this application;[10,131] exact p-values (two-tailed) and q-values are available in Supplementary Dataset 1. **c**, Using the pairwise correlations summarized in b, along with taxonomic information from the correlation table input, proportion of each lineage (class level) that had correlations to Comchoano are shown. There are a variety of important considerations in interpreting correlative network analyses. For example, shared preferences for environmental conditions or indirect effects may also lead to positive correlations between Comchoano and eukaryotic OTUs (for example similar environmental preferences between host and not host eukaryotes)[136]. Additionally, false negatives or false positives may occur due to the compositional aspect of the data (relative abundances scaling to 100%; that is, not absolute abundances)[137,138], as well as imprecision in quantification of the sequence counts (for example, especially in regards to lesser abundant taxa where stochastic variation is higher).

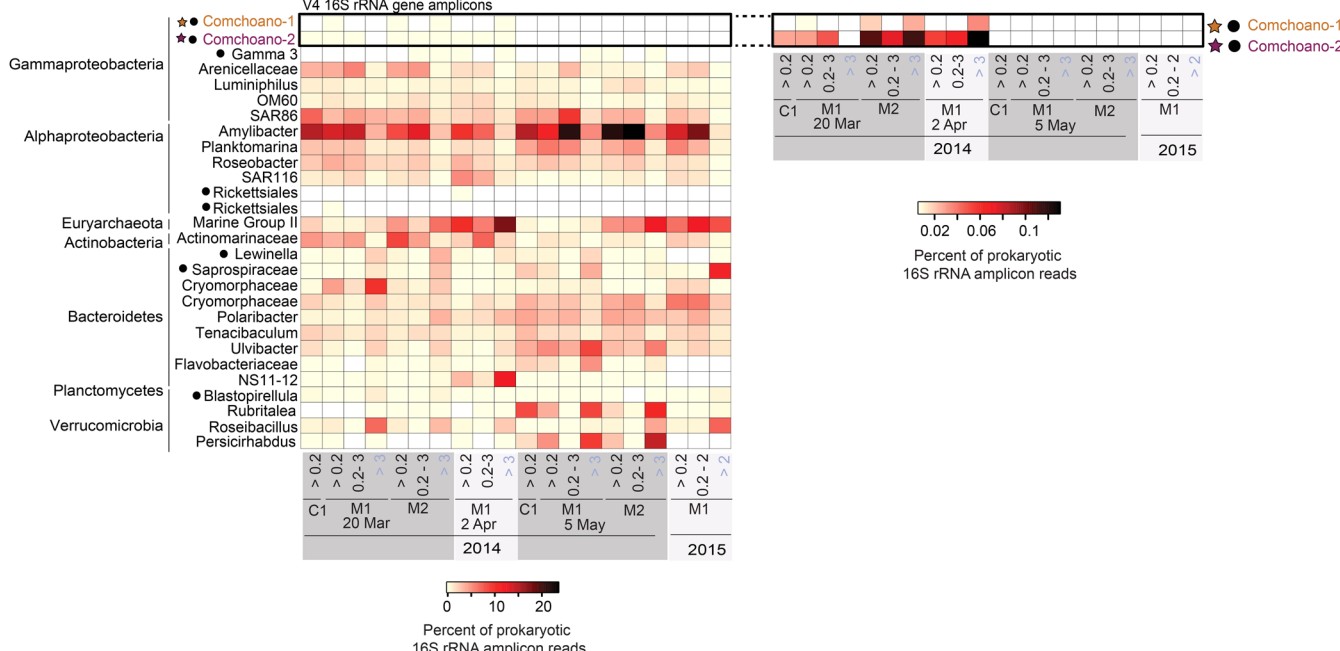

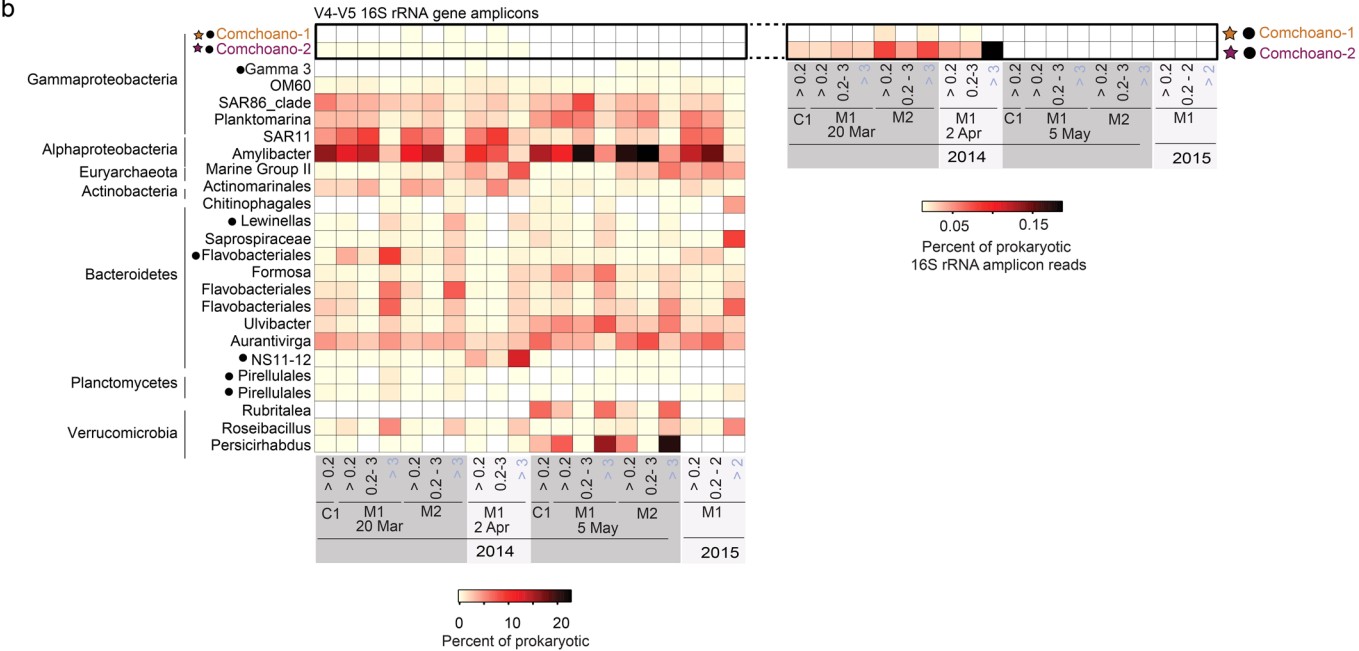

**Extended Data Fig. 2 | Prevalence of Comchoano in Monterey Bay Time-series samples. a**, Relative abundances of prokaryotic taxa based on V4 16S rRNA gene amplicons at Stations M1, M2, and C1 (located ~halfway between station M1 and the coast) (Fig. 1a) averaging >2% across all samples or >5% in any given sample, and those associated with more than one *B. minor* cell (indicated by a circle). Adjacent heatmap (right side) re-scale Comchoano plots due to comparatively low relative abundances. **b**, Same as **a**, except for V4-V5 rRNA gene amplicons. Squares of the heatmap that are white are samples for which no sequences were detected for a given ASV.

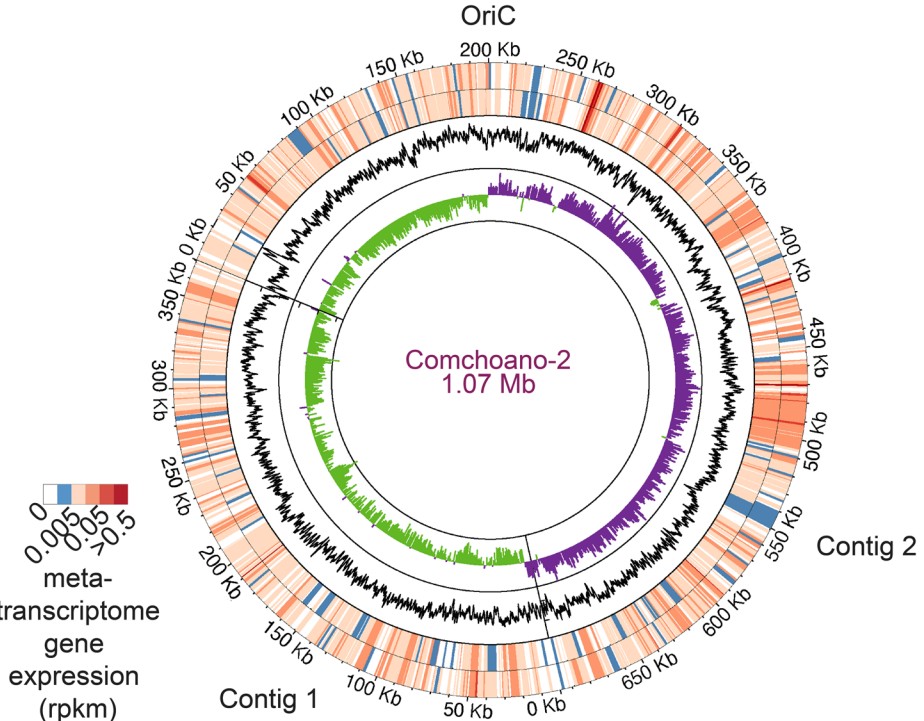

**Extended Data Fig. 3 | Genome map of the Comchoano-2 genome.** The layers are as in Fig. 3a for Comchoano-1. The innermost layer is GC-skew (with axes of -0.4 to 0.4), demonstrating the location of the replication origin and terminus. The second layer indicates GC content (with axes of 27–54% GC; 1000 bp moving average), showing stability of GC content across the genome at 42% G+C. The two outermost layers indicate metatranscriptome mapping from stations M2 and from M1 one month after M2 sampling, demonstrating Comchoano-2 gene expression.

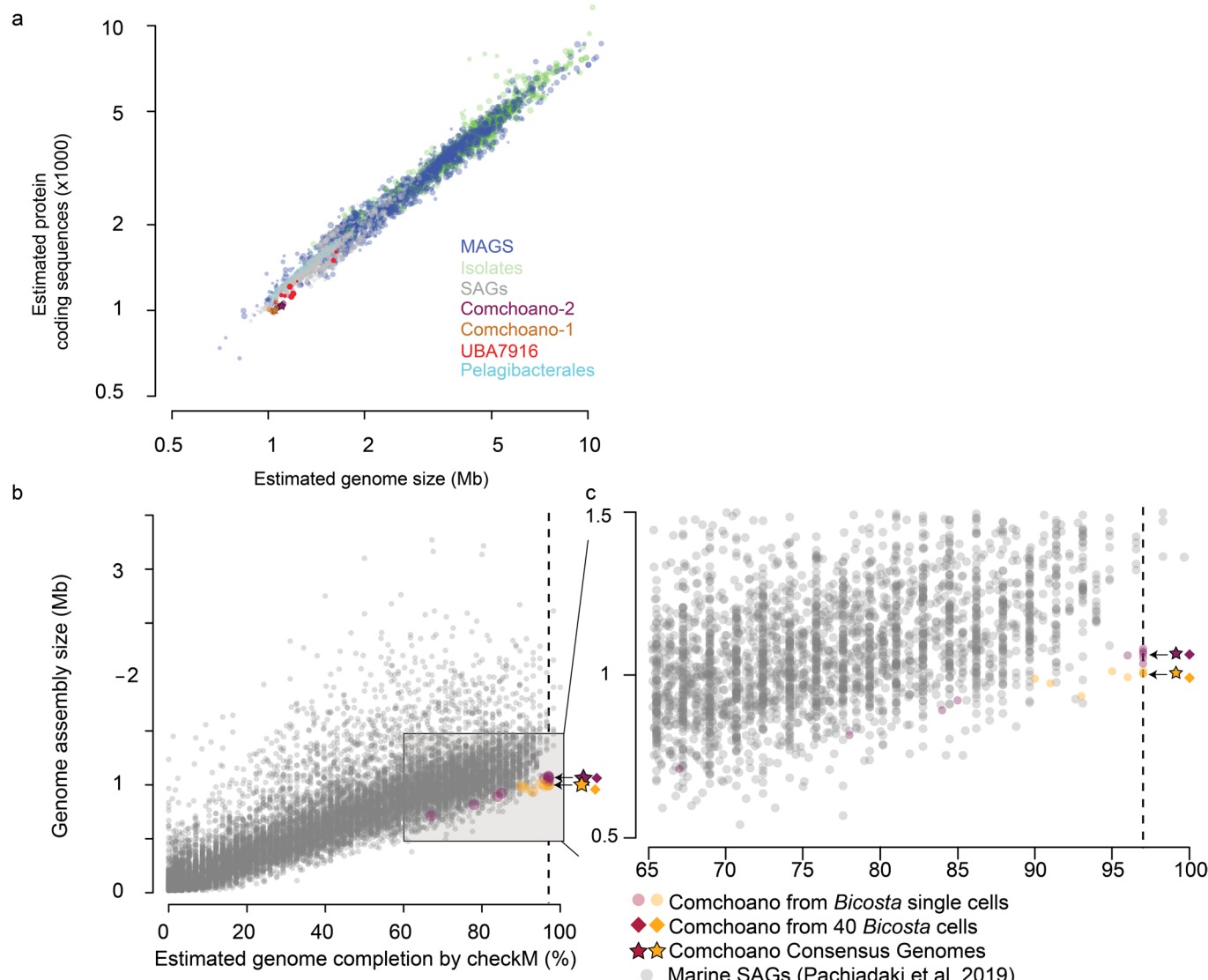

**Extended Data Fig. 4 | Estimated genome sizes and estimated number of protein coding genes across the same 3,652 prokaryotes (same as in Fig. 3b) and comparison of Comchoano genome completion and size (bp) compared to a recent large dataset of marine SAGs.** Colour fills are as indicated in the figure key; genomes from the order Pelagibacterales (SAR11) are outlined in cyan. Estimated genome sizes and number of coding sequences are calculated taking into account the genome completion and redundancy estimates (see methods). As discussed in the text, this probably results in an overestimate in the size and number for Comchoano of genes, due to their estimates of completion likely being underestimates for a variety of reasons as described in the text. Regardless, the completion estimates were used here for consistency to the rest of the dataset. Comchoano-1 and Comchoano-2 share 731 putative orthologs within their 951 and 1,004 predicted proteins. The estimated number of genes and genome sizes for SAR11 genomes (>80% complete and <5% contamination, n=365) are about 40% larger at 1,421 ± 153 genes and 1,327,689 ±146,940 bp, respectively. **b**, Shapes correspond to the different assembly workflows for each individual genome, where the circles are assembled from individual cells (that is, *B. minor* in the case of Comchoano), diamonds are from populations of 40 *B. minor* cells, stars are the consensus genomes for Comchoano. Consensus and population genomes for Comchoano are shifted to the right on the x-axis; their completion values are all 96.6%. The dashed line at 96.6% indicates the CheckM estimated percent completion of Comchoano (likely underestimate, see text). Average CheckM estimated completion of Comchoano-1 and Comchoano-2 from individual single *B. minor* cells are 94.3 ± 2.5 n=8, 89.3 ± 10.0 n=10, respectively vs 38.3 ± 28.0 n=12,714 from the dataset of marine SAGs[27]. This indicates recovery is much higher than would be expected if Comchoano were associated with *B. minor* as single cells, thus it is likely multiple Comchoano were associated with each *B. minor* cell. Assembly statistics for individual, population, and consensus Comchoano are available in Supplementary Data 2.

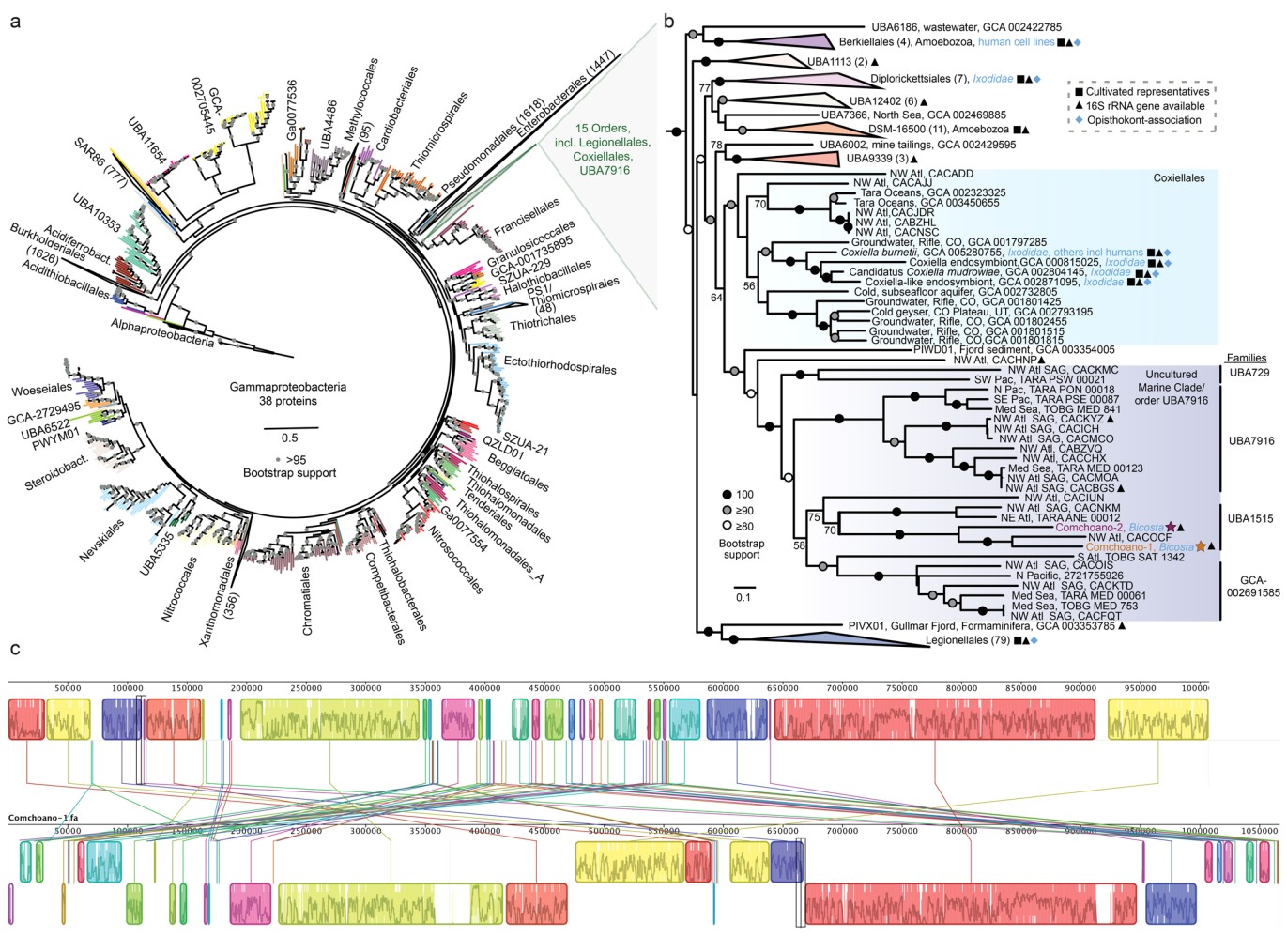

**Extended Data Fig. 5 | Phylogenomic reconstruction of Comchoano. a**, ML phylogenetic reconstruction of Gammaproteobacteria via FastTree based on a concatenated analysis of 38 proteins, with ultrafast bootstrap support (4,502 analyzed amino acid positions). Branch colors represent different orders of Gammaproteobacteria. **b**, ML reconstruction of a subset of UBA7916 and the 15 Gammaproteobacterial orders closest to Comchoano based on **a**, plus Francisellales as outgroup (removed for visualization purposes), under the LG model of substitution (4,571 analyzed amino acid positions) via RAxML. Bootstrap values represent the percentage of 500 rapid replicates. These analyses demonstrate that the Comchoano are members of the uncultivated and diverse bacterial order, UBA7916 (i.e., the novel Candidatus Comchoanobacterales ord. nov. described herein). Labels with "NW Atl SAG'' are from a study from a planktonic single cell genomics study from the Northwest Atlantic[27]. **c**, Synteny plot of Comchoano-1 and Comchoano-2 computed with Mauve[138] default settings.

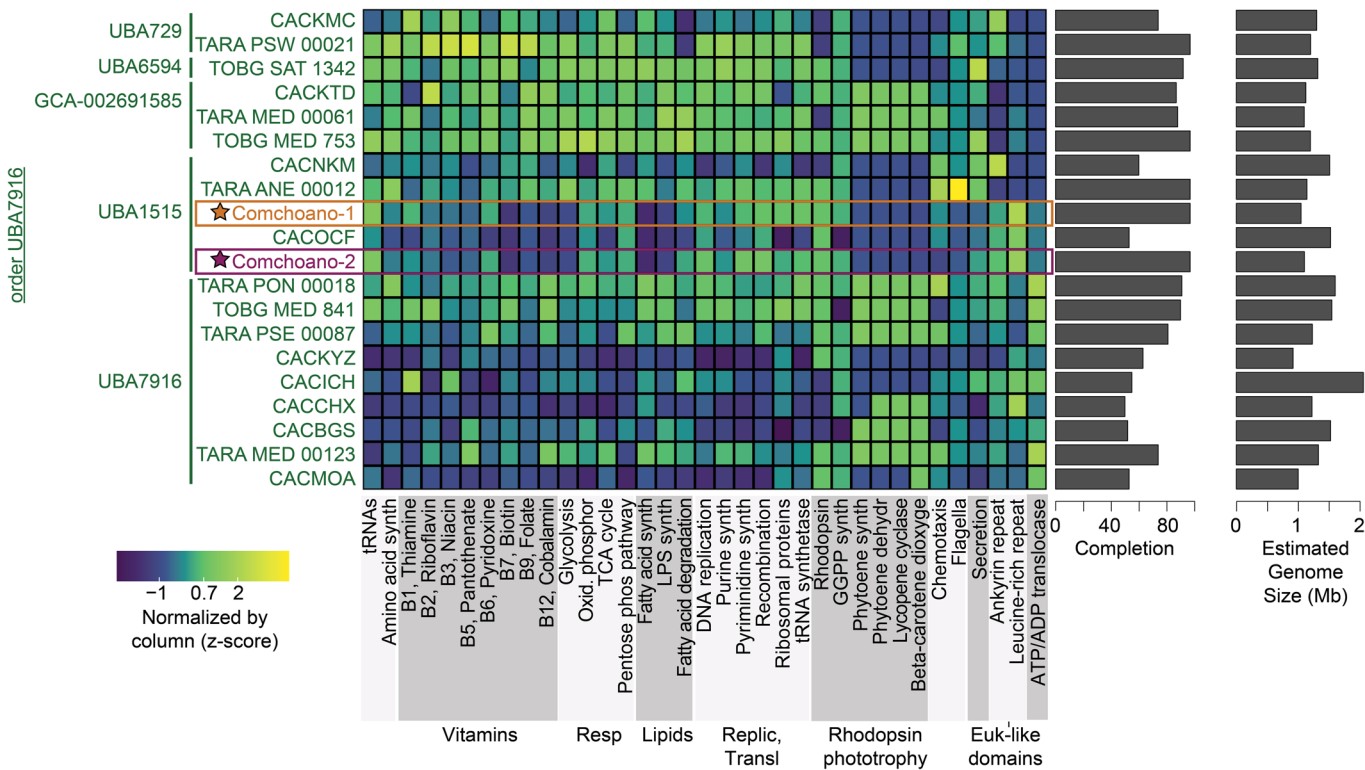

**Extended Data Fig. 6 | Comparative genome and functional analysis of Comchoano-1 and Comchoano-2 with other UBA7916 greater than 50% complete.** All UBA7916, besides Comchoano-1 and Comchoano-2 are metagenome-assembled genomes from the Tara Oceans. As in Fig. 3d, the heatmap color scale corresponds to the number of genes present in each pathway or function as described in the methods and abbreviations are also as in Fig. 3d. The estimated genome completion and estimated genome size of each genome is shown at right.

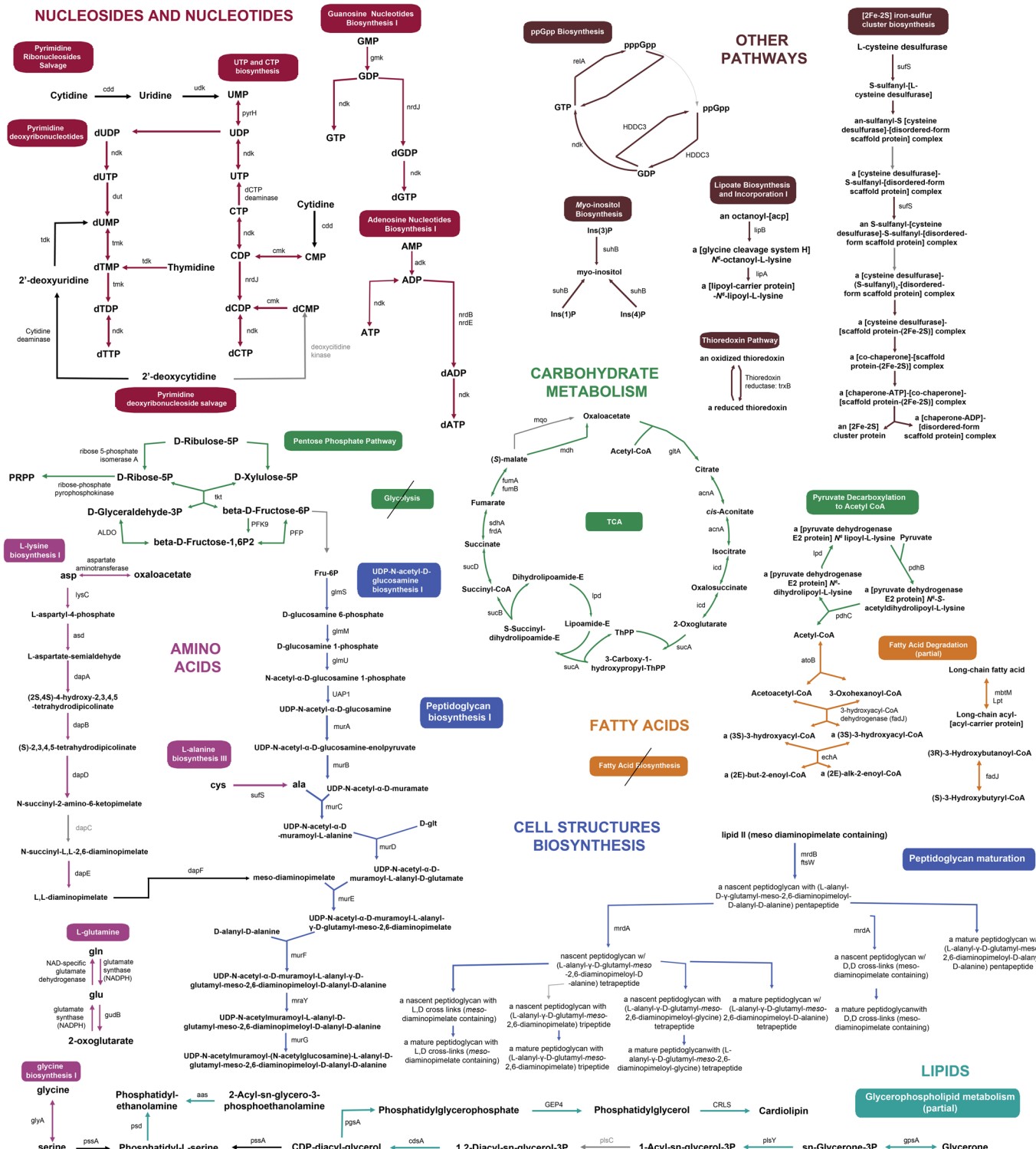

**Extended Data Fig. 7 | Metabolic pathways encoded by Comchoano-1 and Comchoano-2 consolidated from Pathway Tools.** Gray arrows indicate the enzyme is missing in both genomes; black arrows indicate enzymes only found in Comchoano-2. Otherwise, colored arrows indicate the presence of the enzyme in both genomes. Pathways for amino acids, as shown here, are very reduced, with those genes that are encoded in amino acid conversions, also part of other pathways, for example, those in peptidoglycan biosynthesis and the partial lysine pathway that converts aspartic acid into meso-diaminopimelate, an intermediate in peptidoglycan biosynthesis. With absence of biosynthesis of fatty acids, including, Comchoano may meet these requirements via fatty acid degradation and via a fused lipo-phospholipid transporter and acyltransferase (mbtM/aas), and the partial glycerophospholipid biosynthesis pathway.

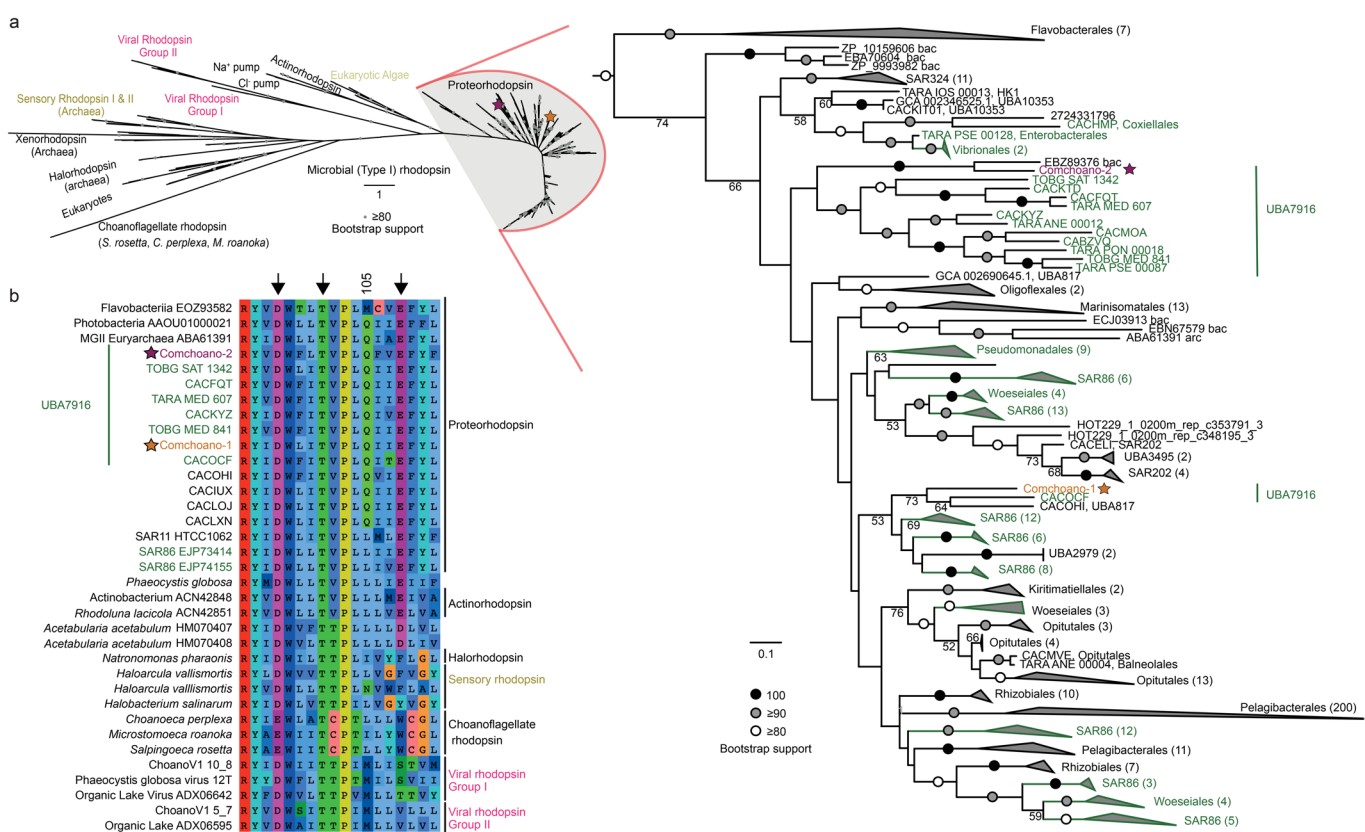

**Extended Data Fig. 8 | Phylogenetic reconstruction and functional characteristics of Comchoano-1 and Comchoano-2 rhodopsin proteins.** These photoactive membrane proteins are common to marine bacteria[36], and were identified in both Comchoano and 17 of the 20 UBA7916 members (Extended Data Fig. 8). **a**, Unrooted ML phylogeny of bacterial, archaeal, eukaryotic and viral rhodopsins, including Comchoano-1 and Comchoano-2, using 250 homologous positions and 1,000 bootstrap replicates, including Comchoano-1 and Comchoano-2. Rhodopsin proteins were gathered from public databases, and metagenomic assembled genomes and contigs from global metagenomic surveys, and oceanic single cell surveys (see Methods). **b**, Pruned tree from the Proteorhodopsin region of the tree from **a**, indicated these rhodopsins were not acquired from their eukaryotic host, but rather, the rhodopsins of the two Comchoano appear to have been acquired from different prokaryotes, in separate transfer events. Taxonomy of collapsed nodes was determined by the GTDB classification of rhodopsins from the genomes. Branches and labels in green are Gammaproteobacteria; UBA7916 are indicated by vertical green lines. **c**, Alignment of residues relevant to rhodopsin function from Comchoano-1, Comchoano-2, and a variety of other Type I rhodopsins. Amino acids in positions indicated by arrows are classically considered relevant for predicting function (DTE amino acid motif for Comchoano-1 and Comchoano-2), that corresponds to proton translocation. Position 105 is the position considered to predict spectral tuning; the Glutamine (Q) in position 105 for Comchoano-1 and Comchoano-2 is predicted to be tuned to blue light. Alignment figure template follows that of[127]. Unlike free-living heterotrophic marine bacteria which commonly biosynthesize the required chromophore, Comchoano lack the chromophore biosynthesis pathway. This means that the Comchoano rhodopsins cannot function without an exogenous source for the chromophore, such as from the choanoflagellate diet. Interestingly, giant viruses associated with *B. minor* also have microbial rhodopsins that can pump protons, based on heterologous expression studies[14], and have various hypothesized roles, including energy transfer in the host or acidification of host-cellular compartments. For both Comchoano and ChoanoViruses, it remains unclear where the rhodopsins localize, and how they influence host cellular function.

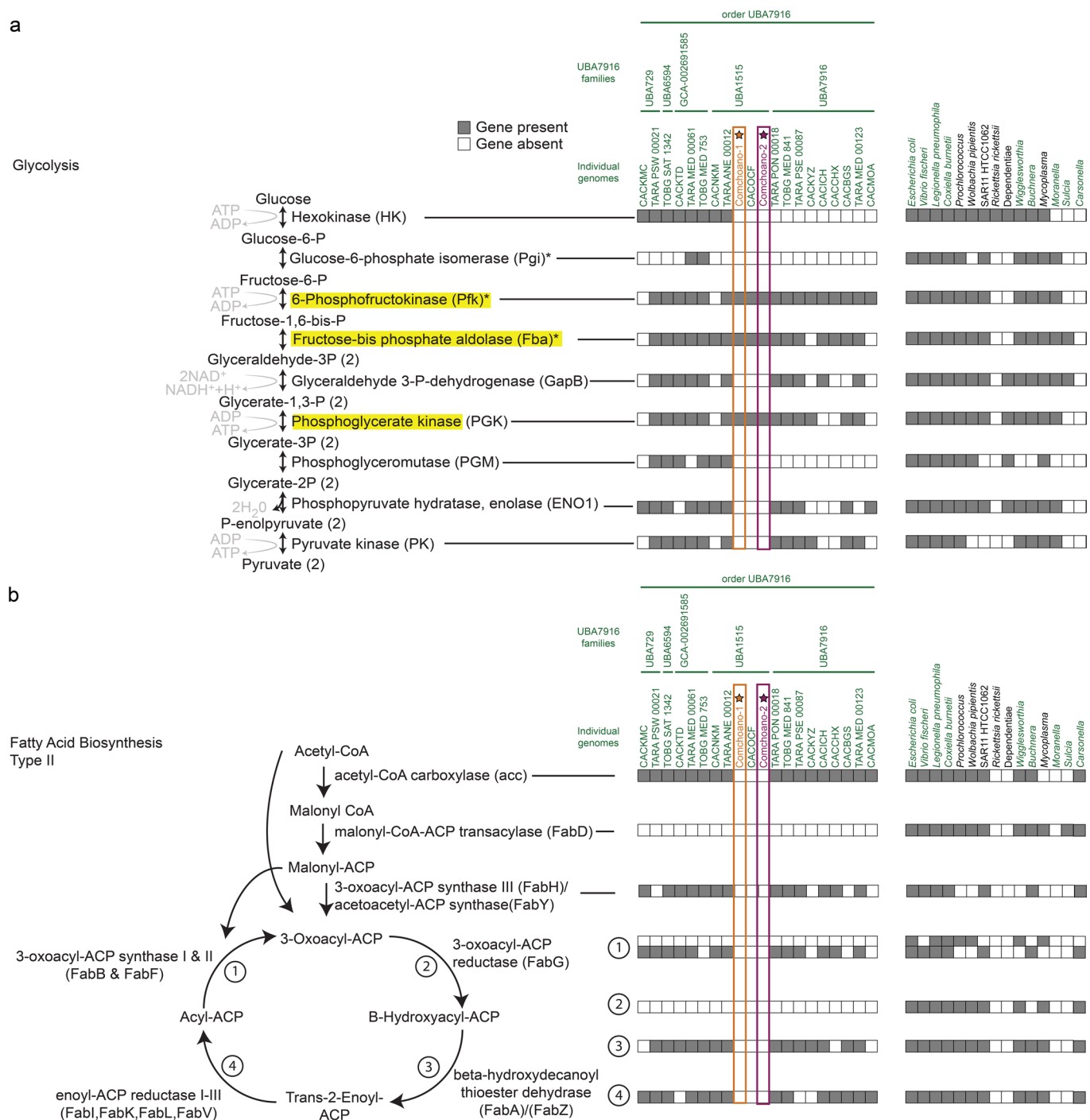

**Extended Data Fig. 9 | Notable mostly or entirely missing pathways from Comchoano-1 and Comchoano-2. a**, Glycolytic pathway is partially encoded in Comchoano-1 and Comchoano-2, but with essential genes at the beginning and end of the pathway missing. Genomes shown are all UBA7916 genomes over 50% complete, as well as those from Fig. 3d. Genes highlighted in yellow are those encoded by Comchoano-1 and Comchoano-2; those with asterisks (*) are genes that are also encoded by the Pentose Phosphate Pathway. **b**, Type II Fatty Acid Biosynthesis genes are entirely missing from Comchoano-1 and Comchoano-2, while commonly encoded in most bacteria, but missing from several endosymbionts and pathogens.

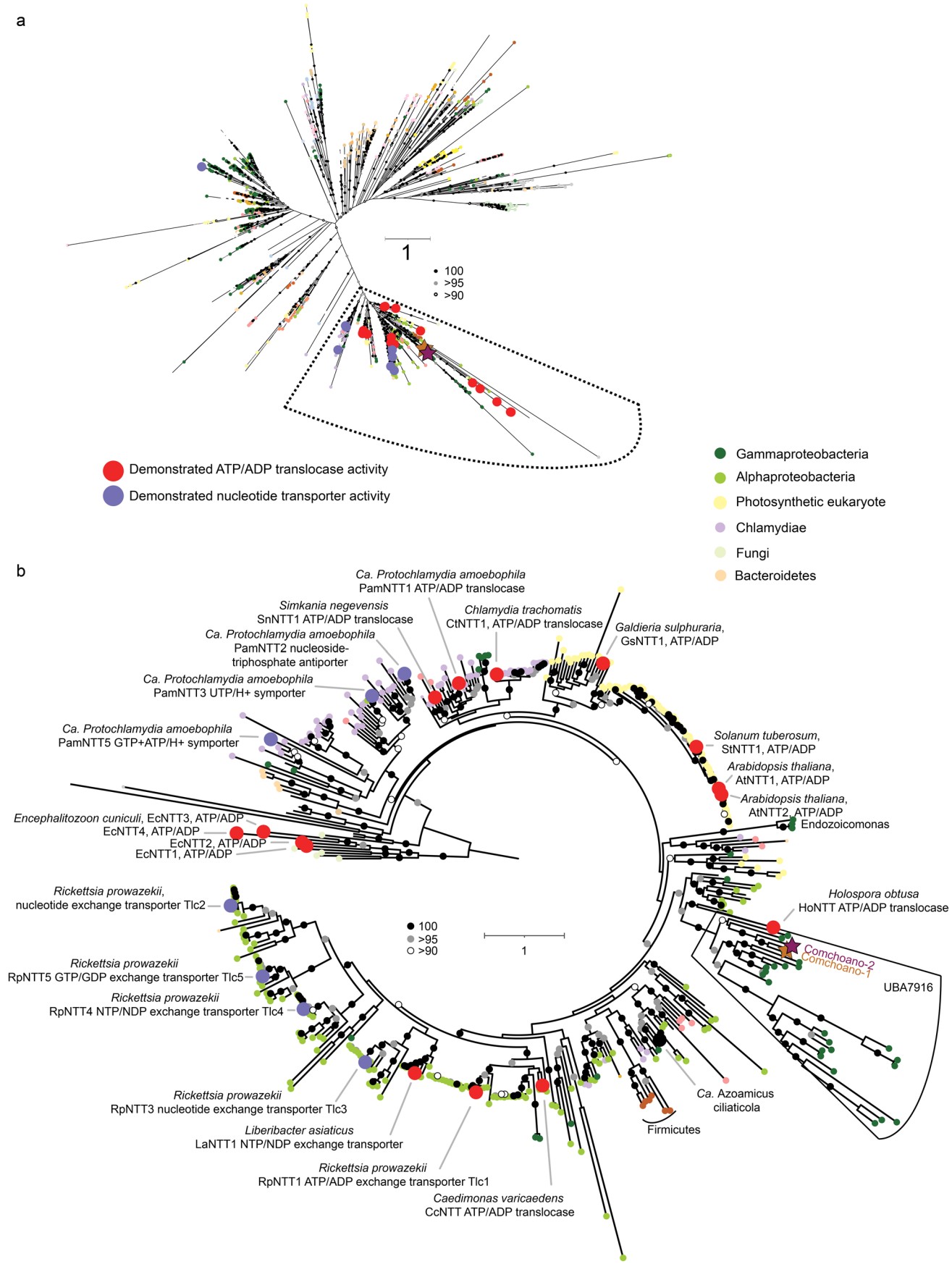

**a**

Demonstrated ATP/ADP translocase activity

Demonstrated nucleotide transporter activity

- Gammaproteobacteria
- Alphaproteobacteria
- Photosynthetic eukaryote
- Chlamydiae
- Fungi
- Bacteroidetes

**b**

*Ca. Protochlamydia amoebophila*
PamNTT1 ATP/ADP translocase

*Simkania negevensis*
SnNTT1 ATP/ADP translocase

*Ca. Protochlamydia amoebophila*
PamNTT2 nucleoside-
triphosphate antiporter

*Ca. Protochlamydia amoebophila*
PamNTT3 UTP/H+ symporter

*Ca. Protochlamydia amoebophila*
PamNTT5 GTP+ATP/H+ symporter

*Encephalitozoon cuniculi*, EcNTT3, ATP/ADP
EcNTT4, ATP/ADP
EcNTT2, ATP/ADP
EcNTT1, ATP/ADP

*Rickettsia prowazekii*,
nucleotide exchange transporter Tlc2

*Rickettsia prowazekii*
RpNTT5 GTP/GDP exchange transporter Tlc5

*Rickettsia prowazekii*
RpNTT4 NTP/NDP exchange transporter Tlc4

*Rickettsia prowazekii*
RpNTT3 nucleotide exchange transporter Tlc3

*Liberibacter asiaticus*
LaNTT1 NTP/NDP exchange transporter

*Rickettsia prowazekii*
RpNTT1 ATP/ADP exchange transporter Tlc1

*Caedimonas varicaedens*
CcNTT ATP/ADP translocase

*Chlamydia trachomatis*
CtNTT1, ATP/ADP translocase

*Galdieria sulphuraria*,
GsNTT1, ATP/ADP

*Solanum tuberosum*,
StNTT1, ATP/ADP

*Arabidopsis thaliana*,
AtNTT1, ATP/ADP

*Arabidopsis thaliana*,
AtNTT2, ATP/ADP

Endozoicomonas

*Holospora obtusa*
HoNTT ATP/ADP translocase

Comchoano-2
Comchoano-1

UBA7916

*Ca. Azoamicus
ciliaticola*

Firmicutes

**Extended Data Fig. 10 | See next page for caption.**

**Extended Data Fig. 10 | Phylogenetic reconstruction of ATP/ADP translocases of Comchoano and other UBA7916. a**, Maximum likelihood reconstruction of nucleotide transporters identified from previous publications and herein and inferred via IQ-Tree, based on 290 positions, 1,379 sequences, and phylogenetic model mtInv+F+R8. **b**, To more accurately resolve the region of the tree where Comchoano and all but one of the characterized nucleotide transport proteins were found, the sequences from (a) were subsetted as indicated from the dashed outline. The sub-tree is based on 412 positions, 413 sequences, and model LG+F+R9. The taxonomy of the host organisms are shown in colored small circles in each tree. The functionally characterized genes are indicated with larger red (ATP/ADP translocase activity) or purple (for those with other nucleotide transport affinities) circles, with labels indicating the respective functions.

# nature research

| | |
|---|---|

# Reporting Summary

Nature Research wishes to improve the reproducibility of the work that we publish. This form provides structure for consistency and transparency in reporting. For further information on Nature Research policies, see our Editorial Policies and the Editorial Policy Checklist.

## Statistics

For all statistical analyses, confirm that the following items are present in the figure legend, table legend, main text, or Methods section.

| n/a | Confirmed | |
|---|---|---|
| ☐ | ☒ | The exact sample size ($n$) for each experimental group/condition, given as a discrete number and unit of measurement |
| ☐ | ☒ | A statement on whether measurements were taken from distinct samples or whether the same sample was measured repeatedly |
| ☐ | ☒ | The statistical test(s) used AND whether they are one- or two-sided<br>*Only common tests should be described solely by name; describe more complex techniques in the Methods section.* |
| ☒ | ☐ | A description of all covariates tested |
| ☒ | ☐ | A description of any assumptions or corrections, such as tests of normality and adjustment for multiple comparisons |
| ☐ | ☒ | A full description of the statistical parameters including central tendency (e.g. means) or other basic estimates (e.g. regression coefficient) AND variation (e.g. standard deviation) or associated estimates of uncertainty (e.g. confidence intervals) |
| ☐ | ☒ | For null hypothesis testing, the test statistic (e.g. $F$, $t$, $r$) with confidence intervals, effect sizes, degrees of freedom and $P$ value noted<br>*Give P values as exact values whenever suitable.* |
| ☐ | ☒ | For Bayesian analysis, information on the choice of priors and Markov chain Monte Carlo settings |
| ☒ | ☐ | For hierarchical and complex designs, identification of the appropriate level for tests and full reporting of outcomes |
| ☐ | ☒ | Estimates of effect sizes (e.g. Cohen's $d$, Pearson's $r$), indicating how they were calculated |

*Our web collection on statistics for biologists contains articles on many of the points above.*

## Software and code

Policy information about availability of computer code

| Data collection | BD FACS Sortware (v1.0.0.650); Winlist (version 7.0, Verity Software House). MiSeq and HiSeq Reporter software as performed at the sequencing center that provided sequence data from our samples. |
|---|---|
| Data analysis | SPADES (v3.11.1); Bowtie 2 (version 2.3.0);  blast 2.6.0; HTSeq-count (2.0.1); R v3.3.2; Prodigal (v2.6.3); tRNAscan-SE (v 1.3.1); eggNOG-mapper search (2.0.1); IQ-tree (v1.5.4); MAFFT (v7.222); trimAL (v1.2); AliView (v1.18.1); RAxML; FastTree (2.1.9); Cytoscape (v3.6.1); OriFinder (2008, webserver version); nucmer (v3); samtools (1.7); bcftools (v1); hmmscan (3.1b2), MacSyFinder (1.0.5), checkM (v1.0.13); MrBayes (3.2.6); GTDB-tk (1.4.0), QIIME2 (2018.8), dada2 (2018.8.0), epa-ng (v0.3.8),  UCLUST (Edgar 2010), Cutadapt v.1.13,BD FACS Sortware (software v1.0.0.650), Geneious (20.2.4), diamond (v0.9.24.125), Predict Genome Auxotrophies tool (v.1.7.6), Pathway Tools (v22.0), txsscan (galaxy web-server version), GToTree (1.4.45), iqtree (2.0.3), iToL (version 5 and 6), cd-hit (4.8.1), and progressiveMauve (Version 2.4.0). |

For manuscripts utilizing custom algorithms or software that are central to the research but not yet described in published literature, software must be made available to editors and reviewers. We strongly encourage code deposition in a community repository (e.g. GitHub). See the Nature Research guidelines for submitting code & software for further information.

## Data

Policy information about availability of data

All manuscripts must include a data availability statement. This statement should provide the following information, where applicable:

- Accession codes, unique identifiers, or web links for publicly available datasets
- A list of figures that have associated raw data
- A description of any restrictions on data availability

Data availability Single and multi-cell sort raw data (short read archives, SRA) are available via NCBI Project Number PRJNA640955, which includes V4 16S rRNA gene

amplicon sequences from single cells, whole genome shotgun sequences from single cells, MBTS 16S and 18S V4 rRNA gene amplicons, and MBTS V4-V5 rRNA gene amplicons (see Supplementary Data 12 for individual list of SRA accessions). 18S V4 rRNA gene amplicons are available as part of Needham et al. 2019[18]. Comchoano-1 and Comchoano-2 whole genome sequences are available via accessions CP092900 and JAKUDN000000000 and their full-length 16S rRNA gene sequences are deposited as OM801198 and OM801197. Alignments, tree files, and processed amplicon data are available via FigShare (doi: 10.6084/m9.figshare.c.5850662).

# Field-specific reporting

Please select the one below that is the best fit for your research. If you are not sure, read the appropriate sections before making your selection.

☐ Life sciences ☐ Behavioural & social sciences ☒ Ecological, evolutionary & environmental sciences

For a reference copy of the document with all sections, see nature.com/documents/nr-reporting-summary-flat.pdf

# Ecological, evolutionary & environmental sciences study design

All studies must disclose on these points even when the disclosure is negative.

| | |
|---|---|
| Study description | Single cells were sorted from seawater and genome sequenced. There was a total of 188 cells sorted from this experiment all of which had 18S and 16S PCR performed on them. We report the ASV data for all of those that produced results. We found that 1% of the choanoflagellate cells had one of two types of Comchoano bacteria associated with them which we report here. We also contextualize these findings by demonstrating the Comchoano and choanoflagellate abundance in available datasets such as from the location of study the Monterey Bay, as well as a survey that also had 18S and 16S data from the San Pedro Ocean Time-series. We also surveyed ASV data from global circumnavigations to demonstate the abundance of Comchoano and choanoflagellates in the global ocean. Most of the environmental ASV data were not in replicate technically (whether they are original to our manuscript or the other ASV datasets) but were in either time-series or spatial distributions allowing a sense of their variability over space and time. Otherwise, the technical sample design structures of e.g. factorial, nested, hierarchical per se were not systematically followed in our environmental study. |
| Research sample | Seawater from offshore central California, USA, see below within "Data Collection". The sample is chosen to be basically representative of the offshore central California coast during spring and coincided with sampling opportunity. Amplicon sequencing from bulk samples and metatranscriptomes are from seawater filtered via vacuum pump, with no further manipulation. For the flow cytometry, lysotracker staining as added to help identify heterotrophic protists by staining of their acidic components (such as food vacuoles). The cells sorted of B. minor are expected to be representative of the choanoflagellate community on the day of sorting in the surrounding waters. Malaspina, Tara, and San Pedro Ocean time-series data were downloaded from previously published datasets as described in the manuscript. Amplicon sequencing and metatranscriptome sequencing were all sequenced on Illumina sequencing and deposited as raw data via NCBI Project Number PRJNA640955. |
| Sampling strategy | NA, no sample size calculation was necessary. There are no statistical analyses of populations etc. |
| Data collection | Seawater for sorting was collected on 20 March 2014 at Station M2 (36.688°N, 122.386°W, Fig. 1a) using Niskin bottles mounted on a CTD rosette. Water from 20 m depth was pre-filtered through a 30 μm mesh, concentrated by gravity over a 0.8 μm Supor filter to about 250 times concentration, stained with LysoTracker Green DND-26 (final concentration, 25 nM), and run on a BD Influx flow cytometer equipped with a 488 nm laser using sterile nuclease-free 1x PBS as sheath fluid. The sorted population was discriminated based on positive LysoTracker signal (i.e., fluorescence detected in 520 ± 35 nm bandpass filter under 488 nm excitation) as compared to an unstained sample and absence of chlorophyll-a autofluorescence (i.e., 692 ± 40 nm filter), and similar Forward Angle Light Scatter (FALS) to select coherent populations of heterotrophic eukaryotes (Fig. 1b). Single cells were sorted into a 384-well plate using the Single-Cell sorting mode from the BD FACS Sortware (software v1.0.0.650). A subset of wells was left empty or received 20 cells for negative and positive controls, respectively. The plate was illuminated with UV for 2 min prior to performing the sort and covered with foil and frozen at -80°C immediately after sort completion. Camille Poirier performed and recorded all flow cytometric analyses. David Needham recorded all genome sequencing data and choanoflagellate single cell amplicon data. Charmaine Yung processed amplicon gene surveys from MBTS. Lisa Sudek performed PCR on MBTS samples. AJ Limardo helped with MBTS sampling. |
| Timing and spatial scale | In addition to the flow cytometric single cell sorting sampling described above, seawater samples were taken for rRNA gene surveys and metatranscriptomic surveys from a total of 3 months in 2014 and one in 2015 from surface waters inside and outside of the Monterey Bay. Other months sampled were not analyzed for this project due to timing and applicability to the manuscript (overlapping with single cell sorting most closely so provide the most relevent context). Additionally, previously published data were used from the southern california oceanographic region's San Pedro Time-series (2010) as well as surface waters from the circumnavigating Malaspina and Tara Ocean's cruises were used. |
| Data exclusions | None |
| Reproducibility | 188 single choanoflagellate cells were investigated with 12% of them found to be associated with 1 of 2 types of Comchoano. Multiple nearly identical Comchoano genomes were recovered from single cell sequencing of choanoflagellate cells. Extensive analyses were performed to demonstrate the level of single nucleotide variation between the genomes. All the single cell analyses report come from a single experiment with B. minor never becoming prevalent enough to repeat the experiment, though we do find that B. minor and Comchoano and their relatives globally distributed |
| Randomization | not applicable, samples were sampled from diverse, discrete microbial community where randominzation is not relevent, no human subjects |

| Blinding | not applicable, no human subjects were used. |
|---|---|

Did the study involve field work?  ☒ Yes  ☐ No

## Field work, collection and transport

| Field conditions | The ocean sample on which sorting was performed was collected from 20 m depth in the mixed layer at station M2 (36.688 °N; 122.386 °W, 56 km from shore, Fig. 1a) on 20 March 2014 where temperatures were 12.7 at the depth of sampling, 12.8 at the surface and above 12 to the thermocline around 45 m. The 20m depth sample was the chlorophyll maximum depth where the chlorophyll concentration was 2.1 µg/L (1.0 µg/L at the surface and 1.8 µg/L at 40m) |
|---|---|
| Location | main flow cytometric single cell sorting undertaken at 20 m depth: 20 March 2014 in the northern pacific ocean (36.688°N, 122.386°W). Like this sample, the others are presented with their latitude and longitude in our manuscript (or in previously published reports, their original publication). |
| Access & import/export | Sampling was from surface ocean waters in the US economic zone or international waters and processed primarily by scientists and facilities in the US, since moved to the EU (Germany). The samples were water samples of typically 10 L from various depth in the water column, especially the surface photic zone and were as such non invasive, do not harm the environment and did not involve CITES species or any other endangered or animals of any economic value. |
| Disturbance | a maximum of 30 L of ocean water was collected for a given sample, this does not disturb the ecosystem. |

# Reporting for specific materials, systems and methods

We require information from authors about some types of materials, experimental systems and methods used in many studies. Here, indicate whether each material, system or method listed is relevant to your study. If you are not sure if a list item applies to your research, read the appropriate section before selecting a response.

## Materials & experimental systems

| n/a | Involved in the study |
|---|---|
| ☒ | ☐ Antibodies |
| ☒ | ☐ Eukaryotic cell lines |
| ☒ | ☐ Palaeontology and archaeology |
| ☒ | ☐ Animals and other organisms |
| ☒ | ☐ Human research participants |
| ☒ | ☐ Clinical data |
| ☒ | ☐ Dual use research of concern |

## Methods

| n/a | Involved in the study |
|---|---|
| ☒ | ☐ ChIP-seq |
| ☐ | ☒ Flow cytometry |
| ☒ | ☐ MRI-based neuroimaging |

## Flow Cytometry

### Plots

Confirm that:

☐ The axis labels state the marker and fluorochrome used (e.g. CD4-FITC).

☐ The axis scales are clearly visible. Include numbers along axes only for bottom left plot of group (a 'group' is an analysis of identical markers).

☐ All plots are contour plots with outliers or pseudocolor plots.

☐ A numerical value for number of cells or percentage (with statistics) is provided.

### Methodology

| Sample preparation | Sorted seawater was collected on 20 March 2014 at Station M2 (36.688°N, 122.386°W, Fig. 1B) using Niskin bottles mounted on a CTD rosette. Water from 20 m depth was pre-filtered through a 30 µm mesh, concentrated by gravity over a 0.8 µm Supor filter to about 250 times concentration and stained with LysoTracker Green DND-26 (final concentration, 25 nM). |
|---|---|
| Instrument | BD Influx flow cytometer equipped with a 488 nm laser using sterile nuclease-free 1x PBS as sheath fluid. |
| Software | BD FACS(TM) Software v 1.2.0.142 (run software); Verity Software House WinList 9.0 (figure display software) |
| Cell population abundance | There is no population abundance analysis in this manuscript. The sorting was used to separate cells into individual wells that were then sequenced (as described in methods). The pre-concentration methods used preclude derivation of numerical information for the flow cytometric analyses. Cell population abundances were mostly inferred from rRNA gene sequencing locally and at locations around the global ocean. |
| Gating strategy | The sorted population was discriminated based on positive LysoTracker signal (i.e., fluorescence detected in 520 / 35 nm |

Gating strategy

bandpass filter under 488 nm excitation) as compared to an unstained sample and absence of chlorophyll-a autofluorescence (i.e., 692 / 40 nm filter), and similar Forward Angle Light Scatter (FALS) to select coherent populations of heterotrophic eukaryotes (Extended Data Fig. 1a). Single cells were sorted into a 384-well plate using the Single-Cell sorting mode from the BD FACS Sortware (software v1.0.0.650).

☒ Tick this box to confirm that a figure exemplifying the gating strategy is provided in the Supplementary Information.

