## [Peer Review File. · Nature Microbiology]

Peer Review Information

Journal: Nature Microbiology

Manuscript Title: The microbiome of a bacterivorous marine choanoflagellate contains a resource-demanding obligate bacterial associate

Corresponding author name(s): Alexandra Worden

Reviewer Comments & Decisions:

Decision Letter, initial version:

Dear Professor Worden,

Thank you for your patience while your manuscript "Single marine bacterivore microbiomes reveal members of a diverse uncultivated bacterial lineage are resource demanding symbionts" was under peer-review at Nature Microbiology. It has now been seen by 3 referees, whose expertise and comments you will find at the end of this email. Although they find your work of some potential interest, they have raised a number of concerns that will need to be addressed before we can consider publication of the work in Nature Microbiology.

In particular, the referees state that it will be important to provide identification and intracellular localization of the putative symbionts by fluorescence in situ hybridizations (FISH). Furthermore, the referees have concerns about the level of support for some of the conclusions reached, specifically they question whether the association really is symbiotic. Editorially, we feel it will be important to perform the additional FISH experiments, and to provide additional support for the idea of a symbiosis relationship, including how the bacterium benefits the host. The remaining referees' concerns should also be addressed in full.

Should further experimental data allow you to address these criticisms, we would be happy to look at a revised manuscript.

Please include a data availability statement as a separate section after Methods but before references, under the heading "Data Availability". This section should inform readers about the availability of the data used to support the conclusions of your study. This information includes accession codes to public repositories (data banks for protein, DNA or RNA sequences, microarray, proteomics data etc...),

2references to source data published alongside the paper, unique identifiers such as URLs to data repository entries, or data set DOIs, and any other statement about data availability. At a minimum, you should include the following statement: "The data that support the findings of this study are available from the corresponding author upon request", mentioning any restrictions on availability. If DOIs are provided, we also strongly encourage including these in the Reference list (authors, title, publisher (repository name), identifier, year). For more guidance on how to write this section please see: <http://www.nature.com/authors/policies/data/data-availability-statements-data-citations.pdf>

- * Include a "Response to referees" document detailing, point-by-point, how you addressed each referee comment. If no action was taken to address a point, you must provide a compelling argument. This response will be sent back to the referees along with the revised manuscript.
- * If you have not done so already we suggest that you begin to revise your manuscript so that it conforms to our Letter format instructions at <http://www.nature.com/nmicrobiol/info/final-submission>. Refer also to any guidelines provided in this letter.
- * Include a revised version of any required reporting checklist. It will be available to referees (and, potentially, statisticians) to aid in their evaluation if the manuscript goes back for peer review. A revised checklist is essential for re-review of the paper.

{redacted}

Note: This url links to your confidential homepage and associated information about manuscripts you may have submitted or be reviewing for us. If you wish to forward this e-mail to co-authors, please delete this link to your homepage first.

2Nature Microbiology is committed to improving transparency in authorship. As part of our efforts in this direction, we are now requesting that all authors identified as 'corresponding author' on published papers create and link their Open Researcher and Contributor Identifier (ORCID) with their account on the Manuscript Tracking System (MTS), prior to acceptance. This applies to primary research papers only. ORCID helps the scientific community achieve unambiguous attribution of all scholarly contributions. You can create and link your ORCID from the home page of the MTS by clicking on 'Modify my Springer Nature account'. For more information please visit www.springernature.com/orcid.

If you wish to submit a suitably revised manuscript we would hope to receive it within 6 months. If you cannot send it within this time, please let us know. We will be happy to consider your revision, even if a similar study has been accepted for publication at Nature Microbiology or published elsewhere (up to a maximum of 6 months).

{redacted}

Reviewer Expertise:

Referee #1: Biogeochemistry, environmental microbiology

Referee #2: Microbial ecology, symbiosis

Referee #3: Microbial ecology, microbial evolution

Reviewer Comments:

Reviewer #1 (Remarks to the Author):

Congratulations to the authors for generating a wonderful and fascinating paper reporting the discovery of bacterial associates in a widely-distributed oceanic choanoflagellate, *Bicosta minor*. This paper is a fantastic example of technology-enabled discovery: the novel association was discovered by flow activated cell sorting and sequencing *B. minor* from open ocean water samples. The proposed associate, called *Comchoano*, is most similar to the human pathogen, *Coxiella burnetii*. Evidence for the association was discovered in samples from across different ocean basins. Indication of an associative life-style include co-association of *Comchoano* and *B. minor*, a reduced genome size of *Comchoano*, rare occurrence of *Comchoano* as free-living bacteria, and lack of key metabolic functions, including loss of pathways for glycolysis and membrane lipid components and for amino acid and B-vitamin production. Additional evidence arises from the fact that the *Comchoano* encode Type IV secretion systems, which is involved in infection and potential pathogenicity, that they appear to be able to import energy (as ATP) from the host, and that they encode a number of proteins that facilitate host-pathogen interactions.

I am using the words associate/association instead of symbiont, because I am not convinced that this is symbiosis. While the authors show ample evidence that this association is obligate (*Comchoano* seems

3ill-equipped for a free-living lifestyle), it is not clear to me what the host obtains from the association. To the contrary, the Comchoano appears to be a metabolic sink for the host, which suggests a pathogenic rather than a symbiotic lifestyle. The Coxiella use their secretion system to inject effectors into the host and these effectors encourage Coxiella survival while inhibiting the hosts immune system (and more) and Comchoano appear to do the same. The fact that the infection mechanism is similar in Comchoano and Coxiella is extremely interesting.

In the absence of any evidence pointing to a metabolic benefit for the host *B. minor*, why is this association presented a symbiosis instead of a pathogen? I ask that the authors provide additional evidence for symbiosis or instead present this as an obligate association driven by yet-to-be-determined metabolic exchanges.

The methods employed in this study are state-of-the-art and comprehensive. The data from the Monterey Bay are complemented from information mined from the San Pedro Ocean Time series and the Malaspina 2010 and Tara Oceans circumnavigation data sets. The evidence for association of Comchoano and *B. minor* is convincing.

I do not view the lack of evidence of true symbiosis as a significant weakness – I would like to see the terminology utilized be fully reflective of the documented nature of the relationship. Microscopic evidence of the association – I recognize that this is a big ask! - would be extremely useful.

Finally, I was surprised that the 2021 paper paper by Jon Graf et al 2021, 'Anaerobic endosymbiont generates energy for ciliate host by denitrification' (Nature - <https://doi.org/10.1038/s41586-021-03297-6>) is not cited in this paper. Yes the paper reports discovery of a freshwater ciliate with a gammaproteobacterial symbiont that generates energy (as ATP) from the host by denitrification but re-reading the Graf paper during preparation of this review, it seems there is relevance.

Specific Comments

Manuscript

Line 111 – 113 – though this is likely true, this comment is a bit of a stretch and could easily be deleted.

Line 149 – is manifest (instead of MANIFESTS)

Line 201 – To utilize the PPP... (instead of 'To function the PPP)

Supplemental

Pg 4, line 11 – GC content (add content)

Pg 9, line 11: suggest: Genomes with fewer than five of the ribosomal proteins (n=3) were removed [removed instead of moved]; however, one of these (CACFNL) was included in the 16S rRNA tree described below.

Pg 20 – figure caption – ... and Comchoano-2, as is the case for several endosymbionts and pathogens, but which are encoded by free-living and particle-associated bacteria.

Reviewer #2 (Remarks to the Author):

This manuscript reports on the genome sequence-based identification of a yet uncultured gammaproteobacterial lineage as putative symbionts of the choanoflagellate *Bicosta minor*. This is supported by the co-occurrence of bacteria and choanoflagellates sequences in published amplicon data-sets and after FACS-sorting of environmental samples, as well as by genome features characteristic of host-associated microbes.

This is an exceptionally well-written manuscript I truly enjoyed reading. The comparative genome analysis is sound, and the evidence for the gammaproteobacteria being associated with *B. minor* is convincing. Unfortunately, the final proof, i.e. the identification and intracellular localization of the putative symbionts by fluorescence in situ hybridizations is lacking. This is particularly surprising as the sorting by FACS seems to work very efficiently as suggested by the data provided in Figure 1b.

The discovery of bacterial symbionts in choanoflagellates in this study is undoubtedly very interesting; the study identifies a yet unknown clade of bacteria as putative symbionts, and choanoflagellates are an attractive model systems from different points of view. The ecosystem-level implications of the study as summarized in the abstract and the last paragraph of the results/discussion section, however, appear overstated as microbial symbionts in heterotrophic marine protists have been described before.

1. Is the name Comchoano for the two *B. minor*-associated bacteria intended to represent a new Candidatus name? If that was the case, the addition of a formal 'Candidatus' description in the supplement would be useful.

2. Are Comchoano 1 and 2 separate species or even genera based on ANI and AAI analyses?

3. Fig 1b: *B. minor* cells were apparently also associated with Rickettsiales ASVs. As rickettsiae are known intracellular microbes, it would be interesting to know more about these ASVs.

4. L. 183: Members of the Coxiellales, the Berkiellales, and Legionellales include prominent symbionts and pathogens of protists. This should be mentioned here.

5. L. 192: I understand that genes for retinal synthesis are absent in the Comchoano genomes. If so, I suggest mentioning this in the main text as this renders the function as a proton pump unlikely.

6. Metabolic pathway analyses and Figure 3d: The selection of species for comparative analysis of the completeness of metabolic pathways includes only a single protist-associated symbiont. Would it be helpful to include other known (obligate) intracellular symbionts of protists in this analysis?

I am not sure whether the ordering of species according to their genome size in Fig 3d is very helpful. Perhaps a clustering based on the completeness information would be more intuitive to show similarities among species with respect to their metabolic potential.

57. Can the authors confirm that the identified ATP/ADP translocase belongs to the subfamily of these nucleotide transporters that is specific for ATP/ADP antiport?
8. Host-relationship of Comchoano symbionts is discussed exclusively in the context of microbial symbionts of animals (ll. 242-259). I suggest including findings from other protists-associated symbionts as well. Their genomes can be enriched in proteins with eukaryotic domains, exemplified by the amoeba symbiont *Amoebophilus asiaticus* (PMID 20023027).
9. Figure 4b is not essential and could be moved to the supplement.
10. L 287. Typo: *Bdellovibrio*-like, Coxiellales
11. Data availability: Genbank accession numbers and DOI to related files are missing.

Reviewer #3 (Remarks to the Author):

Needham and colleagues report the discovery of a novel bacterial lineage ("Comchoano") inferred to be physically associated with the choanoflagellate *Bicosta minor*, most likely as endosymbionts. Choanoflagellates typically graze on bacteria and a metabolic reconstruction of genomes representing two Comchoano species recovered directly from single choanoflagellate cells or population sorts show several features consistent with host dependence and evasion of the hosts digestive processes.

Overall, this is a carefully conducted study with numerous lines of mostly circumstantial evidence supporting the conclusion that Comchoano are endosymbionts of *B. minor*. However, the most compelling evidence is missing: direct visualisation of Comchoano in their hosts using specific FISH probes. Did the authors attempt this? Are there sound reasons not to include visualisation evidence? This should at least be discussed briefly somewhere in the ms as visualisation of the lineage is conspicuous by its absence, and ideally FISH images should be part of the study.

Specific comments

Line 33. I recommend proposing Candidatus genus and species names for Comchoano-1 and Comchoano-2 which are fine as unique genome identifiers, but not adequate for nomenclature. The names could still easily be based on Comchoano, for example, *Candidatus Comchoanobacter bicostavorans* or *Ca. C. intracellularis*. If you feel particularly motivated, you could even propose higher rank names based on the genus name, e.g. family Comchoanobacteraceae, and order Comchoanobacterales since there are no existing Latin names in this taxonomic neighborhood. I also highly recommend specifying type genomes for Comchoano-1 and -2 as described by Chuvochina et al, 2019:

<https://www.sciencedirect.com/science/article/pii/S0723202018302005>

See also comment below about Extended Data Fig. 6

Line 111. Doesn't the time delay between host and endosymbiont population relative abundances suggest a lytic pathogen lifestyle more consistent with parasitism?

6Line 205. Some have questioned whether ATP/ADP translocases definitively indicate energy parasitism given the widespread distribution of nucleotide transport proteins in bacteria and other potential roles. See Major et al., 2017
<https://academic.oup.com/gbe/article/9/2/480/2970297>

Line 270. Can the authors infer where host association originated in the UBA7916 on a bacterial species tree? Could it predate the common ancestor of this order?

Line 272. Are there known T4SS effector protein signalling domains in species with homologous systems indicating secretion using this system that could be searched for? Are there other features of the described pathogens carrying the T4SS (line 286) relating to their pathogenicity that could be searched for in Comchoano? Since *B. minor* has previously been observed to develop multicellularity or switch to sexual reproduction as a result of bacterial effector lipids and proteins, could the presence of the T4SS be related as the mating cycle would spread Comchoano between hosts. I'd like to see if the known effectors (eg. *EroS*) are present in the Comchoano genomes, as you speculate that Comchoano are being released from lysed host cells but don't suggest any mechanisms for transmission/infection. Were any other secretion signalling domains identified? Would be interesting to identify the repertoire of potentially secreted proteins to inform hypotheses related to host interaction.

Line 287. Note correct spelling of *Bdellovibrio*

Fig. 2. Are there any other euks identified in the SPOT data that could be correlated with Comchoano abundance? Data is a little variable but there's a suggestion of a spike in the free-living fraction ~day 18 without identified change in *B. minor* abundance. Conversely, would the time delayed increase in Comchoano in the free-living fraction be consistent with a lytic pathogen lifestyle?

Fig. 3. Panel b could be more informative if rows were allowed to cluster by gene content rather than the apparent sorting by genome size which complicates comparison between members of the UBA7916 lineage and between UBA7916 members and other species.

Extended Data Fig. 6. Based on panel b, Comchoano-1 and -2 are at least separate species if not in separate genera. GTDB-Tk, should provide some indication of the degree of novelty for this based on ANI and RED values. What is the ANI between the two Comchoano genomes? How much synteny do they share? Also, where do the genomes CACOFC and others in the order UBA7916 come from? They don't have genome accession ids so were they also obtained in the present study? If not, please provide genome accessions as you do for other reference genomes in panel b.

Author Rebuttal to Initial comments

Reviewer #1:

Congratulations to the authors for generating a wonderful and fascinating paper reporting the discovery of bacterial associates in a widely-distributed oceanic choanoflagellate, *Bicosta minor*. This paper is a fantastic example of technology-enabled discovery: the novel association was discovered by flow activated cell sorting and sequencing *B. minor* from open ocean water samples. The proposed associate, called Comchoano, is most similar to the human pathogen, *Coxiella burnetii*. Evidence for the association was discovered in samples from across different ocean basins. Indication of an associative life-style include co-association of Comchoano and *B. minor*, a reduced genome size of Comchoano, rare occurrence of Comchoano as free-living bacteria, and lack of key metabolic functions, including loss of pathways for glycolysis and membrane lipid components and for amino acid and B-vitamin production. Additional evidence arises from the fact that the Comchoano encode Type IV secretion systems, which is involved in infection and potential pathogenicity, that they appear to be able to import energy (as ATP) from the host, and that they encode a number of proteins that facilitate host-pathogen interactions.

We thank the reviewer for their enthusiasm for our discoveries.

I am using the words associate/association instead of symbiont, because I am not convinced that this is symbiosis. While the authors show ample evidence that this association is obligate (Comchoano seems ill-equipped for a free-living lifestyle), it is not clear to me what the host obtains from the association. To the contrary, the Comchoano appears to be a metabolic sink for the host, which suggests a pathogenic rather than a symbiotic lifestyle. The *Coxiella* use their secretion system to inject effectors into the host and these effectors encourage *Coxiella* survival while inhibiting the hosts immune system (and more) and Comchoano appear to do the same. The fact that the infection mechanism is similar in Comchoano and *Coxiella* is extremely

interesting.

Thank you for sharing these thoughts. We agree that it is most likely (or indeed there is ample evidence as the reviewer says) that the Comchoano are obligately associated and metabolic sinks for the host. We apologize for not clearly communicating that we were using the term ‘symbiont’ in a broad sense which can include relationships along a continuum from parasite to mutualist.

To address these comments, we have updated the manuscript to include a new reference Drew et al. “Microbial evolution and transitions along the parasite-mutualist continuum”, Nature Reviews Microbiology, 2021 describing this continuum so that the first paragraph of the introduction now states:

“However, identification of symbioses, which encapsulate a range of relationships along a continuum from mutualistic to antagonistic¹, has been impeded by challenges behind capturing or culturing physically-interacting marine microbes, inhibiting the advancement of this critical area of ocean science^{2,3}.”

Additionally, throughout the manuscript we have moved away from the use of “symbiont” in preference for “obligately associated” or “obligate associates”. Furthermore, we’ve modified wording later in the manuscript to reflect complexity in the potential metabolic sink scenario so that it now states as:

“Thus, Comchoano’s requirements appear to impose an energy and resource ‘tax’ on hosts, with no apparent return benefit. The impact of this ‘tax’ could range from mostly neutral for the host, potentially shifting under ocean conditions that are challenging for host growth and energy acquisition to strict pathogenicity.”

In the absence of any evidence pointing to a metabolic benefit for the host *B. minor*, why is this association presented a symbiosis instead of a pathogen? I ask that the authors provide additional evidence for symbiosis or instead present this as an obligate association driven by yet-to-be-determined metabolic exchanges.

When we submitted this manuscript employing the broader definition of symbiosis (as in the NRM review) and in the past when presenting it as a pathogenetic relationship received comments that we should instead work with this broader definition rather than narrowing in on pathogenicity (which indeed we find very interesting and important). We have now tried to clarify this and modified the manuscript to address (as in details provided in the above response). Additionally, we replaced the term "symbiont" in title with “obligate association” (the term recommended by this reviewer), as we mentioned above.

The methods employed in this study are state-of-the-art and comprehensive. The data from the Monterey Bay are complemented from information mined from the San Pedro Ocean Time series and the Malaspina 2010 and Tara Oceans circumnavigation data sets. The evidence for association of Comchoano and *B. minor* is convincing.

We appreciate the reviewer's assessment of the distributional data and evaluation of the convincingness of the evidence for association.I do not view the lack of evidence of true symbiosis as a significant weakness – I would like to see the terminology utilized be fully reflective of the documented nature of the relationship. Microscopic evidence of the association – I recognize that this is a big ask! - would be extremely useful.

As mentioned above, we have now altered usage on symbionts to wording the reviewer recommends and also (in the manuscript) point to the recent NRM review¹ that characterizes a broad continuum of relationships as representing symbioses.

Because neither the host, nor the bacterium are cultured, and neither are in enrichment versions, the most informative types of microscopy for visualizing subcellular aspects of small protists and interactions (e.g., TEM etc.) are not currently possible. Recognizing the limitations to visualization (expanded below for FISH), we focused additional effort in this study on making sure that we had a gapless, circularized genome assembly of the uncultivated Comchoano wherein gene absences could be confirmed; our success is most likely due to the Comchoano cells being in multiple clonal copies within the choanoflagellate host. The complete genome sequence, as well as analysis of its content, membranes etc. and phylogenetic distances to other taxa combined with FACS isolation of the co-associated entities provide robust grounds for establishing the relationship. Other studies of uncultivated predatory marine protists have not reached this level of genome completion.

To further discuss epifluorescence microscopy paired with FISH, several factors would make results largely inconclusive. First, given the small size of the single-celled heterotrophic host, visualization of the bacterium would likely leave it unclear as to whether the bacterium was a food particle or endosymbiont. FISH visualization of now established symbionts of uncultivated marine protists target photosynthetic protists that do not consume bacteria as prey, making the interpretation of FISH results from symbiont-targeted probes more conclusive. Also problematic are the statistics of *in situ* FISH studies (since both members are uncultured), which make it challenging to capture robust relationships that would be above the signal to noise (that can occur in field samples). This is particularly an issue for small marine protistan predators which are most often present at orders of magnitude lower abundance than e.g., eukaryotic algae, or than many free-living marine bacterial taxa. Additionally, in this case, Comchoano was observed in 12% of hosts (based on our data) by single cell sorting, thus host abundances would need to be much higher than they are for small predatory protists because the statistics for observing the relationship become yet another order of magnitude more difficult to achieve than just observing the hosts.

In order to alert the reader of the potential limitations and reasons we instead pursued additional sequencing, assembly and genome analyses (alongside the other methods/results presented) the main text now states:

“Visualization of the *Bicosta*-Comchoano relationship has not yet been achieved largely due to their uncultivated status and ephemeral abundances in their dynamic habitat where the likelihood of reencountering the targeted interaction on the spatial and temporal scales at which oceanographic research is conducted is low. Additionally, interpretation of bacterial roles in small uncultivated heterotrophic protists like choanoflagellates, that also contain bacterial prey, has limitations by epifluorescence microscopy and fluorescence in situ hybridization, especially combined with statistical challenges arising

for uncultured entities that live at relatively low in abundance in the ocean and resulting factors like insufficient signal to noise/negative fluorescence ratios in diverse field samples that contain naturally autofluorescent particles^{4,5}. With respect to choanoflagellates and more resolved transmission electron microscopy (which is possible for abundant cells in culture), a possible endosymbiont has been noted in a cultured Baltic Sea species from brackish waters, however the putative host-microbe interaction, functional and phylogenetic features of the putative bacterium⁶ all remain unknown. Hence, in many regards, the completion and analysis of the Comchoano genomes and technology-enabled methods used herein for establishing their physical association with choanoflagellates provides the most compelling evidence yet possible for their relationship and its ramifications.”

Finally, please note, the lab does have experience with implementing FISH probes on uncultured marine predatory protists⁴ and to our knowledge FISH has not been performed on uncultivated choanoflagellates (only on cultures⁷). We did develop FISH probes (in particular a hybridization chain reaction FISH approach) and results for two probes were successful on lab “clone-FISH” samples wherein we generated three types of *E. coli* mutants that expressed (in an orientation specific manner) the 18S or 16S rRNA gene(s) of each of the uncultured entities. These probes were not implemented in field samples because most of the collections occurred over a period of years before we had knowledge of the relationship (and therefore did not have the sequence of purpose to collect field FISH samples due to overall limitations to cruise activities) – and follow-up cruises in this highly dynamic ecosystem did not encounter choanoflagellate-like populations at suitable abundance (based on flow cytometric screening). These probe sequences can be added if desired by the reviewer and might support future efforts to pursue these taxa in the environment.

Finally, I was surprised that the 2021 paper paper by Jon Graf et al 2021, ‘Anaerobic endosymbiont generates energy for ciliate host by denitrification’ (Nature - <https://doi.org/10.1038/s41586-021-03297-6>) is not cited in this paper. Yes the paper reports discovery of a freshwater ciliate with a gammaproteobacterial symbiont that generates energy (as ATP) from the host by denitrification but re-reading the Graf paper during preparation of this review, it seems there is relevance.

We are delighted to have the opportunity to add discussion of the Graf et al. 2021 paper which came out after the preparation of our manuscript. We have now included this freshwater endosymbiont of a ciliate in the metabolic comparisons between Comchoano and other obligate and facultative symbionts (mutualists and pathogens) as well as free-living bacteria (Figure 3d). We have added additional discussion on it (wonderful that the ciliate is relatively large, ~25

microns across, making visualization more meaningful).

“Additionally, a long-branching group (*Candidatus Azoamicus*) in an unsupported position adjacent to Legionellales and Francisellales (with 84% nucleotide identity to Comchoano) is an obligate endosymbiont of an anaerobic ciliate from waste sludge and other freshwater environments, wherein the much reduced bacterium appears to provide energy to the host, seemingly akin to mitochondria⁸.”

In addition, we discuss these findings for this freshwater symbiotic relationship/the translocase.

“The ATP/ADP translocase encoded by Comchoano and most UBA7916 (Fig. 3d, Extended Data Fig. 8) is phylogenetically similar to those reported in other ‘energy parasites’ (Extended Data Fig. 12a,b)⁹. Some translocases within this broad protein family have been reported to have affinity for nucleotides such as guanosine di/tri-phosphate, calling for further experimental studies of translocase affinities^{8,10}. Recently, this translocase was implicated as being the mechanism by which *Ca. Azoamicus* endosymbionts provide ATP to their freshwater ciliate hosts. Here, we find the closest affiliation of the Comchoano translocases being with an ATP/ADP translocase in an alphaproteobacterial endosymbiont (Extended Data Figure 12b), with both maintaining all the motifs for ATP/ADP translocation that have been demonstrated as essential based on site-specific mutations (Extended Data Fig. 13) in versions present in parasitic fungi¹¹.”

Specific Comments

Manuscript

Line 111 – 113 – though this is likely true, this comment is a bit of a stretch and could easily be deleted.

Currently thoughts captured in this sentence have been included because they are embedded in analyses performed due to Reviewer 3 queries and requests for exploration of additional environmental information; new analyses on frequencies and relationships precede the comment. The section now reads:

“We then turned to time-resolved data to gain insight into ecological relationships, specifically to data collected daily-to-weekly at SPOT, a time-series study in Pacific waters of the Southern California Bight (Fig. 1a). Across the daily portion of the time-series, Comchoano and *B. minor* relative amplicon abundances increased contemporaneously in the ‘Protistan size fraction’ (1 – 80 μm) ($r = 0.74, 0.77$; $p=0.0001, 0.00004$ for Comchoano-1 and 2, respectively; Fig. 2b, Supplementary Data 1). Moreover, across the full daily-to-monthly time-series at SPOT (53 samples over six months), pairwise Spearman correlation analyses showed that in the Protistan size fraction Comchoano relative abundance correlates with choanoflagellates (p and $q < 0.05$), with a higher percentage of correlations to choanoflagellates than to other eukaryotes apart from telonemids (Extended Data Fig. 2a,b). However, in the ‘Free-living

prokaryote fraction' (0.2 – 1 μm) of the daily time-resolved part of the time-series, Comchoano 16S rRNA amplicon relative amplicon abundances increased as *B. minor* 18S rRNA amplicon relative abundances decreased. Thus, a significant time-lagged positive correlation was observed between *B. minor* in the Protistan size fraction and Comchoano in the Free-living prokaryote fraction (2-day time-shifted $r= 0.62, 0.78$; $p=0.019, 0.0011$; Supplementary Data 1). These results suggest Comchoano were potentially released from lysed or dying *B. minor* cells (of course depending on the limitations of relative abundance-based analyses), and the time-delayed increase in Comchoano is consistent with a lytic parasitic or pathogenic role, akin to extensions of

Lotka-Volterra equations for host and parasite¹², or could arise by other phenomena whereby the choanoflagellate decreases in abundance while Comchoano are released into the environment.”

Line 149 – is manifest (instead of MANIFESTS)

Thank you for pointing this out and it has now been fixed.

Line 201 – To utilize the PPP... (instead of ‘To function the PPP)

Thank you for pointing this out and it has now been fixed.

Supplemental

Pg 4, line 11 – GC content (add content)

Added here and elsewhere.

Pg 9, line 11: suggest: Genomes with fewer than five of the ribosomal proteins (n=3) were removed [removed instead of moved]; however, one of these (CACFNL) was included in the 16S rRNA tree described below.

Edited as suggested, thank you.

Pg 20 – figure caption – ... and Comchoano-2, as is the case for several endosymbionts and pathogens, but which are encoded by free-living and particle-associated bacteria.

Edited as suggested, thank you.

Reviewer #2:

This manuscript reports on the genome sequence-based identification of a yet uncultured gammaproteobacterial lineage as putative symbionts of the choanoflagellate *Bicosta minor*. This is supported by the co-occurrence of bacteria and choanoflagellates sequences in published amplicon data-sets and after FACS-sorting of environmental samples, as well as by genome features characteristic of host-associated microbes.

We appreciate the reviewer summary and positive assessment of the evidence.

This is an exceptionally well-written manuscript I truly enjoyed reading. The comparative genome analysis is sound, and the evidence for the gammaproteobacteria being associated with *B. minor* is convincing. Unfortunately, the final proof, i.e. the identification and intracellular localization of the putative symbionts by fluorescence in situ hybridizations is lacking. This is particularly surprising as the sorting by FACS seems to work very efficiently as suggested by the data provided in Figure 1b.

We are grateful for the reviewer's positive comments on the manuscript and assessment of genome analysis; as well as acknowledgement that the association is convincing. Early recognition of the limitations to visualization and the inconclusive results FISH would likely provide (see below), led us to focus significant effort on making sure that we had a gapless, circularized genome assembly of *Comchoano* wherein gene absences (including membrane aspects etc. that would preclude a free-living lifestyle) could be confirmed alongside our technology-enabled (single-protist cell FACS) establishment of their co-association, and field studies.

Regarding FISH, the uncultivated host we are working with is very small which impacts the effectiveness of imaging to confirm results. Neither the host, nor the bacterium are cultured, or available in enrichment versions, precluding other types of highly informative microscopy (e.g., TEM, etc.). The small size of the single-celled host and the fact that it consumes bacteria would likely leave it unclear as to whether the bacterium was a food particle or endosymbiont. FISH visualization of established symbionts of (originally) uncultivated marine protists target photosynthetic protists that do not consume bacteria as prey, making the interpretation of FISH results from symbiont-targeted probes more conclusive. Also problematic are the statistics of *in situ* FISH studies since both members are uncultured, which make it challenging to capture robust relationships that would be above the signal to noise. This is particularly an issue for small marine protistan predators which are most often present at orders of magnitude lower abundance than e.g., eukaryotic algae, or many free-living marine bacterial taxa. Additionally, *Comchoano* was observed in 12% of hosts (based on our data) by single cell sorting, thus host

abundances would need to be much higher than they are for small predatory protists because the statistics (for observing the relationship) become yet another order of magnitude more difficult to achieve than just observing the hosts.

In order to orient the reader in terms of the types of evidence presented, we have now added a section that states:

“Visualization of the *Bicosta*-Comchoano relationship has not yet been achieved largely due to their uncultivated status and ephemeral abundances in their dynamic habitat where the likelihood of encountering the targeted interaction on the spatial and temporal

scales at which oceanographic research is conducted is low. Additionally, interpretation of bacterial roles in small uncultivated heterotrophic protists like choanoflagellates, that also contain bacterial prey, has limitations by epifluorescence microscopy and fluorescence in situ hybridization, especially combined with statistical challenges arising for uncultured entities that live at relatively low in abundance in the ocean and resulting factors like insufficient signal to noise/negative fluorescence ratios in diverse field samples that contain naturally autofluorescent particles^{4,5}. With respect to choanoflagellates and more resolved transmission electron microscopy (which is possible for abundant cells in culture), a possible endosymbiont has been noted in a cultured Baltic Sea species from brackish waters, however the putative host-microbe interaction, functional and phylogenetic features of the putative bacterium⁶ all remain unknown. Hence, in many regards, the completion and analysis of the Comchoano genomes and technology-enabled methods used herein for establishing their physical association with choanoflagellates provides the most compelling evidence yet possible for their relationship and its ramifications.”

As mentioned by the reviewer, and with the above issues in mind, one might still hope to enrich for choanoflagellates through cell sorting (as done with the primary population we encountered and single cell sequenced). Several factors challenge this approach: First, expeditions to sea generally do not capture the same communities/events or “blooms”, unless they are performed in highly repeatable environments such as open ocean gyres, or “boutique” environments (where the targeted organisms dominate a niche, making FISH more approachable). The Pacific region we worked in is an ecologically important and highly dynamic upwelling zone, making it very difficult to ‘resample’ the same community wherein such a population the choanoflagellate could be recovered and fixed for FISH. Second, several expeditions presented herein overlapped temporally with the long road to insights from single cell sorting (involving screening, metagenomic sequencing of individual wells, assembly, to perfection of the genome [to make sure gene absence was not an artifact of an incomplete genome], and establishment of the relationship through genomic analyses). Hence, although multiple cruises were performed and presented in this manuscript, the results from single cell sort through to complete genome and analyses extended longer than the funded related cruise work.

Third, the sorting protocol uses live staining (stains that only work in live cells) which means the protists pass through and exit the sorter into respective wells in an exceptionally small volume (a requirement for MDA), a disruption that can make fixation and visualization of intact cells difficult to achieve depending on the eukaryotic cell type and abundance (where loss of some percentage can be absorbed; unlike single cell sorting used for the genomes assembled herein). It is unclear whether these predatory protists would be amenable to fixation post-exiting the

instrument in small liquid volumes of sterile PBS, and testing is difficult because this choanoflagellate is not in culture. From the literature, FISH visualization of marine symbionts of protists in studies that did sort and fix, have largely focused on obligately photosynthetic algae that do not consume bacteria as prey, again making the FISH results of symbiont-targeted probes more conclusive. Finally, setting up for the sort and running all the controls at sea is difficult (operation at sea is extremely challenging as evidenced by the few labs that have published with such methods) and so the number of samples/depths that can be screened is low (a few distinct samples in a 24 hr period), reducing the possibility of finding the cells in highly dynamics areas.Note the lab has experience with FISH on predatory protists⁴ and we did develop probes tested with clone-FISH (wherein three *E. coli* mutant strains expressing the 18S and 16S rRNA gene of the respective uncultured entities are used for testing probe efficacy) and could include it in the supplement if desired.

The discovery of bacterial symbionts in choanoflagellates in this study is undoubtedly very interesting; the study identifies a yet unknown clade of bacteria as putative symbionts, and choanoflagellates are an attractive model systems from different points of view. The ecosystem-level implications of the study as summarized in the abstract and the last paragraph of the results/discussion section, however, appear overstated as microbial symbionts in heterotrophic marine protists have been described before.

We appreciate the reviewer's comments on how interesting this new bacterial lineage is and for pointing out that choanoflagellates are important model systems. We are not sure which publications the reviewer is referring to for microbial symbionts in bacteriovorous marine protists that capture aspects of how that relationship manifests, especially from the open ocean water column. We are also not aware of other studies of uncultivated predatory marine protists with co-associated bacteria that have reached this level of genome completion wherein the true nature of the bacterial genome can be examined. Additionally, such complete genomes from MAGs and FACS-SAGs are so far very rare. For FACS-SAGs, we are unaware of any such examples, and the average genome recovery from marine SAGs is $38 \pm 28\%$ SD completeness based on single copy core gene estimates ($n=12,715$). For MAGs, the list is few with a recent review outlining the challenges and benefits of closing MAGs reporting knowledge of 59 total¹³.

To respond to this reviewer's comment, in addition to other sections that e.g. discuss genome sequencing to various levels of completion performed for bacteria residing within some larger manually picked ciliates (~25 microns in size) from freshwater environments (see e.g. Graf et al., 2021⁸), We have now added more discussion of reports we are aware of as covered in a recent review (wherein the majority of examples come from freshwater, but covers those known from marine environments). The main text now states:

“Examples of microbial endosymbionts for which genomic information is available abound for heterotrophic protists belonging to other (non-Opisthokonta) eukaryotic supergroups, especially taxa residing in freshwater, soil, sediment, and host-associated (e.g., protists residing in termite guts) environments¹⁴. Fewer are known from marine habitats apart from sediments where, for example, symbionts with foraminiferans, excavates, and ciliates have been reported^{15–18}. Those reported from seawater appear to generally come from ciliates and diplomonads isolated from coastal environments¹⁹ and saltwater aquaria²⁰. The best-known examples of microbial symbioses in the pelagic

ocean between protists and endosymbiotic bacteria generally involve eukaryotic algal species and e.g., nitrogen-fixing cyanobacteria providing organic nitrogen to eukaryotic phytoplankton in exchange for carbon resources^{21,22}.”

We have also modified the abstract so that it now more specifically points to the choanoflagellate-Comchoano relationship:

“Our findings highlight interactions between uncultivated bacteria and choanoflagellates

that reshape the canonical direction of energy flow and resource transfer attributed to these widespread microbial predators, adding further complexity to core marine food webs.”

We also modified the conclusions by specifically calling out the recent discoveries on symbionts of protists, particularly the increasing number from freshwater and terrestrial ecosystems, some of which also include genome sequences for the symbiont. We also modified the last paragraph of the conclusions, with the following text:

“Recently, obligate bacterial associates of heterotrophic protists have been increasingly noted and genome sequenced from freshwater and terrestrial habitats¹⁴, however the vast majority of bacteria sequenced or cultured to date from the ocean maintain pathways for energy conversions and biosynthesis of essential compounds important to survival as free-living cells in the often resource deplete marine environment^{23,24}.”

“The widespread distribution of both the bacterivorous host and bacterial symbiont discovered herein, as well as the diversity of potentially host-associated uncultivated bacteria related to Comchoano, call for intensified efforts to identify cryptic symbioses and deeper knowledge of the strength and directionality of their influence on resource flow in the ocean.”

1. Is the name Comchoano for the two B. minor-associated bacteria intended to represent a new *Candidatus* name? If that was the case, the addition of a formal ‘*Candidatus*’ description in the supplement would be useful.

Per recommendations of the reviewers, we have now added a formal description – and yes a new *Candidatus* name/description. In the main text (there is also a longer supplemental section) the addition states:

“Our findings to this point call for the naming of what has thus far been an enigmatic order of uncultivated bacteria with little known about its ocean roles or lifestyle. We propose the following status: *Candidatus* Comchoanobacterales ord. nov., *Candidatus* Comchoanobacteraceae fam. nov. for what has been known to date as the UBA7916 order and family, respectively, following the protocols of order and family naming after type species (see below; and protologue in Supplementary Information). This status is proposed due to Comchoano and related UBA7916 members being phylogenetically distinct from other Gammaproteobacteria orders (Fig. 2a, Extended Data Fig. 7a,b) and their distance based on RED and AAI metrics. For the type species, i.e., Comchoano-1,

24we propose the status *Candidatus Comchoanobacter bicostacola* gen. et sp. nov. and for Comchoano-2, *Candidatus Symchoanobacter obligatus* gen. et sp. nov.”

2. Are Comchoano 1 and 2 separate species or even genera based on ANI and AAI analyses?

AAI between Comchoano 1 and 2 is 49% based on comparison of the 729 homologous proteins. With this low similarity, the ANI may be unreliable²⁵; OrthoANIu suggests an ANI of 67.03% based on 159,905 bp on average, which is about 15% of the genome size. The AAI

similarity is roughly similar to the amount expected from differences at the class level²⁶ yet the difference between the 16S rRNA gene sequences are 95%, which is similar to genus-level difference.

To circumvent such a problem of rapid evolutionary divergence in, e.g., symbionts, Parks et al. 2018²⁷, recommend the Relative Evolutionary Distance (RED; also recommended by Reviewer 3, below;²⁷), a metric that takes this into account. These values are both 0.83 for Comchoano, which is closer to the median distance between families (77%) than genus (93%) in GTDB, indicating they likely represent different (novel) genera, and likely different (novel) families.

“The phylogenomic reconstruction provided robust statistical support for Comchoano placement within the UBA7916 order (Extended Data Fig. 7a,b), as first indicated by full-length 16S rRNA gene analysis (which also included environmental 16S rRNA sequences for which no genomic information is available; Fig. 2a). The relative evolutionary distance (RED) value²⁷ between Comchoano and existing UBA7916 genomes (0.83) suggests Comchoano represent a novel family level of divergence within UBA7916. The genomes do retain some large scale syntenic patterns (Extended Data Fig. 7c) and the amino acid identity between Comchoano-1 and Comchoano-2 is 49% based on comparison of homologous proteins. Together, these results suggests that despite their close relationship to each other relative to other sequenced bacteria (Fig. 2a, Extended data Fig. 7), the two Comchoanos are at least separate genera, a conclusion supported by the 16S rRNA gene nucleotide sequence identity level (95%).”

3. Fig 1b: *B. minor* cells were apparently also associated with Rickettsiales ASVs. As rickettsiae are known intracellular microbes, it would be interesting to know more about these ASVs.

We appreciate the reviewer’s keen eye and interest in our study. Given the lower frequency of the other associations, including to Rickettsiales, we have not fully examined the genomic characteristics of these other associations. However, given the interesting attributes of the Rickettsiales (e.g., their propensity to be host-associated) and quite unique challenges (confusion with mitochondria, phylogenetic studies challenged by compositional biases, etc), we may attempt a follow-up study. We have edited the Fig. 1 legend to include more information on the % of cells and identity for each of the other groups, so that it now states:

“Inset, choanoflagellates sorted were almost exclusively *B. minor* (family Salpingoecidae). Here, “*B. minor* ASV 1” (100% nucleotide identity to uncultivated *B. minor*) refers to the dominant *B. minor* ASV (90% of all choanoflagellate cells), while the others are less frequently observed choanoflagellate ASVs. Inset, the most common

bacteria detected with the dominant *B. minor* ASV (only bacterial ASVs present with more than two *B. minor* cells are shown) comprised two Gammaproteobacteria (Comchoano-1 and Comchoano-2) and a third, less common Gammaproteobacterium (86% 16S rRNA amplicon identity to Comchoano with 2% of cells), alongside a Planctomycete (*Blastopirellula*, 93% identity to *Mariniblastus fucicola* with 3% of cells), two Flavobacteria (*Saprospiraceae* and *Lewinella* 93% and 91% identity to closest relatives *Membranicola marinus* and *Portibacter lacus*, respectively, with 2% and 1% of

cells), and two Rickettsiales (87% identity to an endosymbiont of *Oligobranchia haakonmosbiensis* with 2% and 1% of cells, respectively).”.

Additionally, the ASV sequences have been provided in FigShare alongside trees and alignments (DOI will become public upon publication of manuscript [10.6084/m9.figshare.c.5850662](https://doi.org/10.6084/m9.figshare.c.5850662); private link available here: <https://figshare.com/s/52ea341d6e67403ece52>), and read data is available via SRA accession PRJNA640955, and all 239 individual SRA samples, with metadata, are listed in Supplementary Data 12, which will be release upon release of our manuscript. Reviewer private links are available: <https://dataview.ncbi.nlm.nih.gov/object/PRJNA640955?reviewer=c64rrml2ee5o1hom1shc0heto> p.

4. L. 183: Members of the Coxiellales, the Berkiellales, and Legionellales include prominent symbionts and pathogens of protists. This should be mentioned here.

We thank the reviewer for bringing this up, and we now state:

“While lifestyle attributes or associations of UBA7916 members apart from Comchoano remain unknown, the Coxiellales, which again branched as the closest cultivated relatives, and other related orders, including the Berkiellales, Diplorickettsiales, and Legionellales (Extended Data Fig. 7a,b), are noted pathogens of insects and terrestrial mammals, as well as amoebozoan and ciliate protists^{14,28,29} unrelated to choanoflagellates.”

5. L. 192: I understand that genes for retinal synthesis are absent in the Comchoano genomes. If so, I suggest mentioning this in the main text as this renders the function as a proton pump unlikely.

We appreciate this observation from the reviewer and had erroneously deleted a sentence in the submitted version that dealt with this – it is now specifically addressed in the manuscript. Some aquatic, rhodopsin-encoding bacteria lack the (apparent) ability to produce retinal, including the abundant Gammaproteobacterium SAR86³⁰, this is also the case for archaea (Marine Group II³¹); additionally other bacteria have non-traditional or unknown ways to make retinal^{32,33}. Thus, Comchoano are not especially unique in this trait of not making retinal.

Perhaps more relevantly, it has been shown recently that a choanoflagellate has a functional rhodopsin (using it to alter its feeding behavior in the light vs dark) despite not encoding retinal.

Rather, the choanoflagellate scavenges retinal from its food³⁴. Thus, Comchoano could conceivably scavenge retinal (from host predation of retinal-containing marine microbes).

The text now states:

“These contain motifs suggestive of a proton pumping function with various hypothesized roles, including energy transfer or acidification of host-cellular compartments^{35–37}. Comchoano do not encode genes for biosynthesis of retinal, the chromophore required for rhodopsin function (Fig. 3d). Thus, Comchoano would need to scavenge the chromophore from prey ingested by the host – akin to the chromophore

scavenging that has been demonstrated for the cultured choanoflagellate species *Choanoeca flexa*³⁴, or potentially Comchoano produce retinal in an uncharacterized manner.”

Metabolic pathway analyses and Figure 3d: The selection of species for comparative analysis of the completeness of metabolic pathways includes only a single protist-associated symbiont. Would it be helpful to include other known (obligate) intracellular symbionts of protists in this analysis?

Yes, we thank the reviewer for this suggestion. We have added several obligate intracellular symbionts of protists in the analysis, namely: *Parachlamydia acanthamoebae* (Chlamydiales), *Protochlamydia amoebophila*, *Ca. Nesciobacter abundans* (Alphaproteobacteria), *Amoebophilus asiaticus*, (Bacteroidetes), *Ca. Azoamicus ciliticola* (Gammaproteobacteria), and *Berkiella aquae* (Gammaproteobacteria).

We also added a column to Figure 3d to indicate which taxa have known protistan hosts (13 total).

I am not sure whether the ordering of species according to their genome size in Fig 3d is very helpful. Perhaps a clustering based on the completeness information would be more intuitive to show similarities among species with respect to their metabolic potential.

Thank you for this suggestion (also from Reviewer 3). We have now performed a clustering, as described in the Figure 3d Legend:

“Genomes are clustered by the scaled values of the metabolic pathways as shown in the heatmap (Manhattan distance and complete linkage clustering), with bootstrap analysis (n=1000) performed with pvclust. Clustering is for visualization purposes, and only corresponds to the pathways displayed and thus may change if additional pathways were considered.”

7. Can the authors confirm that the identified ATP/ADP translocase belongs to the subfamily of these nucleotide transporters that is specific for ATP/ADP antiport?

We thank the reviewer for this question. We have further reviewed the literature and phylogenetics of this protein. Our newly added phylogenetic reconstruction (Extended Data Figure 11) places the Comchoano protein in the same regions of the tree as the ATP/ADP

30translocases in this gene family and Comchoano retain all the motifs supposedly controlling specificity and affinity for ATP/ADP in *A. thaliana* (we also now show this alignment with these positions highlighted). In any case, we, like with others, such as Graf et al., Nature 2021, cannot conclusively determine specificity for ATP/ADP without biochemical characterization.

In the manuscript we now state:

“The ATP/ADP translocase encoded by Comchoano and most UBA7916 (Fig. 3d, Extended Data Fig. 8) is phylogenetically similar to those reported in other ‘energy parasites’ (Extended Data Fig. 12a,b)⁹. Some translocases within this broad protein

family have been reported to have affinity for nucleotides such as guanosine di/tri-phosphate, calling for further experimental studies of translocase affinities^{8,10}. Recently, this translocase was implicated as being the mechanism by which *Ca. Azoamicus* endosymbionts provide ATP to their freshwater ciliate hosts. Here, we find the closest affiliation of the Comchoano translocases being with an ATP/ADP translocase in an alphaproteobacterial endosymbiont (Extended Data Figure 12b), with both maintaining all the motifs for ATP/ADP translocation that have been demonstrated as essential based on site-specific mutations (Extended Data Fig. 13) in versions present in parasitic fungi¹¹.”

8. Host-relationship of Comchoano symbionts is discussed exclusively in the context of microbial symbionts of animals (ll. 242-259). I suggest including findings from other protists-associated symbionts as well. Their genomes can be enriched in proteins with eukaryotic domains, exemplified by the amoeba symbiont *Amoebophilus asiaticus* (PMID 20023027).

We thank the reviewer for this comment, and have addressed this specifically now in the manuscript, partially due to the additional comment in point “6” above. We added six new protist-associated bacteria to our metabolic comparison, including *Amoebophilus asiaticus*, as suggested here.

“The identified gene repertoire in Comchoano is generally present in other UBA7916 as well, often in similarly high numbers as in Comchoano (Extended Data Fig. 8, Supplementary Data 10), and numerous protist-associated endosymbionts (Fig. 3d), as exemplified by the amoeba symbiont *Amoebophilus asiaticus*³⁸.”

We also now state:

“Some translocases within this broad protein family have been reported to have affinity for nucleotides such as guanosine di/tri-phosphate, calling for further experimental studies of translocase affinities^{8,10}. Recently, this translocase was implicated as being the mechanism by which *Ca. Azoamicus* endosymbionts provide ATP to their freshwater ciliate hosts. Here, we find the closest affiliation of the Comchoano translocases being with an ATP/ADP translocase in an alphaproteobacterial endosymbiont (Extended Data Figure 12b), with both maintaining all the motifs for ATP/ADP translocation that have been demonstrated as essential based on site-specific mutations (Extended Data Fig. 13) in versions present in parasitic fungi¹¹.”

9. Figure 4b is not essential and could be moved to the supplement.

32In the current version we have included this figure in the main text. We felt it is essential for establishing the protein/taxonomic sampling that supported the construction and validity of the subtree, since pT4SSi are a particular sub-type of T4SS, with a specific function, but are also quite diverse. To increase the value of this phylogenetic analysis we have made sure that symbols for bootstrap support are more visible, and hope that this helps.10. L 287. Typo: Bdellovibrio-like, Coxiellales

Thank you. Fixed.

11. Data availability: Genbank accession numbers and DOI to related files are missing.

We now provide Genbank accessions for 239 SRA samples, Comchoano genomes, and 16S rRNA genes via NCBI:

“Data availability Single and multi-cell sort raw data (short read archives, SRA) are available via NCBI Project Number PRJNA640955, which includes V4 16S rRNA gene amplicon sequences from single cells, whole genome shotgun sequences from single cells, MBTS 16S and 18S V4 rRNA gene amplicons, and MBTS V4-V5 rRNA gene amplicons (see Supplementary Data 12 for individual list of SRA accessions). 18S V4 rRNA gene amplicons are available as part of Needham et al. 2019³⁶. Comchoano-1 and Comchoano-2 whole genome sequences are available via accessions CP092900 and JAKUDN000000000 and their full-length 16S rRNA gene sequences are deposited as OM801198 and OM801197. Alignments, tree files, and processed amplicon data are available via FigShare (doi: 10.6084/m9.figshare.c.5850662).”

We additionally make alignments, trees and ASV data available via FigShare.

All data will be released upon publication; reviewer links are available at this time:

<https://dataview.ncbi.nlm.nih.gov/object/PRJNA640955?reviewer=c64rrml2ee5o1hom1shc0hetop> and <https://figshare.com/s/52ea341d6e67403ece52>.

Reviewer #3:

Needham and colleagues report the discovery of a novel bacterial lineage (“Comchoano”) inferred to be physically associated with the choanoflagellate *Bicosta minor*, most likely as endosymbionts. Choanoflagellates typically graze on bacteria and a metabolic reconstruction of genomes representing two Comchoano species recovered directly from single choanoflagellate cells or population sorts show several features consistent with host dependence and evasion of the hosts digestive processes.

Overall, this is a carefully conducted study with numerous lines of mostly circumstantial evidence supporting the conclusion that Comchoano are endosymbionts of *B. minor*. However, the most compelling evidence is missing: direct visualisation of Comchoano in their hosts using specific FISH probes. Did the authors attempt this? Are there sound reasons not to include visualisation evidence? This should at least be discussed briefly somewhere in the ms as visualisation of the lineage is conspicuous by its absence, and ideally FISH images should be part of the study.

We appreciate the reviewer’s positive evaluation and assessment of our numerous lines of evidence and bringing up whether imagery would provide compelling evidence. We feel that the state-of-the-art single cell sorting of live choanoflagellates (and their Comchoano associates), along with the genomic features of Comchoano (with a finished genome) and co-occurrence patterns in the natural environment are strong evidence. FISH by itself would likely be no more ‘circumstantial’ than these, as the choanoflagellates are heterotrophic and their small size likely would render it very difficult to determine if a bacterial cell is prey or an endosymbiont. In the revised manuscript we now explain why imagery is not presented by stating:

“Visualization of the *Bicosta*-Comchoano relationship has not yet been achieved largely due to their uncultivated status and ephemeral abundances in their dynamic habitat where the likelihood of reencountering the targeted interaction on the spatial and temporal scales at which oceanographic research is conducted is low. Additionally, interpretation of bacterial roles in small uncultivated heterotrophic protists like choanoflagellates, that also contain bacterial prey, has limitations by epifluorescence microscopy and fluorescence in situ hybridization, especially combined with statistical challenges arising for uncultured entities that live at relatively low in abundance in the ocean and resulting factors like insufficient signal to noise/negative fluorescence ratios in diverse field samples that contain naturally autofluorescent particles^{4,5}. With respect to choanoflagellates and more resolved transmission electron microscopy (which is possible for abundant cells in culture), a possible endosymbiont has been noted in a cultured Baltic Sea species from brackish waters, however the putative host-microbe interaction,

35

functional and phylogenetic features of the putative bacterium⁶ all remain unknown. Hence, in many regards, the completion and analysis of the Comchoano genomes and technology-enabled methods used herein for establishing their physical association with choanoflagellates provides the most compelling evidence yet possible for their relationship and its ramifications.”

Putting aside the limitations to interpretation, the ephemeral abundances of *B. minor* in their open ocean habitat made follow-up targeted studies difficult. Predatory protists are much lower

in abundance than photosynthetic protists, and because Comchoano are only associated with ~12% of host cells (as shown in our manuscript), this makes the challenge an order of magnitude more difficult to achieve robust statistics from FISH. It should be noted that successful FISH establishing endosymbionts in pelagic marine protists using epifluorescence microscopy has been performed on photosynthetic algae, where there is no confusion about possible presence of prey. Additionally, as a somewhat analogous approach, virusFISH (FISH, with host and virus labelled) was achieved on uncultivated host and virus, but from an environment (biofilm) where the host and virus are reliably the most dominant taxon³⁹. However, in our case, the location that the *B. minor*-Comchoano relationship occurs in a highly-productive upwelling, but also highly dynamic environment. One might imagine that it is possible to enrich the *B. minor* and Comchoano by single cell sorting, but the need to stain live cells at sea for sorting (whereas FISH can be performed on fixed samples) limits the possibilities for screening large numbers of samples. We did however attempt to do so upon multiple related cruises during the intervening time that we were performing the single cell screening, whole genome sequencing, and genome completion analyses reported herein. However, we did not encounter such an enriched *B. minor* population in the period of the funded-related field work after we had made the initial discovery. The lab does have experience with FISH on small heterotrophic protists⁴ and we did prepare and evaluate a FISH approach with specific probes that worked based on clone-FISH (via a hybridization chain reaction FISH), whereby we generated *E. coli* mutants with the 18S and 16S of *B. minor* and each Comchoano, respectively. Using clone-FISH we successfully evaluated the specificity of the probes based on cross-hybridization between *B. minor* and other choanoflagellate cloned 18S sequences, as well as for Comchoano. These probes can be provided as part of our manuscript, if the reviewer feels that would be helpful (making this protocol open access to allow a 'jumping off point' for future studies).

Specific comments

Line 33. I recommend proposing Candidatus genus and species names for Comchoano-1 and Comchoano-2 which are fine as unique genome identifiers, but not adequate for nomenclature. The names could still easily be based on Comchoano, for example, Candidatus Comchoanobacter bicostavorans or Ca. *C. intracellularis*. If you feel particularly motivated, you could even propose higher rank names based on the genus name, e.g. family Comchoanobacteraceae, and order Comchoanobacterales since there are no existing Latin names in this taxonomic neighborhood. I also highly recommend specifying type genomes for Comchoano-1 and -2 as described by Chuvochina et al, 2019:

<https://www.sciencedirect.com/science/article/pii/S0723202018302005>

See also comment below about Extended Data Fig. 6

We thank the reviewer for this comment which was also requested by Reviewer 2. At the

37reviewer's suggestion, we conservatively define the two Comchoano as different genera, and give them the names *Candidatus Comchoanobacter bicostacola* gen. et sp. nov. and *Symchoanobacter obligatus* gen. et sp. nov. We believe this is warranted due to the amino acid identity, their relative evolutionary distance (RED, see this reviewer's last comment, below), and rRNA gene similarity. We designate each of these genomes as the type-strains of their genus. The genomes are perhaps novel enough to be potentially be distinct families based on RED, but probably not 16S rRNA gene similarity (perhaps due to fast evolution within protein coding genes of obligate associates), so therefore we maintain them all within a single family, and since

the current family name is an alphanumeric placeholder, we propose to name it after *Candidatus Comchoanobacter bicostacola* (designated to be the type-genus of the order since it is the most high-quality genome, in a single circular contig), Comchoanobacteraceae, in accordance with type strain conventions. Finally, due to the order in which Comchoano are affiliated, also only being known as an alphanumeric placeholder, in accordance with conventions of nomenclature, we propose re-naming the order, Comchoanobacterales.

In the manuscript we have added discussion of the RED value etc., so that it now states:

“The relative evolutionary distance (RED) value²⁷ between Comchoano and existing UBA7916 genomes (0.83) suggests Comchoano represent a novel family level of divergence within UBA7916. The genomes do retain some large scale syntenic patterns (Extended Data Fig. 7c) and the amino acid identity between Comchoano-1 and Comchoano-2 is 49% based on comparison of homologous proteins. Together, these results suggests that despite their close relationship to each other relative to other sequenced bacteria (Fig. 2a, Extended data Fig. 7), the two Comchoanos are at least separate genera, a conclusion supported by the 16S rRNA gene nucleotide sequence identity level (95%).”

We also provide a protologue describing the proposed naming:

“Protologues for new *Candidatus* taxa identified from metagenomic analysis of *Bicosta minor* sorted single-cells.

Description of Candidatus Comchoanobacterales ord. nov.

Candidatus Comchoanobacterales (*Com.choa.no.bac.te.ra'les*. N.L. masc. n. *Comchoanobacter* type genus of the order; N.L. suff. *-ales* to denote an order; N.L. fem. pl. n. *Comchoanobacterales*, the *Comchoanobacter* order).

A bacterial order identified by metagenomic analyses from bulk seawater samples and single-cell sorted cells collected from various marine sites. This is the name for the alphanumeric GTDB order UBA7916²⁷. This order has been assigned by GTDB working on GTDB Release 89^{27,40} and by the phylogenomic tree displayed in this study (Extended Data Fig. 7) to the class

39Gammaproteobacteria.

Description of *Candidatus* *Comchoanobacteraceae* fam. nov.

Candidatus Comchoanobacteraceae (Com.choa.no.bac.te.ra.ce'ae. N.L. masc. n.

Comchoanobacter type genus of the family; N.L. suff. *-aceae* to denote a family; N.L. fem. pl.

n. *Comchoanobacteraceae*, the *Comchoanobacter* family).

A bacterial family identified by metagenomic analyses (bulk seawater sampling and single-cell sorting). This is a name for the alphanumeric GTDB family UBA1515^{27,40}. The family is assigned to the order Comchoanobacterales.

Description of *Candidatus Comchoanobacter* gen. nov.

Candidatus Comchoanobacter (Com.choa.no.bac'ter. L. pref. *Cum-*, with; Gr. fem. n. *khoánē*, funnel; N.L. masc. n. *bacter*, staff; N.L. masc. n. *Comchoanobacter* a bacterium associated with a choanoflagellate).

A bacterial genus identified by single-cell sorted metagenomic analyses. The genus includes all bacteria with genomes that show $\geq 60\%$ average amino acid identity (AAI) to the type genome from the type species *Candidatus Comchoanobacter bicostacola*. This genus is assigned to the order Comchoanobacterales and to the family Comchoanobacteraceae.

Description of *Candidatus Comchoanobacter bicostacola* sp. nov.

Candidatus Comchoanobacter bicostacola (bi.cos.ta'co.la. N.L. fem. n. *bicosta* the choanoflagellate genus *Bicosta*; L. suff. *-cola* inhabitant of; N.L. n. *bicostacola* an inhabitant of *Bicosta*).

A bacterial species identified by single-cell sorted metagenomic analyses. This species includes all bacteria with genomes that show $\geq 95\%$ average nucleotide identity (ANI) to the type genome, which has been assigned the SAG ID Comchoano-1 and which is available via NCBI accession CP092900. The Comchoano-1 16S rRNA gene sequence is available via NCBI accession OM801198. The relative evolutionary distance (RED) is 0.83 to the closest relatives in GTDB release 95, as well as after re-calculation of all Comchoanobacterales identified in the present study. The phylogenetic position within the Comchoanobacterales order and the Comchoanobacteraceae family has been analyzed using both the reference genome (Extended Data Fig. 7b) and the 16S rRNA gene sequence (Fig. 2a). The GC content of the type genome is 39.3% and the genome length is 1.01 Mbp. The genome is high-quality, in a single circular chromosome, with standard completion estimates (based on presence of single copy marker genes of Bacteria⁴¹) suggesting 96.55% completion and 0% contamination. However, this is likely an underestimate of the genome completion due to the single circular chromosome sequence for the genome, as well as the fact that no additional novelty was detected across multiple single cells. The type genome *Candidatus Comchoanobacter bicostacola* sp. nov. originated from the North Pacific Ocean, from 20 m water depth. The genomes have limitations metabolically, apparently unable to synthesis fatty acids, vitamins, amino acids, or perform glycolysis, while encoding for a Type IV secretion system, pentose phosphate pathway, oxidative phosphorylation, a rhodopsin, and a putative ATP/ADP translocase.

Description of *Candidatus Symchoanobacter* gen. nov.

Candidatus Symchoanobacter (Sym.choa.no.bac'ter. L. pref. *Sum-*, with; Gr. fem. n. *khoánē*, funnel; N.L. masc. n. *bacter*, staff; N.L. masc. n. *Symchoanobacter* a bacterium associated with a choanoflagellate).

A bacterial genus identified by single-cell sorted metagenomic analyses. The genus includes all bacteria with genomes that show $\geq 60\%$ average amino acid identity (AAI) to the type genome from the type species *Candidatus* Comchoanobacter bicostacola. This genus is assigned to the order Comchoanobacterales and to the family Comchoanobacteraceae.

Description of *Candidatus* Symchoanobacter obligatus sp. nov.

Candidatus Symchoanobacter obligatus (o.bli.ga'tus. L. pass. part. nom. n. *obligatus* bound by obligation).

A bacterial species identified by metagenomic analyses. This species includes all bacteria with genomes that show $\geq 95\%$ average nucleotide identity (ANI) to the type genome, which has been assigned the SAG ID Comchoano-2 and which is available via NCBI Accession JAKUDN000000000 (The version described in this paper is version JAKUDN010000000). The Comchoano-2 16S rRNA gene sequence is available via NCBI accession OM801197. The

relative evolutionary distance (RED) is 0.83 to the closest relatives in GTDB release 95, as well as after re-calculation of all Comchoanobacterales identified in the present study. The phylogenetic position within the Comchoanobacterales order and the Comchoanobacteraceae family has been analyzed using both the reference genome (Extended Data Fig. 7b) and the 16S rRNA gene sequence (Fig. 2a). The GC content of the type genome is 41.6% and the genome length is 1.07 Mbp. The genome is high-quality, in two contigs, with standard measures (based on presence of single copy marker genes of Bacteria⁴¹) suggesting 96.55% completion and 0% contamination. However, this is likely an underestimate of the genome completion due the fact that no additional novelty was detected across multiple single cells. The type genome *Candidatus* Symchoanobacter obligatus sp. nov. originated from the North Pacific Ocean, from 20 m water depth. The genomes have limitations metabolically, apparently unable to synthesis fatty acids, vitamins, amino acids, or perform glycolysis, while encoding for a Type IV secretion system, pentose phosphate pathway, oxidative phosphorylation, a rhodopsin, and a putative ATP/ADP translocase.

Line 111. Doesn't the time delay between host and endosymbiont population relative abundances suggest a lytic pathogen lifestyle more consistent with parasitism?

We agree that this time-delay suggests a lytic pathogen or parasitic lifestyle, but also may be the result of other indirect effects (since of course this part of the analysis is based on relative abundances). We discuss this in more detail now (als per request of another reviewer):

“We then turned to time-resolved data to gain insight into ecological relationships, specifically to data collected daily-to-weekly at SPOT, a time-series study in Pacific waters of the Southern California Bight (Fig. 1a). Across the daily portion of the time-series, Comchoano and *B. minor* relative amplicon abundances increased contemporaneously in the ‘Protistan size fraction’ (1 – 80 μm) ($r = 0.74, 0.77$; $p=0.0001, 0.00004$ for Comchoano-1 and 2, respectively; Fig. 2b, Supplementary Data 1). Moreover, across the full daily-to-monthly time-series at SPOT (53 samples over six months), pairwise Spearman correlation analyses showed that in the Protistan size fraction Comchoano relative abundance correlates with choanoflagellates (p and $q < 0.05$), with a higher percentage of correlations to choanoflagellates than to other eukaryotes apart from telonemids (Extended Data Fig. 2a,b). However, in the ‘Free-living prokaryote fraction’ (0.2 – 1 μm) of the daily time-resolved part of the time-series, Comchoano 16S rRNA amplicon relative amplicon abundances increased as *B. minor* 18S rRNA amplicon relative abundances decreased. Thus, a significant time-lagged positive correlation was observed between *B. minor* in the Protistan size fraction and Comchoano in the Free-living prokaryote fraction (2-day time-shifted $r = 0.62, 0.78$;

43$p=0.019, 0.0011$; Supplementary Data 1). These results suggest Comchoano were potentially released from lysed or dying *B. minor* cells (of course depending on the limitations of relative abundance-based analyses), and the time-delayed increase in Comchoano is consistent with a lytic parasitic or pathogenic role, akin to extensions of Lotka-Volterra equations for host and parasite¹², or could arise by other phenomena whereby the choanoflagellate decreases in abundance while Comchoano are released into the environment.”

Line 205. Some have questioned whether ATP/ADP translocases definitively indicate energy parasitism given the widespread distribution of nucleotide transport proteins in bacteria and other potential roles. See Major et al., 2017

<https://academic.oup.com/gbe/article/9/2/480/2970297>

We thank the reviewer for mentioning this paper and raising this question. We now performed a phylogenetic reconstruction that includes the Comchoano ATP/ADP translocases, using the sequences from Major et al, as well as GTDB as well as MAGs and SAGs from the ocean (Extended Data Fig. 12a,b). This analysis and review of the literature indeed establish that there may be other roles for ATP/ADP translocases reported to date, though the closest related sequence is an ATP/ADP translocase and Comchoano retain all the motifs supposedly controlling specificity and affinity for ATP/ADP in *A. thaliana* (we also show this alignment with these positions highlighted). The manuscript now states:

“Comchoano may meet energy demands using an ATP/ADP translocase it possesses which can directly import ATP from the host environment. This has been reported as the mechanism by which obligate endosymbiotic bacteria^{9,42} attain energy from the host, for example the obligate intracellular pathogen *Chromulinavorax destructans* found in a freshwater stramenopile protist⁴³. The ATP/ADP translocase encoded by Comchoano and most UBA7916 (Fig. 3d, Extended Data Fig. 8) is phylogenetically similar to those reported in other ‘energy parasites’ (Extended Data Fig. 12a,b)⁹. Some translocases within this broad protein family have been reported to have affinity for nucleotides such as guanosine di/tri-phosphate, calling for further experimental studies of translocase affinities^{8,10}. Recently, this translocase was implicated as being the mechanism by which *Ca. Azoamicus* endosymbionts provide ATP to their freshwater ciliate hosts. Here, we find the closest affiliation of the Comchoano translocases being with an ATP/ADP translocase in an alphaproteobacterial endosymbiont (Extended Data Figure 12b), with both maintaining all the motifs for ATP/ADP translocation that have been demonstrated as essential based on site-specific mutations (Extended Data Fig. 13) in versions present in parasitic fungi¹¹.”

Line 270. Can the authors infer where host association originated in the UBA7916 on a bacterial species tree? Could it predate the common ancestor of this order?

We appreciate the reviewer’s question about where host-association originated in UBA7916. It seems likely that many UBA7916 are host-associated, which would be a shared trait to other related lineages (Coxiellales, Legionellales, Berkiella). However, some UBA7916 (not

Comchoano) may be capable of growth outside a host, encoding, e.g., vitamin biosynthesis, fatty acids, and glycolysis, and fewer eukaryotic-like domains thought to influence hosts (see Figure 3d, Extended Data Fig. 8), similar to the related lineages (*Coxiella*, *Legionella*, *Berkiella*). Therefore, it seems that this apparent obligate host-association may be a more recent event.

“The presence, synteny, and phylogenetic conservation of the pT4SSi of Comchoano and distant relatives such as Coxiellales and Legionellales, implies that host-association is likely an ancient trait for these lineages broadly. However, the pT4SSi and other features

(e.g., eukaryotic-like domains, ATP/ADP translocase) that are conserved among Comchoano and close relatives, paired with variations in the extent of reduction in genome size and metabolic capacities, suggest that the obligate nature of association is a more recent and sporadic trait..”

Line 272. Are there known T4SS effector protein signalling domains in species with homologous systems indicating secretion using this system that could be searched for? Are there other features of the described pathogens carrying the T4SS (line 286) relating to their pathogenicity that could be searched for in Comchoano? Since *B. minor* has previously been observed to develop multicellularity or switch to sexual reproduction as a result of bacterial effector lipids and proteins, could the presence of the T4SS be related as the mating cycle would spread Comchoano between hosts. I'd like to see if the known effectors (eg. EroS) are present in the Comchoano genomes, as you speculate that Comchoano are being released from lysed host cells but don't suggest any mechanisms for transmission/infection. Were any other secretion signalling domains identified? Would be interesting to identify the repertoire of potentially secreted proteins to inform hypotheses related to host interaction.

Yes, Comchoano encode a large number of the most well-known signaling domains such as ankyrin repeats, and leucine rich repeats. These proteins, which symbionts often use to manipulate host proteins, are particularly enriched in Comchoano, many other UBA7916, and often host associated symbionts in general.

In regards to specific pathogenicity features related to T4SS in described pathogens, we can look to *Coxiella* and *Legionella* for examples which are the most closely-related cultivated relatives and are well-studied pathogens. Bioinformatic approaches to examine such factors in *Coxiella*, *Legionella*, and related taxa have revealed potentially thousands more (but with no known experimental validation)⁴⁴ While the Comchoano certainly have genes that are homologous to these effector proteins in *Coxiella* and *Legionella*, we feel that the distance, differences in host, and sheer number would render it too speculative at to go down the road of ascribing pathogenicity factors from *Coxiella* and *Legionella* to Comchoano.

In regards to EroS, we did immediately search the genome as soon as it was assembled but did not find homologous proteins in Comchoano or UBA7916 (e-value $<e^{-5}$); the best hits to EroS in Comchoano had much better hits to unrelated proteins, namely a Sell effector protein. We now state:

“We did not find a recently described polysaccharide lyase (EroS) shown to induce sexual reproduction in the cultivated choanoflagellate, *Salpingoeca rosetta*⁴⁵.”

Line 287. Note correct spelling of Bdellovibrio

Thanks for pointing out – it is now fixed.

Fig. 2. Are there any other euks identified in the SPOT data that could be correlated with Comchoano abundance? Data is a little variable but there's a suggestion of a spike in the free-living fraction ~day 18 without identified change in B. minor abundance. Conversely, would the

time delayed increase in Comchoano in the free-living fraction be consistent with a lytic pathogen lifestyle?

To address the first question, we have now analyzed the SPOT data from the full dataset for correlations to Comchoano-1 and Comchoano-2 abundance. Interestingly, this analysis revealed in the 1-80 μm size fraction (where protists and host/particle-associated bacteria are found), correlations to Choanoflagellates besides *Bicosta*, as well as numerous other protists. Because it is well-known that correlation-based analyses can yield false positives or false negatives⁴⁶, we assessed to what extent the correlations to the various eukaryotic lineages were enriched compared to the other taxa based on the total numbers of OTUs for a given lineage in the dataset. These analyses have now been provided in the manuscript, through 1.) a network analysis of the correlations of protists to Comchoano-1 and Comchoano-2 and 2.) the enrichment analyses (see below for full in-text response).

In regards to the other point of the reviewer, we assessed the correlations between Comchoano and eukaryotes in the free-living size fraction. Interestingly, there were no statistical correlations to Choanoflagellates in this size fraction when no time-lag was allowed. However, at the beginning of the time-series (where time-lags could be best investigated due to its extended daily sampling resolution) there was an association as the reviewer points out with a time-lag, where Comchoano increased ~ 2 -3 days after the decline of *Bicosta*. Unfortunately, this time-lag analysis could not be performed for the full time-series because daily resolution was not always performed after the first 18 days.

Finally, we agree with the reviewer that this is consistent with a lytic pathogen or parasitic lifestyle.

Addressing all the points above, we write:

“We then turned to time-resolved data to gain insight into ecological relationships, specifically to data collected daily-to-weekly at SPOT, a time-series study in Pacific waters of the Southern California Bight (Fig. 1a). Across the daily portion of the time-series, Comchoano and *B. minor* relative amplicon abundances increased contemporaneously in the ‘Protistan size fraction’ (1 – 80 μm) ($r = 0.74, 0.77$; $p=0.0001, 0.00004$ for Comchoano-1 and 2, respectively; Fig. 2b, Supplementary Data 1). Moreover, across the full daily-to-monthly time-series at SPOT (53 samples over six months), pairwise Spearman correlation analyses showed that in the Protistan size fraction Comchoano relative abundance correlates with choanoflagellates (p and $q < 0.05$), with a higher percentage of correlations to choanoflagellates than to other eukaryotes apart from telonemids (Extended Data Fig. 2a,b). However, in the ‘Free-living prokaryote fraction’ (0.2 – 1 μm) of the daily time-resolved part of the time-series,

Comchoano 16S rRNA amplicon relative abundances increased as *B. minor* 18S rRNA amplicon relative abundances decreased. Thus, a significant time-lagged positive correlation was observed between *B. minor* in the Protistan size fraction and Comchoano in the Free-living prokaryote fraction (2-day time-shifted $r = 0.62, 0.78$; $p = 0.019, 0.0011$; Supplementary Data 1). These results suggest Comchoano were potentially released from lysed or dying *B. minor* cells (of course depending on the limitations of relative abundance-based analyses), and the time-delayed increase in Comchoano is consistent with a lytic parasitic or pathogenic role, akin to extensions of Lotka-Volterra equations for host and parasite¹², or could arise by other phenomena whereby the choanoflagellate decreases in abundance while Comchoano are released into the environment.”

We also provide the correlation values as Supplementary Dataset 1.

Fig. 3. Panel b could be more informative if rows were allowed to cluster by gene content rather than the apparent sorting by genome size which complicates comparison between members of the UBA7916 lineage and between UBA7916 members and other species.

As also suggested by Reviewer 2, we have now performed clustering based on gene content.

Extended Data Fig. 6. Based on panel b, Comchoano-1 and -2 are at least separate species if not in separate genera. GTDB-Tk, should provide some indication of the degree of novelty for this based on ANI and RED values. What is the ANI between the two Comchoano genomes? How much synteny do they share? Also, where do the genomes CACOFC and others in the order UBA7916 come from? They don't have genome accession ids so were they also obtained in the present study? If not, please provide genome accessions as you do for other reference genomes in panel b.

We agree they are at least separate species, likely separate genera or families. As covered above, the AAI is 49% and the RED scores are 0.83, corresponding to more to the median distance between families than genera in GTDB, however we consider them, conservatively, to be different genera, but not different families (see above description for further explanation).

Despite the low ANI similarity (estimated 67% across 15% of the genome via OrthoANIu), we performed and included a synteny analysis with Mauve revealing numerous rearrangements (Extended Data Fig. 7). This analysis revealed large syntenic regions, while many other segments are not syntenic.

“The relative evolutionary distance (RED) value²⁷ between Comchoano and existing UBA7916 genomes (0.83) suggests Comchoano represent a novel family level of divergence within UBA7916. The genomes do retain some large scale syntenic patterns (Extended Data Fig. 7c) and the amino acid identity between Comchoano-1 and Comchoano-2 is 49% based on comparison of homologous proteins. Together, these results suggests that despite their close relationship to each other relative to other sequenced bacteria (Fig. 2a, Extended data Fig. 7), the two Comchoanos are at least separate genera, a conclusion supported by the 16S rRNA gene nucleotide sequence identity level (95%).”

The CACOCF are from another single cell genome paper of free-living bacteria, and these labels are the assembly identifications at NCBI. We add “SAG” to the respective labels, and add to the figure:

Legend: “Labels with “NW Atl SAG” are from a study from a planktonic single cell genomics study from the Northwest Atlantic⁴⁷

Decision Letter, first revision:

Dear Dr. Worden,

I hope this email finds you well. This is {redacted}, a senior editor at Nature Microbiology who has taken over as handling editor of your manuscript, "Single marine bacterivore microbiomes reveal members of a diverse uncultivated bacterial lineage are resource demanding obligate associates" (NMICROBIOL-21092289A). Your work has now been seen by two of the original referees and their comments are below (Reviewer #1 was no longer able to supply a report this round). The reviewers find that the paper has improved in revision, and therefore we'll be happy in principle to publish it in Nature Microbiology, pending minor revisions to satisfy the referees' final requests and to comply with our editorial and formatting guidelines.

Thank you again for your interest in Nature Microbiology Please do not hesitate to contact me if you have any questions.

2Sincerely,

{redacted}

Reviewer #2 (Remarks to the Author):

Thank you very much for the thoughtful response to my earlier questions and the additional analyses.

I have no further comments in light of the revised manuscript other than a brief remark regarding the section on the visualization of the Bicosta-Comchoano relationship: This new paragraph is rather generic and reads like FISH would generally not be possible in these dynamic environments and would be an unsuitable method for small uncultivated heterotrophic protists. I tend to disagree with this statement but found the reasons for the absence of FISH data in the present study and provided in the rebuttal letter more convincing, i.e. the challenges with the timing of sampling and pre-sorting by FACS. Including these reasons would help the reader to understand how these aspects could be considered in an experimental design that includes FISH analysis (which still and for good reasons is the gold standard for the identification of uncultured microbe).

Matthias Horn

Reviewer #3 (Remarks to the Author):

I am satisfied with the authors responses to my comments and the associated changes/ additions to the ms. I apologise for my original use of the term "circumstantial", I really meant "indirect" specifically in relation to visualisation as per the old adage "seeing is believing". However, I agree with the authors that the cumulative evidence is strong and appreciate the authors efforts to attempt FISH albeit unsuccessfully. It probably would not hurt to include the probe sequences in the supplementary material in case others want to try their luck at visualisation.

I thank the authors for going to the trouble of giving Candidatus names to their bacteria. According to my colleague with expertise in etymology, the names are fine with the following feedback: i) *C. bicostacola* should either be *C. bicostae* as a genitive form of the generic name or *C. bicosticola*, and ii) *Symchoanobacter* should be corrected to *Synchoanobacter*.

Decision Letter, final checks:

3Dear Alex,

Thank you for your patience as we've prepared the guidelines for final submission of your Nature Microbiology manuscript, "Single marine bacterivore microbiomes reveal members of a diverse uncultivated bacterial lineage are resource demanding obligate associates" (NMICROBIOL-21092289A). Please carefully follow the step-by-step instructions provided in the attached file, and add a response in each row of the table to indicate the changes that you have made. Please also check and comment on any additional marked-up edits we have proposed within the text. Ensuring that each point is addressed will help to ensure that your revised manuscript can be swiftly handed over to our production team.

In recognition of the time and expertise our reviewers provide to Nature Microbiology's editorial process, we would like to formally acknowledge their contribution to the external peer review of your manuscript entitled "Single marine bacterivore microbiomes reveal members of a diverse uncultivated bacterial lineage are resource demanding obligate associates". For those reviewers who give their assent, we will be publishing their names alongside the published article.

Nature Microbiology offers a Transparent Peer Review option for new original research manuscripts submitted after December 1st, 2019. As part of this initiative, we encourage our authors to support increased transparency into the peer review process by agreeing to have the reviewer comments, author rebuttal letters, and editorial decision letters published as a Supplementary item. When you submit your final files please clearly state in your cover letter whether or not you would like to participate in this initiative. Please note that failure to state your preference will result in delays in accepting your manuscript for publication.

Cover suggestions

As you prepare your final files we encourage you to consider whether you have any images or illustrations that may be appropriate for use on the cover of Nature Microbiology.

4We accept TIFF, JPEG, PNG or PSD file formats (a layered PSD file would be ideal), and the image should be at least 300ppi resolution (preferably 600-1200 ppi), in CMYK colour mode.

Nature Microbiology has now transitioned to a unified Rights Collection system which will allow our Author Services team to quickly and easily collect the rights and permissions required to publish your work. Approximately 10 days after your paper is formally accepted, you will receive an email in providing you with a link to complete the grant of rights. If your paper is eligible for Open Access, our Author Services team will also be in touch regarding any additional information that may be required to arrange payment for your article.

Please note that *Nature Microbiology* is a Transformative Journal (TJ). Authors may publish their research with us through the traditional subscription access route or make their paper immediately open access through payment of an article-processing charge (APC). Authors will not be required to make a final decision about access to their article until it has been accepted. [Find out more about Transformative Journals](https://www.springernature.com/gp/open-research/transformative-journals)

Authors may need to take specific actions to achieve [compliance with funder and institutional open access mandates](https://www.springernature.com/gp/open-research/funding/policy-compliance-faqs). If your research is supported by a funder that requires immediate open access (e.g. according to [Plan S principles](https://www.springernature.com/gp/open-research/plan-s-compliance)) then you should select the gold OA route, and we will direct you to the compliant route where possible. For authors selecting the subscription publication route, the journal's standard licensing terms will need to be accepted, including [self-archiving policies](https://www.nature.com/nature-portfolio/editorial-policies/self-archiving-and-license-to-publish). Those licensing terms will supersede any other terms that the author or any third party may assert apply to any version of the manuscript.

Please use the following link for uploading these materials:
{redacted}

Best regards,

{redacted}

Reviewer #2:

Remarks to the Author:

Thank you very much for the thoughtful response to my earlier questions and the additional analyses.

I have no further comments in light of the revised manuscript other than a brief remark regarding the section on the visualization of the Bicosta-Comchoano relationship: This new paragraph is rather generic and reads like FISH would generally not be possible in these dynamic environments and would be an unsuitable method for small uncultivated heterotrophic protists. I tend to disagree with this statement but found the reasons for the absence of FISH data in the present study and provided in the rebuttal letter more convincing, i.e. the challenges with the timing of sampling and pre-sorting by FACS. Including these reasons would help the reader to understand how these aspects could be considered in an experimental design that includes FISH analysis (which still and for good reasons is the gold standard for the identification of uncultured microbe).

Matthias Horn

Reviewer #3:

Remarks to the Author:

I am satisfied with the authors responses to my comments and the associated changes/ additions to the ms. I apologise for my original use of the term "circumstantial", I really meant "indirect" specifically in relation to visualisation as per the old adage "seeing is believing". However, I agree with the authors that the cumulative evidence is strong and appreciate the authors efforts to attempt FISH albeit unsuccessfully. It probably would not hurt to include the probe sequences in the supplementary material in case others want to try their luck at visualisation.

I thank the authors for going to the trouble of giving Candidatus names to their bacteria. According to my colleague with expertise in etymology, the names are fine with the following feedback: i) *C. bicostacola* should either be *C. bicostae* as a genitive form of the generic name or *C. bicosticola*, and ii) *Symchoanobacter* should be corrected to *Synchoanobacter*.

Decision Letter, second revision:

Reviewer #2:
Remarks to the Author:

Thank you very much for the thoughtful response to my earlier questions and the additional analyses.

We thank the reviewer for the reception of our revision.

I have no further comments in light of the revised manuscript other than a brief remark regarding the section on the visualization of the Bicosta-Comchoano relationship: This new paragraph is rather generic and reads like FISH would generally not be possible in these dynamic environments and would be an unsuitable method for small uncultivated heterotrophic protists. I tend to disagree with this statement but found the reasons for the absence of FISH data in the present study and provided in the rebuttal letter more convincing, i.e. the challenges with the timing of sampling and pre-sorting by FACS. Including these reasons would help the reader to understand how these aspects could be considered in an experimental design that includes FISH analysis (which still and for good reasons is the gold standard for the identification of uncultured microbe).

Matthias Horn

We thank Professor Dr. Horn for his consideration of our response in both the MS and rebuttal letter. In line with the concerns, we now amend our MS to be more open to the idea of FISH being appropriate for rare heterotrophic protists, but acknowledging its challenges.

“Visualization of the *Bicosta*-Comchoano relationship has not yet been achieved largely due to their uncultivated status and ephemeral abundances in their dynamic habitat where the likelihood of reencountering the targeted interaction on the spatial and temporal scales at which oceanographic research is conducted is low. This presents challenges and statistical limitations for e.g., efforts by fluorescence in situ hybridization (FISH, as attempted herein, see methods) to capture co-associations that again occur either ephemerally or in relatively low abundances, efforts that can also be hampered by insufficient signal to noise/negative fluorescence ratios in field samples (which also contain naturally autofluorescent particles)^{57,58}, or by difficulties in interpreting signals from small uncultivated heterotrophic protists like *B. minor*, which could also contain prey cells. Collectively, these issues render the genomic data collected in the context of unperturbed cell co-associations invaluable.”

Reviewer #3:

Remarks to the Author:

I am satisfied with the authors responses to my comments and the associated changes/additions to the ms. I apologise for my original use of the term “circumstantial”, I really meant “indirect” specifically in relation to visualisation as per the old adage “seeing is believing”. However, I agree with the authors that the cumulative evidence is strong and appreciate the authors efforts to attempt FISH albeit unsuccessfully. It probably would not hurt to include the probe sequences in the supplementary material in case others want to try their luck at visualisation.

We thank the reviewer for their positive assessment of our revision.

In line with their suggestion to add the probe sequences, as cited in the main MS (see above), we add a section to the methods as follows.

HCR-FISH probes

“To design probes for Comchoano-1, Comchoano-2, and *B. minor* for Hybridization Chain Reaction Fluorescence In Situ Hybridization (HCR-FISH)¹³⁴, 23 probe candidates targeting 16S rRNA for Comchoano-1 and Comchoano-2 and 17 probe candidates targeting 18S rRNA genes for *B. minor* were provided by Molecular Technologies. Subsequently, these sequences were searched against NCBI for to be specific only to their target (Comchoano-1, Comchoano-2, or *B. minor*, respectively), as well as across available sequences from closely-related organisms (for Comchoano-1 and Comchoano-2 that included against each other). Two top candidates were identified for each target based on their sequence specificity. For Comchoano-1: probe “2”, tcggaaaagtgatggcgagtgccggacgggtgagtaatgcgtaggaatcta and probe “5”, tgcgatgaaggcttcgggtcgtaaagcacttcagtggaagatggctta; The probes were as follows: for Comchoano-2: probe “2”, tcggaagaatgatggcgagtgccgaacgggtgagtaatgcgtaggaatcta and probe “14”, cttagtaataaagggtgccttcgggaaccgagatacaggtgttcgatggc; for *B. minor*: probe “5”, tgattctcgagtcttctcctcgtagttgttggcgacttgattgggtgcc and probe “v4-1”, tctgattcgaagatcggtccgccgaaggcgagcactgattctcgagtct. For each probe set, these sequences were converted into “even” and “odd” split-initiator probes in accordance with the HCR v3.0 protocol¹³⁴. Because the three cell types are currently not in culture, we used a Clone-FISH approach¹³⁵ in *E. coli* for positive and negative control testing of probes. Ultimately, both the evaluated *B. minor* and Comchoano-1 probes were deemed to be specific in a limited set of cross-

8reactivity tests, meaning fluorescent signal was not appreciable when the *B. minor* probe was paired with another choanoflagellate sequence and fluorescence was also not appreciable when Comchoano-1 probes were paired with Comchoano-2 sequences (clones). However, only one (probe “14”) of the Comchoano-2 probes were specific, as Comchoano-2 probe “2” also amplified in Comchoano-1 clones. Comchoano-2 probe “14” can be used individually to target only Comchoano-2.”

I thank the authors for going to the trouble of giving Candidatus names to their bacteria. According to my colleague with expertise in etymology, the names are fine with the following feedback: i) *C. bicostacola* should either be *C. bicostae* as a genitive form of the generic name or *C. bicosticola*, and ii) *Symchoanobacter* should be corrected to *Synchoanobacter*.

We thank the reviewer for their feedback on our names, and we have updated our Protologues with our new proposed names in accordance with this expert guidance. (Supplementary Information)

Final Decision Letter:

Dear Professor Worden,

Once again, I apologize for the delay! I am finally pleased to accept your Article "The microbiome of a bacterivorous marine choanoflagellate contains a resource-demanding obligate bacterial associate" for publication in Nature Microbiology. Thank you for having chosen to submit your work to us and many congratulations. It was a pleasure working with you and David, even if only for the final few steps of the process. Congrats again on an excellent paper.

9Acceptance of your manuscript is conditional on all authors' agreement with our publication policies (see <https://www.nature.com/nmicrobiol/editorial-policies>). In particular your manuscript must not be published elsewhere and there must be no announcement of the work to any media outlet until the publication date (the day on which it is uploaded onto our website).

Please note that *Nature Microbiology* is a Transformative Journal (TJ). Authors may publish their research with us through the traditional subscription access route or make their paper immediately open access through payment of an article-processing charge (APC). Authors will not be required to make a final decision about access to their article until it has been accepted. [Find out more about Transformative Journals](https://www.springernature.com/gp/open-research/transformative-journals)

Authors may need to take specific actions to achieve [compliance](https://www.springernature.com/gp/open-research/funding/policy-compliance-faqs) with funder and institutional open access mandates. If your research is supported by a funder that requires immediate open access (e.g. according to [Plan S principles](https://www.springernature.com/gp/open-research/plan-s-compliance)) then you should select the gold OA route, and we will direct you to the compliant route where possible. For authors selecting the subscription publication route, the journal's standard licensing terms will need to be accepted, including [self-archiving policies](https://www.nature.com/nature-portfolio/editorial-policies/self-archiving-and-license-to-publish). Those licensing terms will supersede any other terms that the author or any third party may assert apply to any version of the manuscript.
